# Learning and Covering Sums of Independent Random Variables with Unbounded Support

**Alkis Kalavasis**
National Technical University of Athens
`kalavasisalkis@mail.ntua.gr`

**Konstantinos Stavropoulos**
The University of Texas at Austin
`kstavrop@utexas.edu`

**Manolis Zampetakis**
University of California, Berkeley
`mzampet@berkeley.edu`

## Abstract

We study the problem of covering and learning sums $X = X_1 + \cdots + X_n$ of independent integer-valued random variables $X_i$ (SIIRVs) with *infinite* support. De et al. [2018] at FOCS 2018, showed that even when the collective support of $X_i$'s is of size $4$, the maximum value of the support necessarily appears in the sample complexity of learning $X$. In this work, we address two questions: (i) Are there general families of SIIRVs with infinite support that can be learned with sample complexity independent of both $n$ and the maximal element of the support? (ii) Are there general families of SIIRVs with infinite support that admit proper sparse covers in total variation distance? As for question (i), we provide a set of simple conditions that allow the infinitely supported SIIRV to be learned with complexity $\mathrm{poly}(1/\epsilon)$ bypassing the aforementioned lower bound. We further address question (ii) in the general setting where each variable $X_i$ has unimodal probability mass function and is a different member of some, possibly multi-parameter, *exponential family* $\mathcal{E}$ that satisfies some structural properties. These properties allow $\mathcal{E}$ to contain heavy tailed and non log-concave distributions. Moreover, we show that for every $\epsilon > 0$, and every $k$-parameter family $\mathcal{E}$ that satisfies some structural assumptions, there exists an algorithm with $\widetilde{O}(k) \cdot \mathrm{poly}(1/\epsilon)$ samples that learns a sum of $n$ arbitrary members of $\mathcal{E}$ within $\epsilon$ in TV distance. The output of the learning algorithm is also a sum of random variables within the family $\mathcal{E}$. En route, we prove that any discrete unimodal exponential family with bounded constant-degree central moments can be approximated by the family corresponding to a bounded subset of the initial (unbounded) parameter space.

## 1 Introduction

In this paper, we revisit the problem of learning distributions of the form $X = X_1 + \ldots + X_n$, where $n \in \mathbb{N}$ and the terms $X_i$ are independent integer random variables. We focus on the cases where each $X_i$ has unbounded, even infinite support. Our work follows the literature of learning distributions from independent samples (see e.g., [Dasgupta, 1999, Rabani et al., 2014, Acharya et al., 2015, Canonne, 2015, 2020, Diakonikolas et al., 2019, Moitra and Valiant, 2010]), that has been introduced in Kearns et al. [1994]. In this problem, we observe independent samples from a random variable $X$, which is a priori known to belong to a class of distributions $\mathcal{C}$, and the goal is to compute another random variable $X'$ such that $d_{TV}(X, X') \leq \epsilon$. The main question to ask follows: Given $\mathcal{C}$, how many samples from $X$ do we need to compute the estimate $X'$? If the output $X'$ belongs to $\mathcal{C}$, we say that we have *properly learned $X$*.

The problem of learning distributions is closely related to the problem of *sparsely covering* a class of distributions. Given a class $\mathcal{C}$ of distributions, an $\epsilon$-*cover* for this class is a set $\mathcal{C}_\epsilon$ of distributions such that, for every $D \in \mathcal{C}$, there exists a $D' \in \mathcal{C}_\epsilon$ such that $d_{TV}(D, D') \leq \epsilon$. If $\mathcal{C}_\epsilon \subseteq \mathcal{C}$, then $\mathcal{C}_\epsilon$ is called a *proper cover*. Clearly, the existence of a (small) cover for a class $\mathcal{C}$ is interesting by its own. Furthermore, once we have designed a cover $\mathcal{C}_\epsilon$, then there exist generic algorithms, e.g., the tournament procedure [Daskalakis and Kamath, 2014], that uses $\mathcal{C}_\epsilon$ to produce a learning algorithm with sample complexity $O(\log(|\mathcal{C}_\epsilon|)/\epsilon^2)$ and running time $\widetilde{O}(|\mathcal{C}_\epsilon|/\epsilon^2)$.

A fundamental problem in distribution learning arises when the elements of the class $\mathcal{C}$ can be expressed as a sum $X = X_1 + \cdots + X_n$ of $n \in \mathbb{N}$ independent but not identical random variables (SIIRVs). This problem has been extensively studied in the Theoretical Computer Science literature in the last decade. The seminal work of Daskalakis and Papadimitriou [2015], Daskalakis et al. [2015a], Diakonikolas et al. [2016a] settled the fundamental problem of covering and learning sums $X$ of independent but not identical Bernoulli random variables, where they prove the surprising result that the number of samples needed for learning the random variable $X$ in this case is independent of $n$ and almost the same as the number of samples needed to learn a single Bernoulli random variable. Subsequently, Daskalakis et al. [2013], Diakonikolas et al. [2016b] solved the problem of learning sums of integer random variables with support from 0 to $m - 1$ and otherwise follow arbitrary distribution using $\widetilde{O}(m/\epsilon^2)$ samples (again independent of $n$). Follow-up works have also considered multidimensional distributions again with bounded support size [Daskalakis et al., 2015b, 2016, Diakonikolas et al., 2016c]. This line of work has found applications in Game Theory for computing equilibrium in anonymous games [Daskalakis et al., 2016, Diakonikolas et al., 2016c, Goldberg and Turchetta, 2017, Cheng et al., 2017], in Mechanism Design for designing auctions [Goldberg and Tang, 2015], and in Stochastic Optimization [De, 2018]. Crucially, such applications make use of the delicate structure of such sums (reflected in proper sparse covers) and are not necessarily implied by the learning results. In fact, learning SIIRVs could also be seen as a fundamental (and not trivial) application of the corresponding covering results.

All the previous work in this literature considered learning sums of random variables whose support is bounded in size and the maximum elements in the support are also bounded. In particular, Daskalakis et al. [2013] observed that if the support of the terms is unbounded, then the sample complexity of learning the distribution of the sum will depend on the number of terms in the worst case, even under the assumption that the terms have bounded moments. Moreover, the recent work of De et al. [2018] showed that, even when the size of the support is 4, there should be a dependence of sample complexity on the maximum value of their support in general. In many settings though, both in Game Theory and in Stochastic Optimization, it is natural to encounter random variables with large or even infinite support. In these cases, any algorithm whose sample or time complexity depends on the support size or the maximum value of the random variable can be very inefficient or even useless. The above discussion gives rise to our first challenge:

**Challenge 1** (Infinite Support). *When is it possible to learn SIIRVs with infinite support with a number of samples independent of $n$ and the maximum element of the support?*

Note that, in our setting, the bounds of De et al. [2018] are only interesting from a qualitative perspective, since they focus on the (very weak) dependence of the sample complexity on the maximum value of the collective support (for which they provide tight bounds), but they enable doubly exponential dependence on the size of the collective support in their upper bounds.

Proper sparse covering (and hence proper learning due to the covering method), is a quite delicate requirement: To the best of our knowledge, the only known results for properly covering sums of independent univariate random variables, apply to the class of Poisson Binomial distributions (Daskalakis and Papadimitriou [2015], Diakonikolas et al. [2016c]) and the class of $m$-SIIRVs (each summand is supported on 0 to $m - 1$) (Diakonikolas et al. [2016a]). This is the second challenge:

**Challenge 2** (Proper Covers). *Are there general families of SIIRVs with infinite support that admit* proper *sparse covers in total variation distance?*

## 1.1 Our Contribution

We initiate the study of SIIRVs with unbounded and even infinite support (SIIURVs). There are two aspects of our work. First, we overcome the aforementioned lower bounds with an appropriate set of simple assumptions. Under our assumptions, we prove that the sample complexity of the

learning problem is independent from the number of terms, giving an answer to Challenge 1 (see Section 2). Our result is important from a theoretical perspective, since in the distribution learning setting there is a lack of a tight combinatorial characterization of the sample complexity, unlike, for example, the case of binary classification. The standard upper bound, metric entropy, includes, in our case, a dependence on the number of terms. Second, we give an answer to Challenge 2 (and then to Challenge 1) by properly covering (and then learning) SIIRVs with structured distributions. In particular, we focus on SIIRVs where each term is a member of a given exponential family of distributions $\mathcal{E}$, which we call $\mathcal{E}$-SIIRVs or SIIERVs. The exponential family paradigm is a multi-parametric, extremely expressive framework that captures many interesting families of distributions. Our results identify delicate structural properties for a quite broader class of random variables than what has been previously known and, importantly, demonstrate that the size of the support is not an utter impediment in acquiring such delicate results. We present our result on SIIERVs in Section 3.

**Results for SIIERVs.** Our main results concern the family $\mathcal{E}$-SIIRV of $X = X_1 + ... + X_n$. Each $X_i$ is a member of an exponential family of distributions $\mathcal{E}$, that is the probability mass function of $X_i$ at the point $x \in \mathbb{Z}$ is proportional to the quantity $\exp(-\boldsymbol{a}_i \cdot \boldsymbol{T}(x))$, where $\boldsymbol{T} : \mathbb{Z} \to \mathbb{R}^k$ is the vector of *sufficient statistics* of $\mathcal{E}$ and $\boldsymbol{a}_i$ is the vector of parameters of $X_i$ that belongs to the *parameter space* $\mathcal{A} \subseteq \mathbb{R}^k$ of $\mathcal{E}$. So in our setting, for every $X_i$ the sufficient statistics $\boldsymbol{T}$ are the same but the parameter vector $\boldsymbol{a}_i$ is different for every $i$. The sum $X$ will be called an $\mathcal{E}_{\boldsymbol{T}}(\mathcal{A})$-SIIRV of order $n$.

**Informal Assumptions 1** (Assumption 2). *Assume that there exist constants $L, B, \gamma, \Lambda > 0$ so that the exponential family $\mathcal{E} = \mathcal{E}_{\boldsymbol{T}}(\mathcal{A})$ is well-defined and:*

1. *(Geometry) $\mathcal{A}$ is closed, path-connected and its conical hull is a polyhedral cone.*

2. *(Modes) Every distribution in $\mathcal{E}$ is unimodal and the modes lie in $[-L, L]$.*

3. *(Bounded Moment) Every distribution in $\mathcal{E}$ has fourth central moment at most $B$.*

4. *(Variance) The variance of each distribution in $\mathcal{E}$ is lower bounded by $\gamma$.*

5. *(Covariance) For any $\boldsymbol{a}$ in the convex hull of $\mathcal{A}$, it holds $\mathbf{Cov}_{\boldsymbol{a}}(\boldsymbol{T}(W)) \preceq \Lambda \cdot I_k$.*

For a discussion on the minimality of our assumptions, we refer to Sections 2 and 3. In the next results, the set $\mathcal{A}'$ is a superset of $\mathcal{A} \subseteq \mathbb{R}^k$ (see the discussion after the statements).

**Informal Theorem 1** (Weakly-Proper Covering Theorem 3). *Under Assumption 1, for any $\epsilon > 0$, there exists a set of distributions $\mathcal{C}_\epsilon$ that $\epsilon$-covers the family of $\mathcal{E}_{\boldsymbol{T}}(\mathcal{A})$-SIIRVs of order $n$ in total variation distance. The set $\mathcal{C}$ has size $(n/\epsilon)^{O(k)} + 2^{k \cdot \mathrm{poly}(1/\epsilon)}$ and each element of $\mathcal{C}_\epsilon$ is an $\mathcal{E}_{\boldsymbol{T}}(\mathcal{A}')$-SIIRV of order $\Theta(n)$.*

**Informal Theorem 2** (Learning Theorem 4). *Under Assumption 1, given $m = k \cdot \widetilde{O}(1/\epsilon^2)$ samples from an unknown $\mathcal{E}_{\boldsymbol{T}}(\mathcal{A})$-SIIRV $X$ of order $n$, there exists an algorithm that outputs $\widehat{X}$ so that $d_{TV}(X, \widehat{X}) \leq \epsilon$ with high probability. Moreover, $\widehat{X}$ is an $\mathcal{E}_{\boldsymbol{T}}(\mathcal{A}')$-SIIRV of order $\Theta(n)$.*

**Weakly-Proper Covering.** We say that a cover is a weakly-proper cover for the family of $\mathcal{E}_{\boldsymbol{T}}(\mathcal{A})$-SIIRVs of order $n$ if its elements belong to the family of $\mathcal{E}_{\boldsymbol{T}}(\mathcal{A}')$-SIIRVs with parameters in a slightly larger set $\mathcal{A}' \subseteq \mathbb{R}^k$ and with possibly more than $n$ terms. In the rest of the paper, we mostly stick with the term proper for brevity (see also Appendix A.1.1). We think of $\mathcal{A}$ as input to the problem, but we focus on the various challenges arising by the nature of this problem instead of possible adversarial selections of $\mathcal{A}$.

The covering result gives an answer to Challenge 2 and the learning one provides an algorithm with sample complexity independent of $n$ and the maximum element of the support (Challenge 1). In the above informal theorem, we have treated the relevant parameters of the exponential family $\mathcal{E}$ (e.g., $B$) as constants. If we also consider the accuracy $\epsilon$ to be constant, the learner runs in time $n^{O(k)}$. The assumptions about the central moments as well as the covariance matrix are standard. The assumption regarding the geometry of $\mathcal{A}$ is a mild, technical assumption, that has, however, important technical implications. The variance lower bound is a substitute of particularly subtle – and most probably not omnipotent – technical tools that can be used to discard low variance terms in special cases. Finally, the assumption about the modes provides the structure needed to apply the most powerful tool we possess to confront Challenge 1: quantitative versions of the Central Limit Theorem. Moreover, Challenge 2 restricts the flexibility we have in applying such a tool, which we believe to indicate that our assumption about modes is, in some sense, essential for our purposes.

Our Assumptions 1 do not exclude any reasonable exponential family and our methods capture (among others) Geometric, Bernoulli, Poisson, Zeta, Gamma, Gaussian, Laplacian distributions and interpolations thereof (see Appendix H.1). For instance, our results apply to sums with both Gaussian and Laplacian terms. In particular, the naturally occuring Zeta distributions (Zipf's law [Chao and Zipf, 1949]) are not log-concave and no non-trivial learning results are known on sums thereof, even without requiring proper learning.

**Technical Contributions.** For a more detailed discussion about our novel technical features, we refer to Sections 3.3 and 3.4. First, we provide a fundamental structural result about exponential families that satisfy our assumptions, by reducing the problem of properly sparsely covering an exponential family as such to the standard problem of covering a bounded subset of $\mathbb{R}^k$ (for some $k \in \mathbb{N}$) in Euclidean distance. The main challenge, which familiar results about exponential families do not resolve, is that the parameter space of the exponential family may be unbounded. We show that a bounded subset of the parameter space approximately generates any distribution in the family (Theorem 6), which we believe to be of independent interest. To this end, we identify an analogy between the geometry of exponential families and polyhedral theory; we essentially reduce probabilistic properties of exponential family distributions to geometric properties of polyhedral cones, for which we settle a novel result (Theorem 8). We also prove that for any distribution with parameter vector that has a sufficiently large (yet bounded) norm, the number of important points of the support is bounded, which leads to the resolution of the main technical challenge. Secondly, we provide a continuity argument which implies that for any $\mathcal{E}$-SIIRV $X$ of order $n$, there exists some $\mathcal{E}$-SIIRV $Y$ which is the sum of i.i.d. random variables in the family $\mathcal{E}$ such that the distance between the expectation of $X$ and the expectation of $Y$, as well as the distance between the variance of $X$ and the variance of $Y$ are bounded. This is important in order to prevent our learning algorithm from running in time exponential to the number of terms $n$. Moreover, our proofs require meticulously selecting and handling tools from previous work, as well as accounting for various technical details.

**Limitations and Impact.** Our results are of theoretical nature and we do not identify any direct potential negative societal impact.

## 2 Warm-Up: Structure and Learning of SIIURVs

We show, as a warm-up, that there exists a set of assumptions under which learning in total variation distance the distribution of an unknown sum of (at most) $n$ independent random variables with possibly unbounded support can be done using a number of independent samples that does not increase with $n$. In particular, let $\mathcal{D}$ be a family of distributions over $\mathbb{Z}$. We consider sums of independent integer-valued random variables of order $n$ of the form $X = \sum_{i \in [n']} X_i$, where $n' \le n$, $X_i \sim D_i \in \mathcal{D}$. We call $X$ a $\mathcal{D}$-SIIRV (or SIIURV, i.e., sum of independent integer random variables with unbounded support) of order $n$.

**Assumption 1.** *We make the following assumptions for the family of distributions $\mathcal{D}$.*

1. *Every distribution in $\mathcal{D}$ is* (1a) *unimodal and* (1b) *the mode is assigned probability at most equal to $1 - \gamma$, for some constant $\gamma \in (0, 1)$ (common for all distributions in $\mathcal{D}$).*[1]

2. *Every mode of any distribution in $\mathcal{D}$ lies within a (common) interval of constant length $L > 0$.*

3. *The fourth central moment $\mathbf{E}\left[|W - \mathbf{E}[W]|^4\right]$ of each distribution in $\mathcal{D}$ is upper bounded by a constant $B > 0$ (uniformly for all distributions in $\mathcal{D}$).*

**Minimality of Assumption 1.** Removing condition 1 (unimodality and a bound on the mass assigned on the mode) would activate a lower bound on the sample complexity (Observation 1.3 from Daskalakis et al. [2013]) that involves some dependence on the number of terms $n$; we aim for sample complexity independent from $n$. The terms considered in the lower bound all have zero as a mode (essentially satisfying condition 2 in the case of multimodal distributions) and all of their moments are upper bounded by a sequence of values (stronger than condition 3). They are, however, not unimodal and they assign almost all of their mass to zero (condition 1 does not hold). Waiving condition 2 enables one to form a family which does not have a sparse cover, even when the sums

---

[1] A unimodal distribution could have many consequent modes, each assigned equal amount of mass.

only have a single term. In particular, we can consider a sequence of arbitrarily shifted Bernoulli distributions with parameter $1/2$, each of which has a distance at least equal to $1/2$ from any other distribution of the sequence. Moreover, the aforementioned sequence does not violate conditions 1 or 3. Finally, condition 3 is important, since it rules out, for example, the case of $\mathcal{D}$ containing all Geometric distributions (enabling probability of success arbitrarily close to 0). In this case, there is some constant $\epsilon > 0$ such that we may consider an infinite sequence of geometric distributions with diminishing success probabilities $p_n = 2^{-n}$ with pairwise statistical distance at least $\epsilon$. The degree of the moment we assume to be bounded is $4$, which is useful in the dense case, to establish the rate of convergence of the sum to a discretized Gaussian distribution; importantly, the degree is constant. We get the following result.

**Theorem 1** (Learning). *Set $n \in \mathbb{N}$ and $\mathcal{D}$ some family of distributions satisfying Assumption 1. Let $\epsilon, \delta \in (0, 1)$ and $X$ be an unknown $\mathcal{D}$-SIIRV of order $n$. There exists an algorithm (Figure 1) with the following properties: Given $n, \epsilon, \delta, L, B, \gamma$ and sample access to $X$, the algorithm uses $m = O(\frac{1}{\epsilon^2} \cdot \log(1/\delta)) + O(\text{poly}(B, 1/\gamma, 1/\epsilon) \cdot \log(L))$ samples from $X$ and, in time $\text{poly}(m, L^{\text{poly}(B,1/\gamma,1/\epsilon)})$, outputs a (succint description of a) distribution $\widetilde{X}$ with $d_{TV}(X, \widetilde{X}) \leq \epsilon$, with probability $1 - \delta$.*

Theorem 1 is based on a common technique used in problems related to SIIRVs, which uses quantitative versions of the Central Limit Theorem (like Lemma 19 of Chen et al. [2010]) to reduce the learning problem into two sub-problems; covering $\mathcal{D}$ in total variation distance and estimating the variance and expectation of the unknown SIIRV.

**Theorem 2** (Structure of SIIURVs). *Set $n \in \mathbb{N}$ and $\mathcal{D}$ some family of distributions satisfying Assumption 1. For any $\epsilon > 0$, and any $\mathcal{D}$-SIIRV $X$ of order $n$, there exists some $Y$ such that $d_{TV}(X, Y) \leq \epsilon$ and either (i) $Y$ is a random variable among $L^{\text{poly}(B,1/\gamma,1/\epsilon)}$ candidates that are independent from the particular $X$ (sparse form) or (ii) $Y$ is a discretized Gaussian random variable with $\mathbf{E}[X] = \mathbf{E}[Y]$ and $\mathbf{Var}(X) = \mathbf{Var}(Y)$ (dense form).*

Hence, the learner first runs two different learning procedures, corresponding to the sparse and dense forms of Theorem 2. For the sparse case, it runs a tournament over the possible candidates and in the dense one, it computes the parameters of the (potentially) nearby discretized Gaussian. From the two procedures, two hypotheses are obtained and finally hypothesis testing is performed in order to select the correct one. Our focus on the Gaussian approximation is the reason why we assumed that condition (1) holds. In principle, there might be ways to relax Assumption 1 and learn SIIURVs (independently from $n$) with different techniques, but we are particularly interested in using the Gaussian approximation in the dense case (compare, e.g., with the approach of Daskalakis et al. [2013]), since it will be pivotal to our main technical and conceptual contribution outlined in the following section.

## 3 Structure and Proper Learning of SIIERVs

In the seminal work of Daskalakis and Papadimitriou [2015], it was shown that the class of Poisson Binomial Distributions (i.e., sums of independent indicator random variables) has a structure with similar properties as the one presented in Theorem 2. Crucially, however, their results had an additional property. In both sparse and dense cases, the candidate distributions (i.e., the representatives of the class) were Poisson Binomial Distributions themselves (namely, the class admitted *proper sparse covers*). The result unlocked the possibility of *proper* learning for PBDs (see Daskalakis et al. [2015a], Diakonikolas et al. [2016c]), with sample complexity independent from the number of terms. Besides the result of Daskalakis and Papadimitriou [2015], to the best of our knowledge, there are *no further known results for properly covering sums of integer-valued (univariate) random variables, with terms in some structured-parametric family of distributions*.[2]

We provide significantly general results for the structure of sums of integer (and unbounded) random variables, under the condition that the terms belong in any fixed exponential family that satisfies a set of assumptions. Our results imply proper learning with sample complexity independent from the number of terms. We consider exponential families supported on the whole (unbounded) set of integer numbers, although our results could be extended to exponential families supported on some subset of $\mathbb{Z}$, like $\mathbb{N}_0$.

---

[2]Diakonikolas et al. [2016a] provide proper sparse covers for the class of $m$-SIIRVs; nevertheless, this family does not have the "structure" we focus on this work since it is nonparametric.

### 3.1 Preliminaries and Definitions

**Exponential Families.** For $k \in \mathbb{N}_0$, $\mathcal{A} \subseteq \mathbb{R}^k$ and $\boldsymbol{T} : \mathbb{Z} \to \mathbb{R}^k$, we denote with $\mathcal{E}_{\boldsymbol{T}}(\mathcal{A})$ the **exponential family** with **sufficient statistics** $\boldsymbol{T}$ and **parameter space** $\mathcal{A}$. If $W \sim \mathcal{E}_{\boldsymbol{T}}(\boldsymbol{a})$ for some $\boldsymbol{a} \in \mathcal{A}$, then $\mathbf{Pr}[W = x] \propto \exp(-\boldsymbol{a} \cdot \boldsymbol{T}(x))$ for any $x \in \mathbb{Z}$. We will use $\mathbf{Pr}_{\boldsymbol{a}}[W = x]$ (similarly $\mathbf{E}_{\boldsymbol{a}}[W]$ and $\mathbf{Var}_{\boldsymbol{a}}(W)$ for expectation and variance correspondingly) to refer to the probability that $W = x$ given that $W \sim \mathcal{E}_{\boldsymbol{T}}(\boldsymbol{a})$, whenever it is clear that the distribution of $W$ belongs in $\mathcal{E}_{\boldsymbol{T}}(\mathcal{A})$. [3]

**SIIERVs.** We will consider distributions of sums of the form $X = \sum_{i \in [n']} X_i$, where $n' \leq n$, $(X_i)_i$ independent and $X_i \sim \mathcal{E}_{\boldsymbol{T}}(\boldsymbol{a}_i)$ with $\boldsymbol{a}_i \in \mathcal{A}$. We call this class of distributions as $\mathcal{E}_{\boldsymbol{T}}(\mathcal{A})$**-SIIERVs of order** $n$ or simply SIIERVs when $n, \mathcal{A}$ and $\boldsymbol{T}$ are clear by the context.

**Set Operators.** We say that $\mathtt{Op}$ is an extensive set operator on $\mathbb{R}^k$ if for any set $\mathcal{A} \subseteq \mathbb{R}^k$, we have that $\mathcal{A} \subseteq \mathtt{Op}\mathcal{A} \subseteq \mathbb{R}^k$. We use the following extensive set operators: For the definitions of the **convex hull** $\mathtt{Conv}$ and the **conical hull** $\mathtt{Cone}$ operators, we refer to (1) and (2). The $\varrho$**-conical hull** operator $\varrho\text{-}\mathtt{Cone}\mathcal{A} = \mathcal{A} \cup (\mathtt{Cone}\mathcal{A} \setminus \mathbb{B}_\varrho(\boldsymbol{0}))$, i.e., the $\varrho\text{-}\mathtt{Cone}$ operator inserts in $\mathcal{A}$ all points of the conical hull of $\mathcal{A}$ with norm at least $\varrho$. For additional notation, we refer to the Appendix A.1.

### 3.2 Main results

Our results for $\mathcal{E}_{\boldsymbol{T}}(\mathcal{A})$-SIIERVs hold under the following set of assumptions about $\mathcal{E}_{\boldsymbol{T}}(\mathcal{A})$.

**Assumption 2.** *Let $k \in \mathbb{N}$, $\boldsymbol{T} : \mathbb{Z} \to \mathbb{R}^k$ and $\mathcal{A} \subseteq \mathbb{R}^k$. Denote $\mathcal{A}_\varrho = \varrho\text{-}\mathtt{Cone}\mathcal{A}$ and $\overline{\mathcal{A}}_\varrho = \mathtt{Conv}\mathcal{A}_\varrho$ for $\varrho > 0$. We assume that there exists some constant $\varrho > 0$ so that the exponential family $\mathcal{E}_{\boldsymbol{T}}(\mathcal{A}_\varrho)$ is well-defined and the following hold:*

1. *The parameter space $\mathcal{A}$ is closed and its conical hull $\mathtt{Cone}\mathcal{A}$ is a polyhedral cone.*

2. *Every distribution in $\mathcal{E}_{\boldsymbol{T}}(\mathcal{A}_\varrho)$ is unimodal.*

3. *There exists some constant $L > 0$ such that every mode of every distribution in $\mathcal{E}_{\boldsymbol{T}}(\mathcal{A}_\varrho)$ lies within the interval $[-L, L]$.*

4. *There exists some constant $B > 0$ such that every distribution in $\mathcal{E}_{\boldsymbol{T}}(\mathcal{A}_\varrho)$ has fourth central moment that is upper bounded by $B$, i.e., $\mathbf{E}_{\boldsymbol{a}}\left[ |W - \mathbf{E}_{\boldsymbol{a}}[W]|^4 \right] \leq B$, for any $\boldsymbol{a} \in \mathcal{A}_\varrho$.*

5. *There exists some constant $\Lambda > 0$ such that $\mathbf{Cov}_{\boldsymbol{a}}(\boldsymbol{T}(W)) \preceq \Lambda \cdot I_k$, for any $\boldsymbol{a} \in \overline{\mathcal{A}}_\varrho$.*

6. *The parameter space $\mathcal{A}$ is path-connected.*

7. *There exists some constant $\gamma > 0$ such that $\mathbf{Var}_{\boldsymbol{a}}(W) \geq \gamma$, for any $\boldsymbol{a} \in \mathcal{A}$.*

Assumptions (2)-(4) correspond to conditions in Assumption 1, but we make some additional ones. In particular, variants of assumption (5) have been used in the past (see Diakonikolas et al. [2021]), e.g., for parameter estimation in exponential families and essentially ensure that parameter vectors close in Euclidean distance correspond to distributions close in statistical distance. Assumptions (6) and (7) are important only in the case that the number of terms $n$ in the sum is large. Assumption (7) ensures that as the number of terms increases, the distribution approaches its limit (i.e., the discretized Gaussian distribution), with some constant rate. In fact, the variance lower bound is a substitute of particularly subtle but specialized technical tools that can be used to discard low variance terms in some specific cases (e.g., see the so called massage step of Daskalakis and Papadimitriou [2015] which refers to Poisson Binomial Distributions and our own Appendix H.2, which refers to sums of independent geometric RVs, namely, Poisson Negative Binomial Distributions). Finally, assumption (6) implies some kind of continuity with respect to the parameter vector which is important for proper learning so that the behavior of a sum of a large number of terms can be described by a constant number of parameter vectors (in our case exactly one). For a discussion on the verification of the assumptions, see Appendix H.4.

---

[3] In general, an exponential family $\mathcal{E}$ over $\mathbb{Z}$ is also defined in terms of some carrier measure $h : \mathbb{Z} \mapsto \mathbb{R}_+$ so that if $W \sim \mathcal{E}$, then $\mathbf{Pr}[W = x] \propto h(x) \cdot \exp(-\boldsymbol{a} \cdot \boldsymbol{T}(x))$ and denote $W \sim \mathcal{E}_{\boldsymbol{T},h}(\boldsymbol{a})$. We can reduce this setting to $h \equiv 1$ by considering $\mathcal{A}' = \mathcal{A} \times \{1\}$ and $\boldsymbol{T}' = (\boldsymbol{T}, -\log_e(h(x)))$.

We stress that assumptions (2)-(5) are imposed on a slightly expanded exponential family (by extending the parameter space $\mathcal{A}$ to $\mathcal{A}_\varrho$). In fact, our analysis involves the study of the influence that changes in the parameter vector have on the corresponding distribution, given some properties generated by our assumptions. Extending the space on which such properties hold enables the study of a wider range of changes of the parameter vector. On the contrary, assumption (7) is imposed on $\mathcal{A}$ (and not its extended version). It essentially excludes some parameter vectors with large norms, but, in general, it *does not imply* that $\mathcal{A}$ is bounded. In general, $\mathcal{A}$ should be though of as the space of parameters for the input and $\mathcal{A}_\varrho (\supseteq \mathcal{A})$ the space of parameters for the output.

**Theorem 3** (Structure of SIIERVs). *Set $n \in \mathbb{N}$ and $\mathcal{E}_{\boldsymbol{T}}(\mathcal{A})$ some exponential family satisfying Assumption 2. Let $\mathcal{A}_\varrho = \varrho\text{-}\mathrm{Cone}\mathcal{A}$. There exists some value $\theta = \theta(\mathcal{A}, \boldsymbol{T}) > 0$ such that, for any $\epsilon \in (0, 1)$ and any $\mathcal{E}_{\boldsymbol{T}}(\mathcal{A})$-SIIRV $X$ of order $n$, there exists some random variable $Y$ such that $d_{TV}(X, Y) \leq \epsilon$ and either (i) $Y$ is an $\mathcal{E}_{\boldsymbol{T}}(\mathcal{A}_\varrho)$-SIIRV among $(\frac{\varrho \cdot \sqrt{\Lambda}}{\theta})^{k \cdot \widetilde{O}(\frac{1}{\epsilon^2}) \cdot \mathrm{poly}(B, L, \frac{1}{\gamma})}$ candidates (sparse form) or (ii) $Y$ is a sum of i.i.d. $\mathcal{E}_{\boldsymbol{T}}(\mathcal{A}_\varrho)$-random variables among $(n^2 \cdot \mathrm{poly}(B, \frac{1}{\gamma}) \cdot O(\frac{\varrho \cdot \sqrt{\Lambda}}{\theta \cdot \epsilon}))^k$ candidates (dense form).*

The role of the quantity $\theta$ is thoroughly discussed in Appendix E.1. Roughly, the parameter space $\mathcal{A}$ and the trajectory of the sufficient statistics $\boldsymbol{T}(x)$, $x \in \mathbb{Z}$ are associated with a finite number of polyhedral cones which are important parts of the structure of the family $\mathcal{E}_{\boldsymbol{T}}(\mathcal{A})$. The value of $\theta$ depends on the geometry of the specified polyhedral cones.

The structural result implies a proper learning algorithm (Figure 2) which roughly applies the tournament method and hypothesis selection routines over the cover both in the sparse and (after an additional important step) in the dense case (see Proposition 25). For the learning result, we assume access to some sample and evaluation oracles (see Appendix B.2). Such access is needed in order to apply hypothesis testing over our covers in a formal sense. Let $D$ be a distribution over $\mathbb{Z}$. Consider the sample oracle $\mathrm{EX}(D)$ that, when invoked, returns a sample with law $D$ and the approximate evaluation oracle $\mathrm{EVAL}_D(\beta)$ that, when invoked with query $x \in \mathbb{Z}$, returns a value $q$ that satisfies $D(x)/(1 + \beta) \leq q \leq (1 + \beta)D(x)$ for some $\beta > 0$ (this oracle is used in De et al. [2014]). Below, the relation between $\beta$ and the desired learning accuracy $\epsilon$ is provided by De et al. [2014]. Finally, assume that the cover of Theorem 3 (the set of candidate distributions) of radius $\epsilon$ can be constructed in time $T_c = T_c(\mathcal{A}, n, \epsilon, \varrho, L, B, \Lambda, \gamma, \theta, \boldsymbol{T})$ (see Remark 2 for a discussion on the runtime).

**Theorem 4** (Learning SIIERVs). *Set $n \in \mathbb{N}$ and $\mathcal{E}_{\boldsymbol{T}}(\mathcal{A})$ some exponential family satisfying Assumption 2. Let $\mathcal{A}_\varrho = \varrho\text{-}\mathrm{Cone}\mathcal{A}$. There exists $\theta = \theta(\mathcal{A}, \boldsymbol{T}) > 0$ such that for any $\epsilon, \delta \in (0, 1)$ there exists an algorithm (Figure 2) with the following properties: Given $n, \epsilon, \delta, B, L, \Lambda, \gamma, \theta$ and (i) sample access to an unknown $\mathcal{E}_{\boldsymbol{T}}(\mathcal{A})$-SIIRV $X$ of order $n$, (ii) $\mathrm{EX}(Z(\mu, \sigma^2))$ for any $\mu, \sigma^2$ and (iii) $\mathrm{EX}(D)$ and evaluation oracle access to $\mathrm{EVAL}_D(\beta)$ for any $D \in \mathcal{E}_{\boldsymbol{T}}(\mathcal{A}_\varrho)$ for some $\beta \geq 0$ with $(1 + \beta)^2 \leq 1 + \epsilon/8$, the algorithm uses $m = O(\frac{1}{\epsilon^2} \log(1/\delta)) + k \cdot \widetilde{O}(\frac{1}{\epsilon^2}) \cdot \mathrm{poly}(B, \frac{1}{\gamma}) \cdot \log(\frac{\varrho \cdot \Lambda}{\theta})$ samples from $X$ and, in time $\mathrm{poly}(m, 2^{k \cdot \widetilde{O}(1/\epsilon^2) \cdot \mathrm{poly}(B, L, 1/\gamma)}, n^k, (\varrho\Lambda/\theta)^k, T_c)$, outputs a (succint description of a) distribution $\widetilde{X}$ which satisfies $d_{TV}(X, \widetilde{X}) \leq \epsilon$, with probability $1 - \delta$. Moreover, $\widetilde{X}$ is an $\mathcal{E}_{\boldsymbol{T}}(\mathcal{A}_\varrho)$-SIIRV of order $(\sqrt{B}/\gamma) \cdot n$.*

In particular, for the dense case hypothesis, the learner runs in two steps. First, similarly to the learner of Theorem 1, it estimates the expectation and the variance of $X$ by $\widetilde{O}(1/\epsilon^2)$ samples, thereby specifying a discretized Gaussian that is close to $X$. However, as a second step, it runs the tournament hypothesis testing procedure between *the estimated Gaussian* and the candidate distributions of the dense form. Importantly, the tournament selection does not need to use real samples from $X$, but, instead, it generates draws from the Gaussian.

In the following sections, we analyze the structural result of Theorem 3, which is our main technical contribution. In Appendix H.2, we provide an example corollary of our methods for the case of Poisson Negative Binomial Distributions (Theorem 37).

### 3.3 Sparsifying the Parameter Space of an Exponential Family

We first solve the proper covering problem in the simplest possible case of $n = 1$, i.e., we provide sparse covers for any exponential family $\mathcal{E}_{\boldsymbol{T}}(\mathcal{A})$ satisfying (some of the assumptions in) Assumption 2. The following result constitutes the main building block of our analysis in the case of general $n$ (see Section 3.4). The proof of this Theorem can be found at the Appendix F.

**Theorem 5** (Sparsifying the Parameter Space). *Under assumptions* (1), (2), (3), (4) *and* (5), *there exists* $\theta = \theta(\mathcal{A}, \boldsymbol{T}) > 0$, *so that for any* $\epsilon \in (0, 1)$, *there exists* $\mathcal{B} \subseteq \varrho\text{-Cone}\mathcal{A}$ *with* $|\mathcal{B}| \leq (\widetilde{O}(\frac{\sqrt{\Lambda} \cdot \varrho}{\epsilon} + \frac{\sqrt{\Lambda}}{\epsilon \cdot \theta} + \frac{\sqrt{\Lambda}}{\epsilon \cdot \theta} \cdot \log(B)))^k$ *so that, for any* $\boldsymbol{a} \in \mathcal{A}$, $d_{TV}(\mathcal{E}_{\boldsymbol{T}}(\boldsymbol{a}), \mathcal{E}_{\boldsymbol{T}}(\boldsymbol{b})) \leq \epsilon$, *for some* $\boldsymbol{b} \in \mathcal{B}$.

The idea behind Theorem 5 has two parts. First, we show that, although our assumptions do not exclude the possibility that $\mathcal{A}$ is unbounded, there exists some bounded set $\mathcal{A}' \subseteq \mathcal{A}_\varrho := \varrho\text{-Cone}\mathcal{A}$ so that $\mathcal{E}_{\boldsymbol{T}}(\mathcal{A}')$ covers $\mathcal{E}_{\boldsymbol{T}}(\mathcal{A})$ in TV distance. Second, due to assumption (5), Lemma 14 and Pinsker's inequality, we may discretize $\mathcal{A}'$ (with standard sparse covers in Euclidean distance) to get a sparse cover for $\mathcal{E}_{\boldsymbol{T}}(\mathcal{A}')$ (which will also be a proper sparse cover for $\mathcal{E}_{\boldsymbol{T}}(\mathcal{A})$). In the rest of the current section, we will discuss about the main technical challenge, namely the proof of the first part of our idea, which is formally stated in the following theorem (for the proof, see Appendix E.1).

**Theorem 6** (Bounding the Parameter Space). *Under assumptions* (1), (2), (3) *and* (4), *there exists* $\theta = \theta(\mathcal{A}, \boldsymbol{T}) > 0$, *such that for any* $\epsilon \in (0, 1)$ *and any* $\boldsymbol{a} \in \mathcal{A}$, *there exists* $\boldsymbol{b} \in \varrho\text{-Cone}\mathcal{A}$ *with* $\|\boldsymbol{b}\| \leq (\varrho + \frac{1}{\theta}) \cdot \ln(1/\epsilon) + \frac{\ln(B)}{2\theta} + O(\varrho + \frac{1}{\theta})$ *so that* $d_{TV}(\mathcal{E}_{\boldsymbol{T}}(\boldsymbol{a}), \mathcal{E}_{\boldsymbol{T}}(\boldsymbol{b})) \leq \epsilon$.

In order to bound the parameter space, one has to analyze the behavior of a distribution $\mathcal{E}_{\boldsymbol{T}}(\boldsymbol{a}) \in \mathcal{E}_{\boldsymbol{T}}(\mathcal{A})$ as the norm of the parameter vector $\boldsymbol{a}$ increases (and its direction is fixed). Any point $x \in \mathbb{Z}$ is assigned by $\mathcal{E}_{\boldsymbol{T}}(\boldsymbol{a})$ mass proportional to $\exp(-\boldsymbol{a} \cdot \boldsymbol{T}(x))$. Let $\mathcal{M}_{\boldsymbol{a}}$ be the set of modes of $\mathcal{E}_{\boldsymbol{T}}(\boldsymbol{a})$, i.e., the set of (global) maximizers of its probability mass function. Note that rescaling $\boldsymbol{a}$ does not alter the positions of the modes, since the order of the quantities $\boldsymbol{a} \cdot \boldsymbol{T}(x)$ (for $x \in \mathbb{Z}$) is preserved. Moreover, as the norm of the parameter vector increases, the distribution $\mathcal{E}_{\boldsymbol{T}}(\boldsymbol{a})$ tends to become the uniform distribution over $\mathcal{M}_{\boldsymbol{a}}$.

It turns out, however, that the direction of $\boldsymbol{a}$ affects the rate with which $\mathcal{E}_{\boldsymbol{T}}(\boldsymbol{a})$ tends to its limit. Consider, for example, the case $\boldsymbol{T}(x) = (x, x^2)$ and $\boldsymbol{a}_\delta = r(1, 1 + \delta)$, for some $\delta > 0$ and $r > 0$. The distribution $\mathcal{E}_{\boldsymbol{T}}(\boldsymbol{a}_\delta)$ has a unique mode on $x = 0$, but as $\delta$ tends to 0, for any fixed $r$, the point $x = -1$ tends to become a mode of $\mathcal{E}_{\boldsymbol{T}}(\boldsymbol{a}_\delta)$. Therefore, it is possible that there exists some integer $x$ such that for any fixed parameter vector norm $r \geq \varrho$, there are directions $\widehat{\boldsymbol{a}}$ in $\mathcal{A}$ so that $\mathcal{E}_{\boldsymbol{T}}(r\widehat{\boldsymbol{a}})$ assigns to $x$ mass arbitrarily close to the mass it assigns to a mode. Bounding the parameter space is, hence, not straightforward, since there is no uniform (over the directions in $\mathcal{A}$) threshold for the parameter vector's norm upon which every distribution is close to its limit. In other words, *orthogonally* projecting the parameter vectors with large norms on any fixed radius sphere does not work. We, therefore, have to develop some further geometric intuition.

We claim that, under assumptions (1), (2), (3) and (4), there is a way to project (not necessarily orthogonally) any given parameter vector $\boldsymbol{a} \in \mathcal{A}$ with large norm on an origin-centered sphere with bounded radius so that the resulting distribution is close to $\mathcal{E}_{\boldsymbol{T}}(\boldsymbol{a})$ in TV distance. We prove our claim in two steps. *First,* we prove our claim with respect to a new notion of distance between distributions (instead of TV distance), which we call *structural distance*. For this step, we establish an interesting connection between exponential families and polyhedral cones. *Second,* we prove that by picking the radius of the sphere on which we project to be large enough, the bounds in structural distance from the previous step imply bounds in total variation distance.

We proceed with a formal definition of the structural distance, which is a distance metric (Lemma 10). In a nutshell, structural distance is the minimum possible relative threshold, such that two distributions agree (in relative terms) on every point of their domain with mass higher than the threshold (i.e., any significant point).

**Definition 1** (Structural Distance). *Consider two discrete distributions* $D_1, D_2$ *over* $\mathbb{X}$ *and let* $p_i = \max_{x \in \mathbb{X}} D_i(x)$, $i = \{1, 2\}$. *The structural distance* $d_{\mathrm{ST}}(D_1, D_2)$ *between* $D_1$ *and* $D_2$ *is the minimum* $\epsilon \in [0, 1]$ *such that for any* $x \in \mathbb{X}$, *at least one of the following holds:*

$$(i) \ \ D_1(x) \leq \epsilon \cdot p_1 \ \ \& \ \ D_2(x) \leq \epsilon \cdot p_2, \ \ \text{or} \ \ (ii) \ \ D_1(x)/p_1 = D_2(x)/p_2.$$

For any $\epsilon \in [0, 1]$, and any discrete distribution $D$ over $\mathbb{X}$, let $\widehat{D}^{(\epsilon)}$ denote the truncation of $D$ on the points $x$ such that $D(x) \geq \epsilon \cdot \max_{y \in \mathbb{X}} D(y)$. Then, the structural distance between two distributions $D_1$ and $D_2$ can be described as the minimum threshold $\epsilon \in [0, 1]$ so that the distributions $\widehat{D}_1^{(\epsilon)}$ and $\widehat{D}_2^{(\epsilon)}$ are identical. In that sense, structural distance measures the degree in which two distributions have the same structure (on significant points). Two distributions with different modes have structural distance 1 (maximum possible). Structural distance is meaningful when the two distributions

(or one of them) can be contrived to have the same structure on significant points, which we prove to be possible in the context of exponential families. We prove the following lemma.

**Lemma 7** (Structural Distance and Bounding Norms)**.** *Under assumptions* (1)*,* (2) *and* (3)*, there exists some constant $\theta > 0$ such that for any $r \geq \varrho$ and any $\boldsymbol{a} \in \mathcal{A}$ with $\|\boldsymbol{a}\| \geq r$, there exists some $\boldsymbol{b} \in \mathcal{A}_{\varrho}$ and $\|\boldsymbol{b}\| = r$ so that $d_{\mathrm{ST}}(\mathcal{E}_{\boldsymbol{T}}(\boldsymbol{a}), \mathcal{E}_{\boldsymbol{T}}(\boldsymbol{b})) \leq e^{-\theta \cdot r}$.*

The proof of Lemma 7 can be found in Appendix E.2. However, we will now outline the main ingredients of the proof. Let us translate Definition 1 in terms of the parameter vectors of exponential families. In particular, let $\boldsymbol{a} \in \mathcal{A}$ and $\boldsymbol{b} \in \mathcal{A}_{\varrho}$ and $M_{\boldsymbol{a}}$ (resp. $M_{\boldsymbol{b}}$) be any mode of $\mathcal{E}_{\boldsymbol{T}}(\boldsymbol{a})$ (resp. $\mathcal{E}_{\boldsymbol{T}}(\boldsymbol{b})$). Then $d_{\mathrm{ST}}(\mathcal{E}_{\boldsymbol{T}}(\boldsymbol{a}), \mathcal{E}_{\boldsymbol{T}}(\boldsymbol{b})) = \epsilon$ means that for any $x \in \mathbb{Z}$ either $\boldsymbol{a} \cdot (\boldsymbol{T}(x) - \boldsymbol{T}(M_{\boldsymbol{a}})) \geq \epsilon$ and $\boldsymbol{b} \cdot (\boldsymbol{T}(x) - \boldsymbol{T}(M_{\boldsymbol{b}})) \geq \epsilon$ ($x$ is not significant) or $\boldsymbol{a} \cdot (\boldsymbol{T}(x) - \boldsymbol{T}(M_{\boldsymbol{a}})) = \boldsymbol{b} \cdot (\boldsymbol{T}(x) - \boldsymbol{T}(M_{\boldsymbol{b}}))$ ($x$ is significant). Therefore, reducing the norm of $\boldsymbol{a}$ without moving significantly in structural distance corresponds to a geometric problem regarding the parameter and the sufficient statistics vectors. In particular, given some $\boldsymbol{a} \in \mathcal{A}$ with large norm, $\boldsymbol{b}$ should be chosen so that the quantities $\boldsymbol{b} \cdot \boldsymbol{v}_x$ for $x \in \mathbb{Z}$ and some sequence $(\boldsymbol{v}_x)_x$ of vectors that depends on $\boldsymbol{a}$ are constrained to be (i) equal to $\boldsymbol{a} \cdot \boldsymbol{v}_x$ when $\boldsymbol{a} \cdot \boldsymbol{v}_x$ is small and (ii) large when $\boldsymbol{a} \cdot \boldsymbol{v}_x$ is large. The number of constraints for $\boldsymbol{b}$ is infinite, since $x \in \mathbb{Z}$. However, due to unimodality, one can show that only a finite number of them is crucial (the others can be trivially satisfied); we then make use of ($\mathcal{A}_{\varrho} = \varrho\text{-}\mathrm{Cone}\mathcal{A}$ and) the following (to the best of our knowledge) novel theorem we prove about the geometry of polyhedral cones.

**Theorem 8.** *Consider any polyhedral cone $\mathcal{C} \subseteq \mathbb{R}^k$, $k \in \mathbb{N}$, where $\mathcal{C} = \{\boldsymbol{u} : H^T \boldsymbol{u} \geq \boldsymbol{0}\}$ for some matrix $H \in \mathbb{R}^{k \times t}$, $t \in \mathbb{N}$ is a description of $\mathcal{C}$ as an intersection of halfspaces. Then there exists some $\theta > 0$ such that for any $\boldsymbol{u} \in \mathcal{C}$ with $\|\boldsymbol{u}\| \geq 1$, there exists $\boldsymbol{u}' \in \mathcal{C}$ with $\|\boldsymbol{u}'\| = 1$ so that for any column $\boldsymbol{h}$ of $H$ at least one of the following is true:*

$$(i) \ \text{Either } \boldsymbol{h} \cdot \boldsymbol{u} \geq \theta \text{ and } \boldsymbol{h} \cdot \boldsymbol{u}' \geq \theta, \quad \text{or} \quad (ii) \ \boldsymbol{h} \cdot \boldsymbol{u} = \boldsymbol{h} \cdot \boldsymbol{u}'.$$

The idea behind Theorem 8 (see Appendix E.3) is to subtract from $\boldsymbol{u}$ a vector within the nullspace of the matrix $H_{\mathcal{I}}^T$, where $\mathcal{I}$ is the set of indices of columns $\boldsymbol{h}$ of $H$ such that $\boldsymbol{h} \cdot \boldsymbol{u}$ is small. In order to pick the correct vector, we use, additionally, a pivot vector $\boldsymbol{w}$ which satisfies $\boldsymbol{h} \cdot \boldsymbol{w} \geq \theta$ for any interesting column $\boldsymbol{h}$ of $H$. The vector which we subtract from $\boldsymbol{u}$ depends on the projections of $\boldsymbol{u}$ and $\boldsymbol{w}$ on the nullspace of $H_{\mathcal{I}}^T$. We believe that Theorem 8 is of independent interest.

**Structural Distance & TV Distance.** The last step towards Theorem 6 is to show that Lemma 7 can be transformed in terms of TV distance (see Appendix E.1). We stress that structural distance is a local metric, since it is defined in the terms of a property that every point of the domain satisfies independently, while TV distance is a global metric since it expresses the total difference between two distributions over the whole domain. Therefore, the main technical complication here is that the support is infinite and it is not clear whether structural distance implies bounds for the TV distance. The complication is resolved by the following lemma, whose general form (Lemma 35) is in fact also useful in other parts of the proof of Theorem 3 and states that when the parameter vector's norm is large enough, then almost all the mass lies within an bounded length interval around the mode. Its proof is based on the bounded central moments assumption (4).

**Lemma 9** (Informal (See Lemma 35))**.** *Under assumptions* (2) *and* (4)*, there exists some natural number $\ell = O(\sqrt{B})$ such that for any $\boldsymbol{a} \in \varrho\text{-}\mathrm{Cone}\mathcal{A}$ and any mode $M_{\boldsymbol{a}}$ of $\mathcal{E}_{\boldsymbol{T}}(\boldsymbol{a})$ we have $\mathbf{Pr}_{\boldsymbol{a}}[|W - M_{\boldsymbol{a}}| > \ell] \leq \exp(-\|\boldsymbol{a}\|/\varrho) \cdot O(1)$.*

### 3.4 Sparsifying SIIERVs

We now consider distributions of the form $X = \sum_{i \in [n']} X_i$, where $n' \leq n$, $(X_i)_i$ independent and $X_i \sim \mathcal{E}_{\boldsymbol{T}}(\boldsymbol{a}_i)$ for $\boldsymbol{a}_i \in \mathcal{A}$. When $n'$ is small, then the distribution can be approximated term by term, by setting $\epsilon$ of Theorem 5 equal to $\epsilon/n'$, since the total error of approximation is at most equal to the sum of the errors for each term. When $n'$ is large, the distribution of $X$ resembles the distribution of a discretized Gaussian random variable, due to the Berry-Esseen type bound of Lemma 19. Assumption (7) ensures that the variance of $X$ will be large enough, but also that the shift distance (i.e., $d_{TV}(X, X+1)$) will be small enough. In particular, for unimodal distributions, the shift distance of a single term equals the mass assigned to the mode. Using a more general version of Lemma 9 (see Lemma 35), we show that the variance lower bound implies an upper bound for the mass on the mode.

We only need to account for the case of large $n'$ and, in fact, find some critical value $n'_{\text{crit}} \in \mathbb{N}$ with respect to which sparse and dense cases are split. In the dense case, the distribution of $X$ can be represented by its mean and variance alone. The goal is, therefore, to find some SIIERV $Y$ that is also close to a discretized Gaussian and $\mathbf{E}[X] \approx \mathbf{E}[Y]$ and $\mathbf{Var}(X) \approx \mathbf{Var}(Y)$. We prove in Lemma 36 that (under assumptions (2) and (4)) $\mathbf{Var}_{\boldsymbol{a}}(W)$ and $\mathbf{E}_{\boldsymbol{a}}[W]$ are continuous functions of $\boldsymbol{a}$ on $\mathcal{A}$ (and by assumption (6), $\mathcal{A}$ is path connected). Therefore, one might hope that, although $\mathbf{E}[X]$ is a sum of many different quantities of the form $\mathbf{E}_{\boldsymbol{a}_i}[W]$ with $\boldsymbol{a}_i \in \mathcal{A}$ (and similarly $\mathbf{Var}(X)$, due to independence), continuity implies the existence of a single parameter vector $\boldsymbol{b}$ in $\mathcal{A}$ which expresses the total behavior of the quantities $\mathbf{Var}(X)$ and $\mathbf{E}[X]$. It turns out that the correct way to define $\boldsymbol{b}$ is as the parameter vector for which $\mathbf{Var}(X)/\mathbf{E}[X] = \mathbf{Var}_{\boldsymbol{b}}(W)/\mathbf{E}_{\boldsymbol{b}}[W]$. The reason ratios are used is to eliminate the influence of the number of terms $n'$ (which the parameter vector has no influence on). The idea is to use some kind of intermediate value theorem to prove the existence of $\boldsymbol{b}$, motivated by the fact that in the case $\mathbf{E}[X_i] > 0$ for any $i \in [n']$, then $\mathbf{Var}(X)/\mathbf{E}[X] \in [\min_{i \in [n']} \mathbf{Var}(X_i)/\mathbf{E}[X_i], \max_{i \in [n']} \mathbf{Var}(X_i)/\mathbf{E}[X_i]]$. However, there needs to be a careful handling for the cases that $\mathbf{E}[X_i] = 0$ or $\mathbf{E}[X_i] < 0$ for some $i \in [n']$. The random variable $Y$ is selected to be a SIIERV of order $m = \lceil \mathbf{E}[X]/\mathbf{E}_{\boldsymbol{b}}[W] \rceil$ consisting of i.i.d. terms $(Y_i)_{i \in [m]}$ each following the distribution $\mathcal{E}_{\boldsymbol{T}}(\boldsymbol{b})$.

Finally, we observe that $\boldsymbol{b} \in \mathcal{A}$ and Theorem 5 can be used once again with accuracy $\epsilon/m$ to discretize the parameter space and provide a sparse cover for the dense case.

## Acknowledgments and Disclosure of Funding

We thank Dimitris Fotakis for useful discussions on various stages of this work. Alkis Kalavasis was supported by the Hellenic Foundation for Research and Innovation (H.F.R.I.) under the "First Call for H.F.R.I. Research Projects to support Faculty members and Researchers and the procurement of high-cost research equipment grant", project BALSAM, HFRI-FM17-1424. Konstantinos Stavropoulos was supported by the NSF AI Institute for Foundations of Machine Learning (IFML) and by a scholarship from Bodossaki Foundation. Manolis Zampetakis was supported by the Army Research Office (ARO) under contract W911NF-17-1-0304 as part of the collaboration between US DOD, UK MOD and UK Engineering and Physical Research Council (EPSRC) under the Multidisciplinary University Research Initiative (MURI).

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
