# Appendix

## A  Additional Notation, Preliminaries and General Tools

In this section, we provide some notation that will be useful in the proofs. Moreover, we report a collection of existing results that we will apply in our proofs.

### A.1  Additional Notation

**Standard Notation.**  We denote with $\mathbb{N}$ the set of natural numbers $\{1, 2, 3, \dots\}$ and $\mathbb{N}_0 = \{0\} \cup \mathbb{N}$. We denote with $\mathbb{Z}$ the set of integer numbers $\{\dots, -2, -1, 0, 1, 2, \dots\}$ and with $\mathbb{R}$ the set of real numbers. We denote vectors using bold letters e.g., $\boldsymbol{a}, \boldsymbol{T}$. We let $\|\cdot\|_p$ denote the $L_p$ norm with $p \in \{1, 2, \infty\}$. In general, we let $\|\cdot\| = \|\cdot\|_2$ be the standard Euclidean norm. The inner product between two vectors $\boldsymbol{a}, \boldsymbol{b}$ is denoted by $\boldsymbol{a} \cdot \boldsymbol{b}$. We use $\Gamma$ for the Gamma function and $\zeta$ for the Riemann zeta function. For any $r > 0$, $k \in \mathbb{N}$, $\boldsymbol{a} \in \mathbb{R}^k$, we denote with $\mathbb{B}_r(\boldsymbol{a})$ (resp. $\mathbb{B}_r[\boldsymbol{a}]$) the open (resp. closed) ball centered at $\boldsymbol{a}$ with radius $r$.

**Probability.**  For a random variable $X$, we may denote its distribution by $\mathcal{L}(X)$. We denote by $\mathbf{Pr}, \mathbf{E}, \mathbf{Var}, \mathbf{Cov}$ the probability, the expectation, the variance and the covariance operators. We denote $\mathrm{Be}, \mathrm{Bin}, \mathrm{NBin}, \mathrm{Geo}, \mathcal{TG}, \mathcal{Z}, \mathrm{Poi}, \mathrm{Uni}$; the Bernoulli, Binomial, Negative Binomial, Geometric, Truncated Geometric, discretized Gaussian, Poisson and Uniform probability distribution respectively.

**Notions of Distance for Distributions.**  Let $P, Q$ be two probability measures in the discrete probability space $(\Omega, \mathcal{F})$. The *total variation distance* or *statistical distance* between $P$ and $Q$, denoted $d_{TV}(P, Q)$, is defined as $d_{TV}(P, Q) = \frac{1}{2} \sum_{x \in \Omega} |P(x) - Q(x)| = \max_{A \in \mathcal{F}} |P(A) - Q(A)|$. The *Kullback–Leibler divergence* (or simply, KL divergence), denoted $D_{KL}(P \parallel Q)$, is defined as $D_{KL}(P \parallel Q) = \mathbf{E}_{x \sim P}\left[\log\left(\frac{P(x)}{Q(x)}\right)\right] = \sum_{x \in \Omega} P(x) \log\left(\frac{P(x)}{Q(x)}\right)$.

**Exponential Families**  For $k \in \mathbb{N}_0$, $\mathcal{A} \subseteq \mathbb{R}^k$ and $\boldsymbol{T} : \mathbb{Z} \to \mathbb{R}^k$, we denote with $\mathcal{E}_{\boldsymbol{T}}(\mathcal{A})$ the exponential family with sufficient statistics $\boldsymbol{T}$ and parameter space $\mathcal{A}$. In particular, if $W \sim \mathcal{E}_{\boldsymbol{T}}(\boldsymbol{a})$ for some $\boldsymbol{a} \in \mathcal{A}$, then for any $x \in \mathbb{Z}$

$$\mathbf{Pr}[W = x] \propto \exp(-\boldsymbol{a} \cdot \boldsymbol{T}(x)).$$

We will use the notation $\mathbf{Pr}_{\boldsymbol{a}}[W = x]$ (similarly $\mathbf{E}_{\boldsymbol{a}}[W]$ and $\mathbf{Var}_{\boldsymbol{a}}(W)$ for expectation and variance correspondingly) to refer to the probability that $W = x$ given that $W \sim \mathcal{E}_{\boldsymbol{T}}(\boldsymbol{a})$, whenever it is clear by the context that the distribution of $W$ belongs in $\mathcal{E}_{\boldsymbol{T}}(\mathcal{A})$. Note that we will use the following notation $\mathbf{E}_W[|W|^0] = \mathbf{E}_W[\mathbf{1}\{W \neq 0\}] = \mathbf{Pr}_W[W \neq 0]$, i.e., we interpret $0^0$ as 0. Note that in the general case, an exponential family $\mathcal{E}$ supported on $\mathbb{Z}$ is defined in terms of some additional function $h : \mathbb{Z} \to (0, +\infty)$ so that if $W \sim \mathcal{E}$, then $\mathbf{Pr}[W = x] \propto h(x) \cdot \exp(-\boldsymbol{a} \cdot \boldsymbol{T}(x))$. In this case, we use the notation $W \sim \mathcal{E}_{\boldsymbol{T}, h}(\boldsymbol{a})$. However, we can reduce this setting to $h \equiv 1$ by considering $\mathcal{A}' = \mathcal{A} \times \{1\}$ and $\boldsymbol{T}' = (\boldsymbol{T}, -\log_e(h(x)))$. We also define the logarithmic partition function $\Lambda_{\boldsymbol{T}, h} : \mathbb{R}^k \to \mathbb{R}_+$ as $\Lambda_{\boldsymbol{T}, h}(\boldsymbol{a}) = \log\left(\sum_{x=0}^{\infty} h(x) \exp(-\boldsymbol{a} \cdot \boldsymbol{T}(x))\right)$.

**Modes of Distributions.**  For any distribution $\mathcal{D}$ over the lattice of integers $\mathbb{Z}$, we consider the set of modes of $\mathcal{D}$ to be $\mathcal{M} = \arg\max_{x \in \mathbb{Z}} \mathcal{D}(x)$ (where $\mathcal{D}(x) = \mathbf{Pr}_{W \sim \mathcal{D}}[W = x]$). We say that $\mathcal{D}$ is unimodal if there exists some (mode) $M$ in $\mathbb{Z}$ such that if $W \sim \mathcal{D}$, then

$$\mathbf{Pr}[W = x] \geq \mathbf{Pr}[W = x + 1], \text{ for any } x \geq M \text{ and}$$
$$\mathbf{Pr}[W = x] \leq \mathbf{Pr}[W = x + 1], \text{ for any } x < M.$$

Note that it might be the case that $\mathbf{Pr}[W = x] = \mathbf{Pr}[W = x + 1]$, and therefore $\mathcal{D}$ could have many neighboring modes (although we call it unimodal).

For $\boldsymbol{a} \in \mathcal{A}$, consider the distribution $\mathcal{E}_{\boldsymbol{T}}(\boldsymbol{a})$ which lies in the exponential family $\mathcal{E}_{\boldsymbol{T}}(\mathcal{A})$. We denote with $\mathcal{M}_{\mathcal{E}_{\boldsymbol{T}}(\boldsymbol{a})}$ or simply $\mathcal{M}_{\boldsymbol{a}}$ (whenever $\boldsymbol{T}$ is clear by the context) the set of modes of $\mathcal{E}_{\boldsymbol{T}}(\boldsymbol{a})$. We also use the notation $M_{\boldsymbol{a}}$ to refer to any specific mode of $\mathcal{E}_{\boldsymbol{T}}(\boldsymbol{a})$, like in $\mathbf{Pr}_{\boldsymbol{a}}[W = M_{\boldsymbol{a}}]$ (the referenced

point could equivalently be any mode of $\mathcal{E}_{\boldsymbol{T}}(\boldsymbol{a})$ since all modes are assigned the same probability mass).

We say that the exponential family $\mathcal{E}_{\boldsymbol{T}}(\mathcal{A})$ is unimodal if $\mathcal{E}_{\boldsymbol{T}}(\boldsymbol{a})$ is unimodal for each $\boldsymbol{a} \in \mathcal{A}$. We denote with $\mathcal{M}_{\mathcal{A}}$ the union of the sets of modes of the distributions in $\mathcal{E}_{\boldsymbol{T}}(\mathcal{A})$, i.e., $\mathcal{M}_{\mathcal{A}} = \cup_{\boldsymbol{a} \in \mathcal{A}} \mathcal{M}_{\boldsymbol{a}}$.

**Sets and Set Operators.** A polyhedron is the intersection of finitely many affine halfspaces. A cone is a subset $K \subseteq \mathbb{R}^k$ with $\boldsymbol{0} \in K$ and $\alpha \boldsymbol{y} \in K$ for all $\boldsymbol{y} \in K$ and $\alpha \in \mathbb{R}_+$. A polyhedral cone is a polyhedron that is a cone. We say that $\mathtt{Op}$ is an extensive set operator on $\mathbb{R}^k$ if for any set $\mathcal{A} \subseteq \mathbb{R}^k$, $\mathtt{Op}\mathcal{A}$ is a subset of $\mathbb{R}^k$ and $\mathcal{A} \subseteq \mathtt{Op}\mathcal{A}$. We will make use of some particular extensive set operators: The closure operator $\mathtt{closure}$: $\mathtt{closure}\mathcal{A} = \mathcal{A} \cup \partial \mathcal{A}$. We also use the notation $\overline{\mathcal{A}} := \mathtt{closure}\mathcal{A}$. The convex hull operator $\mathtt{Conv}$:

$$\mathtt{Conv}\mathcal{A} = \left\{ \sum_{i \in [n]} t_i \boldsymbol{a}_i : \boldsymbol{a}_i \in \mathcal{A}, t_i \in [0,1], \sum_{i \in [n]} t_i = 1, n \in \mathbb{N} \right\}, \tag{1}$$

for any $\mathcal{A} \subseteq \mathbb{R}^k$. The conical hull operator $\mathtt{Cone}$:

$$\mathtt{Cone}\mathcal{A} = \left\{ \sum_{i \in [n]} t_i \boldsymbol{a}_i : \boldsymbol{a}_i \in \mathcal{A}, t_i \geq 0, n \in \mathbb{N} \right\}, \tag{2}$$

for any $\mathcal{A} \subseteq \mathbb{R}^k$ The $\varrho$-conical hull operator $\varrho\text{-}\mathtt{Cone}$, where $\varrho > 0$:

$$\varrho\text{-}\mathtt{Cone}\mathcal{A} = \mathcal{A} \cup (\mathtt{Cone}\mathcal{A} \setminus \mathbb{B}_\varrho(\boldsymbol{0})).$$

In other words, $\varrho\text{-}\mathtt{Cone}$ operator inserts in $\mathcal{A}$ all points of the conical hull of $\mathcal{A}$ with norm at least $\varrho$. The shade operator:

$$\mathtt{Shade}\mathcal{A} = \{t\boldsymbol{a} : \boldsymbol{a} \in \mathcal{A}, t \geq 1\}.$$

We will use these set operators to rule out the possibility that $\mathcal{A}$ is a very contrived set that makes the proper covering problem unreasonably difficult or complicated, since we focus on providing the first general approach for the proper covering problem. In particular, we will relax our demand that the cover is proper, by enabling sums of random variables that belong in a slightly wider exponential family by enlarging $\mathcal{A}$ appropriately.

### A.1.1  What do Proper Covering and Learning mean for SIIERVs?

**(Weakly) Proper Covers.** Below, we give a slightly relaxed definition for the notion of proper covers for our setting. We will use these set operators to rule out the possibility that $\mathcal{A}$ is a very contrived set that makes the proper covering problem unreasonably difficult or complicated, since we focus on providing the first general approach for the proper covering problem. In particular, we will relax our demand that the cover is proper, by enabling sums of random variables that belong in a slightly wider exponential family by enlarging $\mathcal{A}$ appropriately.

**Definition 2** (Proper Covers of SIIERVs)**.** *Let $\mathcal{C}$ be a cover of $\mathcal{E}_{\boldsymbol{T}}(\mathcal{A})$-SIIRVs of order $n$, where $\mathcal{A} \subseteq \mathbb{R}^k$. Consider $R \geq 1$ and $\mathtt{Op}$ is an extensive set operator on $\mathbb{R}^k$. If any element of $\mathcal{C}$ is the distribution of an $\mathcal{E}_{\boldsymbol{T}}(\mathtt{Op}\mathcal{A})$-SIIRV of order $R \cdot n$, then we say that $\mathcal{C}$ is $(R, \mathtt{Op})$-proper.*

**(Weakly) Proper Learning.** For any $n, k \in \mathbb{N}$, any $\mathcal{A} \subseteq \mathbb{R}^k$ and any $\boldsymbol{T} : \mathbb{Z} \to \mathbb{R}^k$ (so that $\mathcal{E}_{\boldsymbol{T}}(\mathcal{A})$ is well defined), we say that the the class of $\mathcal{E}_{\boldsymbol{T}}(\mathcal{A})$-SIIRVs of order $n$ can be $(R, \mathtt{Op})$-properly learned if there exist some $R \geq 1$ and some extensive set operator $\mathtt{Op}$ such that for any $\epsilon, \delta \in (0,1)$ there exists some algorithm $A$ and some polynomial $m$ on $1/\epsilon, 1/\delta$ (and other relevant parameters) such that $A$, given $m$ independent samples from some unknown $\mathcal{E}_{\boldsymbol{T}}(\mathcal{A})$-SIIRV $X$ of order $n$, outputs some $\mathcal{E}_{\boldsymbol{T}}(\mathtt{Op}\mathcal{A})$-SIIRV $Y$ of order $R \cdot n$ such that $d_{TV}(X, Y) \leq \epsilon$, with probability at least $1 - \delta$.

### A.2  General Tools

We show that our novel notion of distance, the structural distance is actually a distance metric.

**Lemma 10.** *The structural distance (Definition 1) is a distance metric between discrete distributions.*

*Proof.* Note that $d_{\mathrm{ST}}$ is non-negative. It is sufficient to prove the three following properties:

**Identity of indiscernibles.** $d_{\mathrm{ST}}(\mathcal{D}_1, \mathcal{D}_2) = 0$ if and only if $\mathcal{D}_1$ and $\mathcal{D}_2$ are equivalent. The first direction of the property is satisfied, since when $\epsilon = 0$, any point outside the support of $\mathcal{D}_1$ must be assigned zero mass by $\mathcal{D}_2$ (hence they have common support) and any point in the support must have proportionally equivalent mass (which implies exactly equal mass, since the probabilities must sum to 1). For the other direction, observe that if $\mathcal{D}_1$ and $\mathcal{D}_2$ are equivalent, then the structural distance is zero.

**Symmetry.** Observe that Definition 1 is, in fact, symmetric and so $d_{\mathrm{ST}}(\mathcal{D}_1, \mathcal{D}_2) = d_{\mathrm{ST}}(\mathcal{D}_2, \mathcal{D}_1)$.

**Triangle Inequality.** We have to show that $d_{\mathrm{ST}}(\mathcal{D}_1, \mathcal{D}_3) \leq d_{\mathrm{ST}}(\mathcal{D}_1, \mathcal{D}_2) + d_{\mathrm{ST}}(\mathcal{D}_2, \mathcal{D}_3)$. Let $d_{\mathrm{ST}}(\mathcal{D}_1, \mathcal{D}_2) = \epsilon_1$ and $d_{\mathrm{ST}}(\mathcal{D}_2, \mathcal{D}_3) = \epsilon_2$.

Consider, first, any $x \in \mathcal{X}$ such that $\mathcal{D}_1(x) > (\epsilon_1 + \epsilon_2) \cdot p_1$. Then $\mathcal{D}_1(x) > \epsilon_1 \cdot p_1$ and hence, for the point $x$ and the pair $(\mathcal{D}_1, \mathcal{D}_2)$, it must hold $\mathcal{D}_2(x)/p_2 = \mathcal{D}_1(x)/p_1 > \epsilon_1 + \epsilon_2$. Similarly, we have $\mathcal{D}_2(x)/p_2 > \epsilon_1 + \epsilon_2$ and so $\mathcal{D}_3(x)/p_3 = \mathcal{D}_2(x)/p_2 > \epsilon_1 + \epsilon_2$. This implies that $\mathcal{D}_3(x)/p_3 = \mathcal{D}_1(x)/p_1$.

For the rest points $x \in \mathcal{X}$ with $\mathcal{D}_1(x) \leq (\epsilon_1 + \epsilon_2) \cdot p_1$, we have to take cases for $\mathcal{D}_1(x)$ (either $\leq \epsilon_1 p_1$ or $> \epsilon_1 p_1$) and after applying what we know about $d_{\mathrm{ST}}(\mathcal{D}_1, \mathcal{D}_2)$, take cases for $\mathcal{D}_2(x)$ (and apply knowledge about $d_{\mathrm{ST}}(\mathcal{D}_2, \mathcal{D}_3)$). In any case, we get that $\mathcal{D}_3(x) \leq (\epsilon_1 + \epsilon_2) \cdot p_3$. Therefore $d_{\mathrm{ST}}(\mathcal{D}_1, \mathcal{D}_3) \leq \epsilon_1 + \epsilon_2$. $\qquad\square$

We continue with a collection of general tools that we are going to need for our proofs. The Minkowski-Weyl Theorem shows that any polyhedron can be either represented in a constrained or a finitely generated form.

**Proposition 11** (Minkowski-Weyl Theorem [Minkowski [1896],Weyl [1935]]). *Let $\mathcal{C} \subseteq \mathbb{R}^k$ for $k \in \mathbb{Z}$. Then the following are equivalent:*

1. *There exists $t \in \mathbb{N}$ and some matrix $H \in \mathbb{R}^{k \times t}$ such that $\mathcal{C} = \{\boldsymbol{u} \in \mathbb{R}^k : H^T \boldsymbol{u} \geq \boldsymbol{0}\}$.*

2. *There exists $s \in \mathbb{N}$ and some matrix $Z \in \mathbb{R}^{k \times s}$ such that $\mathcal{C} = \{\boldsymbol{u} \in \mathbb{R}^k : \boldsymbol{u} = Z\boldsymbol{x}, \boldsymbol{x} \geq \boldsymbol{0}\}$.*

The next standard inequality shows that in order to control the statistical distance for two distributions, it suffices to control the KL divergence.

**Proposition 12** (Pinsker's Inequality). *For any probability distributions $P$ and $Q$ on a common measurable space, the following inequality holds.*

$$d_{TV}(P, Q) \leq \sqrt{\frac{1}{2} D_{KL}(P \parallel Q)}.$$

The following lemma is a useful tool in various parts of our proofs. It essentially controls ratios of sums of positive quantities.

**Lemma 13** (Ratio of Sums Inequality). *Let $a_1, \ldots, a_n$ and $b_1, \ldots, b_n$ be* positive *numbers. Then, it holds that*

$$\min_{i \in [n]} \frac{a_i}{b_i} \leq \frac{\sum_{i \in [n]} a_i}{\sum_{i \in [n]} b_i} \leq \max_{i \in [n]} \frac{a_i}{b_i}.$$

*Proof.* We have that

$$\sum_{i \in [n]} a_i = \sum_{i \in [n]} b_i \left(\frac{a_i}{b_i}\right) \leq \sum_{i \in [n]} b_i \max_{j \in [n]} \frac{a_j}{b_j}.$$

Hence, we get that

$$\frac{\sum_{i \in [n]} a_i}{\sum_{i \in [n]} b_i} \leq \max_{j \in [n]} \frac{a_j}{b_j}.$$

The other direction follows similarly. $\qquad\square$

The next tool characterizes the expectation and the variance of the sufficient statistics vector in terms of the log-partition function. Moreover, it provides a way to control the KL divergence of two exponential family distributions.

**Lemma 14** (Moments and KL Divergence for Exponential Families). *Let $\mathcal{E}_{\boldsymbol{T},h}$ be an exponential family with parameters $\boldsymbol{a} \in \mathbb{R}^k$, log-partition function $\Lambda(\cdot) = \Lambda_{\boldsymbol{T},h}(\cdot)$ and range of natural parameters $\mathcal{A} = \mathcal{A}_{\boldsymbol{T},h}$. The following hold for the distribution $\mathcal{E}(\boldsymbol{a}) \in \mathcal{E}_{\boldsymbol{T},h}$.*

   ($i$) *For all $\boldsymbol{a} \in \mathcal{A}$, it holds that*
$$\mathop{\mathbf{E}}_{\boldsymbol{x} \sim \mathcal{E}(a)}[\boldsymbol{T}(\boldsymbol{x})] = \nabla\Lambda(\boldsymbol{a}) \,.$$

   ($ii$) *For all $\boldsymbol{a} \in \mathcal{A}$, it holds that*
$$\mathop{\mathbf{Cov}}_{\boldsymbol{x} \sim \mathcal{E}(a)}[\boldsymbol{T}(\boldsymbol{x})] = \nabla^2\Lambda(\boldsymbol{a}) \,.$$

   ($iii$) *For all $\boldsymbol{a}, \boldsymbol{a}' \in \mathcal{A}$ and for some $\boldsymbol{\xi} \in L(\boldsymbol{a}, \boldsymbol{a}')$[4], it holds that*
$$D_{KL}(\mathcal{E}(\boldsymbol{a}) \parallel \mathcal{E}(\boldsymbol{a}')) = (\boldsymbol{a}' - \boldsymbol{a})^T \nabla^2\Lambda(\boldsymbol{\xi})(\boldsymbol{a}' - \boldsymbol{a}) \,. \tag{3}$$

The upcoming lemma control the statistical distance for sums of independent random variables.

**Lemma 15** (TV-Subadditivity for Sums of Random Variables). *Let $(X_i)_{i \in [n]}$ be a sequence of $n$ independent random variables and $(Y_i)_{i \in [n]}$ be also a sequence of $n$ independent random variables. Then, we have that*
$$d_{TV}\left(\sum_{i \in [n]} X_i, \sum_{i \in [n]} Y_i\right) \leq \sum_{i \in [n]} d_{TV}(X_i, Y_i) \,.$$

This lemma controls the statistical distance between two Poisson distributions.

**Lemma 16** (Statistical Distance of Poisson RVs). *Let $\lambda_1, \lambda_2 > 0$. Then, it holds that:*
$$d_{TV}(\mathrm{Poi}(\lambda_1), \mathrm{Poi}(\lambda_2)) \leq \frac{e^{|\lambda_1 - \lambda_2|} - e^{-|\lambda_1 - \lambda_2|}}{2} \,.$$

This lemma controls the statistical distance between two discretized Gaussian random variables.

**Lemma 17** (Statistical Distance of Discretized Gaussian RVs). *Let $\mu_1, \mu_2 \in \mathbb{R}$ and $0 < \sigma_1 \leq \sigma_2$. Then, it holds that:*
$$d_{TV}(\mathcal{Z}(\mu_1, \sigma_1^2), \mathcal{Z}(\mu_2, \sigma_2^2)) \leq \frac{1}{2}\left(\frac{|\mu_1 - \mu_2|}{\sigma_1} + \frac{\sigma_2^2 - \sigma_1^2}{\sigma_1^2}\right) \,.$$

The next lemma provides a bound on the shift distance for SIIRVs.

**Lemma 18** (Statistical Distance of Shifted SIIRV). *Let $X = \sum_i X_i$ be the sum of independent integer-valued random variables. Then, it holds that*
$$d_{TV}(X, X+1) \leq \frac{\sqrt{2/\pi}}{\sqrt{\frac{1}{4} + \sum_i (1 - d_{TV}(X_i, X_i + 1))}} \,.$$

The next lemma is a standard tool that essentially says that an SIIRV that has bounded third moment and TV shift invariance is close to a discretized Gaussian in statistical distance. The TV shift invariance bound is crucial; note that if we drop this property, standard CLT theorems imply bounds for the weaker Kolmogorov distance.

**Lemma 19** (Discretized Gaussian Approximation (Chen et al. [2010])). *Let $X_1, ..., X_n$ be a finite sequence of independent integer-valued random variables and let $X = \sum_{i \in [n]} X_i$. If $\mu_i = \mathbf{E}[X_i], \sigma_i^2 = \mathbf{Var}(X_i), \beta_i = \mathbf{E}[|X_i - \mu_i|^3], \mu = \sum_{i \in [n]} \mu_i, \sigma^2 = \sum_{i \in [n]} \sigma_i^2, \beta = \sum_{i \in [n]} \beta_i$ and*
$$\sup_{i \in [n]} d_{TV}(X - X_i, X - X_i + 1) \leq \delta \,,$$
*then, if $Z$ is distributed according to the discretized Gaussian distribution $\mathcal{Z}(\mu, \sigma^2)$, we have that*
$$d_{TV}(X, Z) \leq O(1/\sigma) + O(\delta) + O(\beta/\sigma^3) + O(\delta\beta/\sigma^2) \,.$$
*In particular, we have that*
$$d_{TV}(X, Z) \leq \delta\left(1 + \frac{3}{2}\frac{\beta}{\sigma^2}\right) + \frac{1}{\sigma}\left(\frac{1}{2\sqrt{2\pi}} + (5 + 3\sqrt{\pi/8})\frac{\beta}{\sigma^2}\right) \,.$$

---

[4]We denote $L(\boldsymbol{x}, \boldsymbol{y}) = \{\boldsymbol{z} \in \mathbb{R}^k : \boldsymbol{z} = \lambda\boldsymbol{x} + (1 - \lambda)\boldsymbol{y}, \lambda \in [0, 1]\}$.

# B    Learning SIIURVs and SIIERVs

We now formally state our main learning results. To do this, we begin by formally setting up our learning framework for SIIRVs.

**Learning Framework.**    In the problem of learning an SIIRV $X = \sum_{i \in [n]} X_i$, the learner is given the value $n$ (of the number of summands) and has sample access to independent draws from an unknown target $X$. The goal of the learning algorithm is to output a hypothesis distribution $\widetilde{X}$ that is $\epsilon$-close to $X$ in total variation distance, i.e., $d_{TV}(X, \widetilde{X}) \leq \epsilon$, with probability at least $1 - \delta$. The accuracy $\epsilon$ and the confidence $\delta$ are both provided to the learner as input.

In Appendix B.1, the target $X$ will be an SIIURV of order $n$ for some given $n \in \mathbb{N}$ (i.e., a sum with $n' \leq n$ random terms) that belongs to the family of distributions that contain all the sums of at most $n$ random variables that satisfy Assumption 1. We are going to provide a learning algorithm for this class of distributions.

In Appendix B.2, the target $X$ will be an $\mathcal{E}_T(\mathcal{A})$-SIIRV of order $n$, i.e., a sum of at most $n$ random variables each one belonging in the exponential family $\mathcal{E}_T(\mathcal{A})$ satisfying Assumption 2. We will give a (weakly) proper learning algorithm in the sense that the output will be an $\mathcal{E}_T(\mathcal{A}')$-SIIERV of order $m$ where $m$ and $\mathcal{A}'$ will be slightly different that $n$ and $\mathcal{A}$ respectively.

**Common Technical Tool.**    Our learning algorithms (Figure 1 and Figure 2) use hypothesis testing as a distinct tool for the learning procedures. Hypothesis testing will appear in various points of the algorithms; in the sparse learning phase (Claim 23 and Claim 27), in the proper dense one (Claim 28) and in the hypothesis selection of the second stage (Proposition 20). Intuitively, for some desired accuracy $\epsilon > 0$, given a collection of $M$ candidate hypothesis distributions, one of which is $\epsilon$-close in total variation distance to the target distribution of $X$, a hypothesis testing algorithm draws $\widetilde{O}(\log(M)/\epsilon^2)$ samples from $X$, runs in time polynomial in $M$ and $1/\epsilon$ and outputs a hypothesis (among the $M$ candidates) that is $O(\epsilon)$-close to the true $X$, with high probability. Such testing procedures have been studied by a range of authors (see e.g., Yatracos [1985], Daskalakis and Kamath [2014], Acharya et al. [2014], Daskalakis et al. [2013, 2015a], De et al. [2014, 2018]). For concreteness we are going to recall some standard results later.

## B.1    Learning SIIURVs

Assume that the target $X$ is an SIIURV satisfying Assumption 1. We first provide a discussion on some previous results and the oracles we require.

### B.1.1    Hypothesis Testing and Oracle Access

The following proposition about hypothesis selection can be found in Daskalakis et al. [2015a].

**Proposition 20** (Hypothesis selection (Lemma 8 at Daskalakis et al. [2015a]))**.** *There exists an algorithm* SELECTHYPOTHESIS$^X(H_1, H_2, \epsilon, \delta)$*, which is given sample access to a distribution $X$, two hypothesis distributions $H_1, H_2$ for $X$, an accuracy parameter $\epsilon$ and a confidence parameter $\delta > 0$, draws*

$$O(\log(1/\delta)/\epsilon^2)$$

*samples from $X$ and, in time polynomial in the number of samples, returns some $H \in \{H_1, H_2\}$ with the following guarantee: If $d_{TV}(H_i, X) \leq \epsilon$ for some $i \in \{1, 2\}$, then the distribution $H$ that* SELECTHYPOTHESIS$^X$ *returns satisfies $d_{TV}(H, X) \leq 6\epsilon$.*

The routine SELECTHYPOTHESIS$^X(H_1, H_2, \epsilon, \delta)$ runs a competition between $H_1$ and $H_2$ as follows (for more details and a proof, we refer to Daskalakis et al. [2015a]): It first computes the set $\mathcal{W}_1 = \{x | H_1(x) > H_2(x)\}$ and the probabilities $p_i = H_i(\mathcal{W}_1), i \in \{1, 2\}$. The routine draws $O(\log(1/\delta)/\epsilon^2)$ independent samples from $X$ and calculates the fraction $\tau$ of these samples that fall inside $\mathcal{W}_1$. If $\tau > p_1 - \epsilon$, it selects $H_1$ as the winner and if $\tau < p_2 + \epsilon$, it chooses $H_2$; otherwise, it declares a draw and returns either $H_i$.

**Remark 1.** *We underline that the result in Daskalakis et al. [2015a] does not require sample access to $H_1, H_2$. These distributions are given as input (and their description is short enough to be read*

*by the learner) and the set $\mathcal{W}_1$ can then be computed efficiently. The work of De et al. [2014] deals with scenarios where this is not the case (e.g., the domain is exponentially large in $n$) and in this case sample and evaluation oracle access to $H_1, H_2$ is needed too. We discuss this point later.*

The SELECTHYPOTHESIS$^X$ routine is the main tool of the following hypothesis testing mechanism (Proposition 21), for which we refer the reader to [De et al., 2014, 2018] and, more generally, to e.g., Daskalakis et al. [2015a, 2013]:

**Proposition 21** (Tournament Selection De et al. [2018]). *Let $D$ be a distribution over $W \subseteq \mathbb{Z}$ and let $\mathcal{H}_\epsilon = \{H_j\}_{j \in [M]}$ be a collection of $M$ hypothesis distributions over $W$ with the property that there exists $i \in [M]$ such that $d_{TV}(D, H_i) \leq \epsilon$. There exists an algorithm SELECTTOURNAMENT$^D$ which is given $\epsilon$, a confidence parameter $\delta$ and is provided with access to $(i)$ a source of i.i.d. draws from $D$ and from $H_i$ for all $i \in [M]$; and $(ii)$ an 'evaluation oracle' EVAL$_{H_i}$ for each $i \in [M]$, which, on input $w \in W$, deterministically outputs the value $H_i(w)$ and that has the following behavior: It makes $m = O((1/\epsilon^2) \cdot (\log(M) + \log(1/\delta)))$ draws from $D$ and from each $H_i, i \in [M]$ and $O(m)$ calls to each oracle EVAL$_{H_i}, i \in [M]$. It runs in time $\mathrm{poly}(m, M)^5$ and, with probability $1 - \delta$, it outputs an index $i^\star \in [M]$ that satisfies $d_{TV}(D, H_{i^\star}) \leq 6\epsilon$.*

The routine SELECTTOURNAMENT performs a tournament by running the procedure of Proposition 20 SELECTHYPOTHESIS$^X(H_i, H_j, \epsilon, O(\delta/M))$ for every pair $(H_i, H_j), i < j$ of distributions in the collection of Proposition 21 of size $M$ and either outputs the distribution that was never a loser or it fails. The bound on the running time is a result of the corresponding time bound of the SELECTHYPOTHESIS$^X$ routine. The sample complexity is a result of the union bound over the competitions.

**Sampling & Evaluation Oracles.** For this section, we only need sample access to the target $X$ in order to run the version of Proposition 20. During the learning phase of the sparse instances, we will construct the sparse cover and we will perform the tournament procedure for the distributions in the cover. Crucially, the sparse forms have bounded support and its size does not depend on $n$. Hence, for each sparse form, we have access to an efficient evaluation oracle for the purposes of Proposition 21. For any two distributions in the cover $H_i, H_j$ with domain $W \subseteq \mathbb{Z}$, the algorithm can efficiently compute the set $W_{ij} = \{w \in W | H_i(w) \geq H_j(w)\}$ without additional assumptions. In the dense case, the algorithm will estimate the best fitting discretized Gaussian distribution and we do not need to contruct any cover or run any tournament procedure.

### B.1.2 The Result and the Algorithm

Our main learning result for SIIURVs follows.

**Theorem 22** (Learning SIIURVs). *Under Assumption 1, for any $n \in \mathbb{N}$, accuracy $\epsilon > 0$ and confidence $\delta > 0$, there exists an algorithm LEARNERSIIURV$^X$ (see Figure 1) with the following properties: Given $n, \epsilon, \delta$ and sample access to independent draws from an unknown SIIURV $X$ of order $n$, the algorithm uses*

$$m = O\left(\frac{1}{\epsilon^2} \log(1/\delta)\right) + O\left(\mathrm{poly}(B, 1/\gamma, 1/\epsilon)\right)$$

*samples from $X$ and, in time*

$$\mathrm{poly}\left(m, L^{\mathrm{poly}(B, 1/\gamma, 1/\epsilon)}\right),$$

*outputs a (succint description of a) distribution $\widetilde{X}$ which satisfies $d_{TV}(X, \widetilde{X}) \leq \epsilon$, with probability at least $1 - \delta$.*

Our algorithm works as follows.

We continue with a short discussion on how the algorithm works: The learning algorithm (Figure 1) of Theorem 22 is separated in two distinct stages. In the first stage, it runs two different learning procedures, corresponding to the sparse and dense case of our main structural covering result for SIIURVs. At the end of this stage, two hypotheses are obtained and, hence, the second phase of the learning algorithm performs hypothesis testing in order to select the correct one. In the first

---

[5]We count each call to an evaluation oracle EVAL$_{H_i}$ and draw from a $H_i$ distribution as unit time.

---

*Algorithm for SIIURVs*: $(\epsilon, \delta)$-Learning SIIURVS $X = \sum_{i \in [n]} X_i$.

  1. Run LEARNSPARSE$^X(n, \epsilon, \delta/3)$ of Claim 23 and get the distribution $H_S$.

  2. Run LEARNDENSE$^X(n, \epsilon, \delta/3)$ of Claim 24 and get the distribution $H_D$.

  3. Return the distribution that is the output of SELECTHYPOTHESIS$^X(H_S, H_D, \epsilon, \delta/3)$ of Proposition 20.

---

Figure 1: Learning algorithm for SIIURVs.

phase, the processes LEARNSPARSE$^X$ and LEARNDENSE$^X$ are performed. The LEARNSPARSE$^X$ procedure (see Claim 23) performs a tournament over the distributions of the cover of the sparse regime $\mathcal{D}_U^{(s)}(\epsilon)$ with error $\epsilon$ and outputs the hypothesis/distribution $H_S$ that is closer to $X$. On the other hand, the the LEARNDENSE$^X$ process (see Claim 24) estimates the parameters of a discretized Gaussian, which approximates the input sum $X$ and outputs this distribution. We denote by $H_D$ the output hypothesis of the dense procedure. Now, in the second phase, the learning algorithm runs the black-box procedure SELECTHYPOTHESIS$^X$ (see Proposition 20) that chooses the winner between the two hypotheses $H_S$ and $H_D$, i.e., the one that is closer to $X$ with high probability.

**The Proof of Theorem 22.**

*Proof.* The algorithm (Figure 1) runs the routine LEARNSPARSE$^X$ of Claim 23 with input $(n, \epsilon, \delta/3)$ and gets the distribution $H_S$. Then, it runs LEARNDENSE$^X$ of Claim 24 with input $(n, \epsilon, \delta/3)$ and aims to learn the best fitting discretized Gaussian to the input sum $X$ and output the hypothesis distribution $H_D$. In order to conclude the SIIURV learning part, via Proposition 20, one can examine which hypothesis between $H_S$ and $H_D$ is closer to the target $X$, by running SELECTHYPOTHESIS$^X(H_S, H_D)$ with parameter $\epsilon, \delta/3$. In conclusion, with probability at least $1 - \delta$, the algorithm will satisfy the desiderata of Theorem 22. We divide the proof in a series of claims.

In Claim 23, we analyze an algorithm which learns sparse instances and outputs a hypothesis distribution $H_S$.

**Claim 23** (Learning Sparse Instances). *Under Assumption 1, for any $n, \epsilon, \delta > 0$, there is an algorithm* LEARNSPARSE$^X(n, \epsilon, \delta)$ *that given*

$$m = O\left(\text{poly}(B, 1/\gamma, 1/\epsilon) \cdot \log(L) + \frac{1}{\epsilon^2} \log(1/\delta)\right)$$

*samples from the target SIIURV $X$ over $\mathbb{Z}$, outputs a (succint description of a) hypothesis distribution $H_S$ with the following guarantee: If $X$ is $\epsilon$-close to a sparse form (see Theorem 2), then it holds that $d_{TV}(H_S, X) \leq c_1 \epsilon$, for some universal constant $c_1 \geq 1$, with probability at least $1 - \delta$. Furthermore, the running time of the algorithm is* $\text{poly}\left(m, L^{\text{poly}(B, 1/\gamma, 1/\epsilon)}\right)$.

*Proof.* Let $\epsilon > 0$ and assume that Assumption 1 holds. Since we are in the sparse form case, the algorithm can construct a cover of small size as described in the proof of Theorem 2 (see the final part of Appendix C), i.e., by quantizing the probability mass on each point of each of a set of intervals that provably include some interval that contains the support of $X$. The structural result of Theorem 2 implies that there exists a cover $\mathcal{D}_U^{(s)} = \mathcal{D}_U^{(s)}(\epsilon)$ of radius $\epsilon$ (i.e., a collection of probability distributions that contains an $\epsilon$-close – in total variation distance – representative for each distribution in $\mathcal{D}$) whose size is equal to

$$\left|\mathcal{D}_U^{(s)}\right| \leq L^{\text{poly}(B, 1/\gamma, 1/\epsilon)}.$$

By the hypothesis and from the structure of the cover, $X$ is $\epsilon$-close to an element of the set $\mathcal{D}_U^{(s)}$. Note that Proposition 21 is applicable since the learner can read the distributions of the cover as input and so no specific oracle access is required. We can apply the SELECTTOURNAMENT$^X$ algorithm

with input the distributions' collection which lie in $\mathcal{D}_{\mathrm{U}}^{(s)}$ with accuracy $\epsilon$ and confidence $\delta$. This concludes the proof. The sample complexity of the algorithm is

$$m = O\left(\frac{1}{\epsilon^2}\left(\log\left|\mathcal{D}_{\mathrm{U}}^{(s)}\right| + \log(1/\delta)\right)\right),$$

and the running time is poly $\left(m, \left|\mathcal{D}_{\mathrm{U}}^{(s)}\right|\right)$. □

As a next step, we analyze the learning phase concerning the dense instances: In Claim 24, we deal with the SIIURV learning of the dense case, using the Gaussian approximation.

**Claim 24** (Learning Dense Instances). *Under Assumption 1, for any $n, \epsilon, \delta > 0$, there is an algorithm* LEARNDENSE$^X(n, \epsilon, \delta)$ *that given*

$$O(\log(1/\delta)/\epsilon^2)$$

*samples from the target SIIURV $X$ over $\mathbb{Z}$, runs in time $O(\log(1/\delta)/\epsilon^2)$ and outputs a (succint description of a) hypothesis distribution $H_D$ with the following guarantee: If $X$ is $\epsilon$-close to a dense form (see Theorem 2), then it holds that $d_{TV}(H_D, X) \leq O(\epsilon)$, with probability at least $1 - \delta$, and $H_D$ is a discretized Gaussian distribution.*

*Proof.* Let $\epsilon > 0$ and assume that $X$ is $\epsilon$-close to a dense form SIIURV. Let $X = \sum_{i\in[n]} X_i$ be an SIIURV and set $\mu = \mathbf{E}[X]$ and $\sigma^2 = \mathbf{Var}(X)$. There exists an algorithm that uses $O(\log(1/\delta)/\epsilon^2)$ samples from $X$ and runs in time $O(\log(1/\delta)/\epsilon^2)$ and, with probability at least $1 - \delta$, computes estimates $\widehat{\mu}$ and $\widehat{\sigma}^2$ so that

$$|\mu - \widehat{\mu}| \leq \epsilon\sigma \text{ and } |\sigma^2 - \widehat{\sigma}^2| \leq \epsilon\sigma^2 \cdot O(1).$$

Our proof follows the steps presented in Daskalakis et al. [2015a]: We will provide a routine achieving the desired estimation with probability at least $2/3$. Afterwards, there exists a standard procedure[6] that boosts the success probability to $1-\delta$ at the expense of a multiplicative $O(\log(1/\delta))$ overhead in the number of samples.

Mean Estimation. In order to weakly estimate the mean $\mu$, let $\{Z_i\}_{i\in[N]}$ be i.i.d. samples from $X$ and let $\widehat{\mu} = \frac{1}{N}\sum_{i\in[N]} Z_i$. Chebyshev's inequality implies that

$$\mathbf{Pr}\left[|\widehat{\mu} - \mu| \geq t\sqrt{\mathbf{Var}(X)}\right] = \mathbf{Pr}\left[|\widehat{\mu} - \mu| \geq t\sigma/\sqrt{N}\right] \leq 1/t^2.$$

Choosing $t = \sqrt{3}$ and $N = O(3/\epsilon^2)$, we get that $|\widehat{\mu} - \mu| \leq \epsilon\sigma$ with probability at least $2/3$.

Variance Estimation. Similarly, one can compute a weakly estimate for the variance $\sigma^2$. Let $\{Z_i\}_{i\in[N]}$ be i.i.d. samples from $X$ and, using the Bessel's correction, let $\widehat{\sigma}^2 = \frac{1}{N-1}\sum_{i\in[N]}(Z_i - \frac{1}{N}\sum_{i\in[N]} Z_i)^2$. We have that

$$\mathbf{E}\left[\widehat{\sigma}^2\right] = \sigma^2 \text{ and } \mathbf{Var}\left(\widehat{\sigma}^2\right) = \sigma^4\left(\frac{2}{N-1} + \frac{\kappa}{N}\right),$$

where $\kappa := \frac{\mathbf{E}[(X-\mu)^4]}{\sigma^4} - 3$ is the excess kurtosis of the distribution of $X$. For the random variable $X = \sum_{i\in[n]} X_i$ with $X_i \sim \mathcal{E}_T(\boldsymbol{a}_i)$, we have that

$$\kappa = \frac{1}{\sigma^4}\mathbf{E}[(X-\mu)^4] - 3 = \frac{\sum_{i\in[n]}(\mathbf{Var}(X_i))^2 \cdot \left(\frac{\mathbf{E}[(X_i-\mathbf{E}[X_i])^4]}{(\mathbf{Var}(X_i))^2} - 3\right)}{\left(\sum_{i\in[n]}\mathbf{Var}(X_i)\right)^2} \leq \frac{1}{n}\cdot B/\gamma^2,$$

by using independence and conditions (1) and (3). The second equality follows from the next computations: Define $\kappa_4(X) := \mathbf{E}[(X - \mathbf{E}[X])^4] - 3\,\mathbf{Var}(X)^2$ be the fourth cumulant of $X$. Since

---

[6]The boosting argument requires running the weak estimators $O(\log(1/\delta))$ times in order to obtain two sequences of estimates $(\widehat{\mu}_i)_{i\in[O(\log(1/\delta))]}$ and $(\widehat{\sigma}_i^2)_{i\in[O(\log(1/\delta))]}$. Finally, the boosting process will output the median of these sequences of $O(\log(1/\delta))$ weak estimates.

cumulants for sums of independent random variables are additive, we have that the excess kurtosis of $X$ is

$$\kappa = \kappa(X) = \frac{\kappa_4(X)}{\mathbf{Var}(X)^2} = \frac{\sum_{i \in [n]} \kappa_4(X_i)}{(\sum_{i \in [n]} \mathbf{Var}(X_i))^2} = \frac{\sum_{i \in [n]} (\mathbf{Var}(X_i))^2 \cdot \kappa(X_i)}{(\sum_{i \in [n]} \mathbf{Var}(X_i))^2} \,.$$

We expect $\kappa$ to vanish with $n$ since for a Gaussian distribution $W$ we have $\mathbf{E}[(W - \mathbf{E}[W])^4] = 3\sigma^4$ (and $X$ resembles a Gaussian as the number of terms increases). Note that the dense case corresponds to large number of terms in $X$ and, in particular $n \geq \Omega(B/\gamma^2)$ (see Appendix C). Therefore, we might assume here that $\frac{1}{n} \cdot B/\gamma^2 = O(1)$.

Hence, we have that

$$\mathbf{Var}(\widehat{\sigma}^2) \leq \sigma^4 \left( \frac{2}{N-1} + \frac{\kappa}{N} \right) \leq \frac{\sigma^4}{N} \cdot O(1) \,.$$

Chebyshev's inequality implies that

$$\mathbf{Pr} \left[ |\widehat{\sigma}^2 - \sigma^2| \geq t \frac{\sigma^2}{\sqrt{N}} \cdot O(1) \right] \leq \frac{1}{t^2} \,.$$

Choosing $t = \sqrt{3}$ and $N = O(3/\epsilon^2)$, we get that $|\sigma^2 - \widehat{\sigma}^2| \leq O(\epsilon) \cdot \sigma^2$ with probability at least $2/3$.

`Total Variation Gap.` Finally, we have that, using Lemma 17

$$d_{TV}(X, \mathcal{Z}(\widehat{\mu}, \widehat{\sigma}^2)) \leq d_{TV}(X, \mathcal{Z}(\mu, \sigma^2)) + d_{TV}(\mathcal{Z}(\mu, \sigma^2), \mathcal{Z}(\widehat{\mu}, \widehat{\sigma}^2)) \,.$$

Since $X$ is in dense form and the cover has radius $\epsilon$, the first quantity of the right-hand side is at most $\epsilon$ and the second one, applying Lemma 17, gives

$$d_{TV}(\mathcal{Z}(\mu, \sigma^2), \mathcal{Z}(\widehat{\mu}, \widehat{\sigma}^2)) \leq \frac{1}{2} \left( \frac{\epsilon\sigma}{\sigma} + \frac{O(\epsilon) \cdot \sigma^2}{\sigma^2} \right) = O(\epsilon) \,,$$

with high probability (where the randomness is over the estimates $\widehat{\mu}$ and $\widehat{\sigma}^2$), if $\sigma^2$ is large enough. Hence, the total variation distance between the two Gaussians is of order $O(\epsilon)$.

`Conclusion of Claim 24.` So, we get that there exists an algorithm that computes the parameters $(\widehat{\mu}, \widehat{\sigma}^2)$ of a discretized Gaussian distribution so that $d_{TV}(X, \mathcal{Z}(\widehat{\mu}, \widehat{\sigma}^2)) \leq O(\epsilon)$, with high probability, using $\widetilde{O}(1/\epsilon^2)$ samples. We set $H_D = \mathcal{Z}(\widehat{\mu}, \widehat{\sigma}^2)$. As a conclusion, there is an algorithm that given $O(\log(1/\delta)/\epsilon^2)$ samples from the target SIIURV $X$ over $\mathbb{Z}$, runs in time $O(\log(1/\delta)/\epsilon^2)$ and outputs a hypothesis discretized Gaussian distribution $H_D$ with the following guarantee: If $X$ is $\epsilon$-close to a dense form SIIURV, then it holds that $d_{TV}(H_D, X) \leq O(\epsilon)$, with probability at least $1 - \delta$. $\qquad\square$

Combining the above claims concludes the proof. $\qquad\square$

## B.2 Properly Learning SIIERVs

In the problem of learning $\mathcal{E}_T(\mathcal{A})$-SIIRVs, the learner is given the value $n$ (of the number of summands), accuracy and confidence parameters $\epsilon, \delta \in (0, 1)$ and has sample access to independent draws from an unknown $\mathcal{E}_T(\mathcal{A})$-SIIRV $X$. The goal of the learning algorithm is to output a hypothesis distribution $\widetilde{X}$ that is $\epsilon$-close to $X$ in total variation distance, i.e., $d_{TV}(X, \widetilde{X}) \leq \epsilon$, with probability at least $1 - \delta$. Recall that a weakly proper learner is an algorithm that outputs a distribution $\widetilde{X}$ that is itself a $\mathcal{E}_T(\mathcal{A}')$-SIIRV, i.e., $\widetilde{X} = \sum_{i \in [m]} \widetilde{X}_i$ with $\widetilde{X}_i \sim \mathcal{E}_T(\widetilde{a}_i)$ with $\widetilde{a}_i \in \mathcal{A}' \subseteq \mathbb{R}^k$ for any $i \in [m]$, where $m$ may be different than the input's order $n$ and $\mathcal{A}'$ is a set containing $\mathcal{A}$.

### B.2.1 Hypothesis Testing and Oracle Access

**Sampling & Evaluation Oracles.** Apart from sample access to the target distribution $X$, we will require the following: Both in the sparse and the dense case, as we will see in the proof, we must be able to perform the hypothesis selection routine for the dense cover whose elements are $\mathcal{E}_T(\mathcal{A}_\varrho)$-SIIRVs. Hence, for any two distributions in the cover $H_i, H_j$ with domain $\mathbb{Z}$, the algorithm has to

efficiently compute the mass assigned to the set $W_{ij} = \{w \in W | H_i(w) \geq H_j(w)\}$ by $H_i$ and $H_j$. In fact, even an approximate computation of these two values (in the sense of De et al. [2014]) is sufficient[7]. Essentially the SELECTHYPOTHESIS$^X$ routine requires estimates to the probabilities $H_i(W_{ij})$ and $H_j(W_{ij})$ (see the routine ESTIMATE of Claim 24 in De et al. [2014], that estimates the probability $p_i = H_i(W_{ij})$). Such estimates can be obtained using sample access to $H_k$ and access to evaluation oracles $\text{EVAL}_{H_k}$ (even approximate evaluation oracles with multiplicative accuracy) for $k = 1, 2$. In our case, this can be done using an evaluation oracle: We assume that, given the set $\mathcal{A}$, our algorithms have access to an $\mathcal{E}_{\boldsymbol{T}}(\mathcal{A}_\varrho)$ evaluation oracle, that with a parameter $\boldsymbol{a} \in \mathcal{A}_\varrho$ as an input and query $w \in \mathbb{Z}$, it outputs a the probability mass assigned to $x$ (even with some multiplicative error as in De et al. [2014]) by the distribution $\mathcal{E}_{\boldsymbol{T}}(\boldsymbol{a})$, where $\mathcal{A}_\varrho = \varrho\text{-Cone}\mathcal{A}$. Moreover, we assume sample access to the distributions in $\mathcal{E}_{\boldsymbol{T}}(\mathcal{A}_\varrho)$ and to a discretized Gaussian, in the sense that, given $(\mu, \sigma^2)$, the learning algorithm can perform independent draws from the distribution $\mathcal{Z}(\mu, \sigma^2)$.

Specifically, we assume the following. The value of the approximation tolerance $\beta$ of the oracle will be related with the learning accuracy $\epsilon$ and this relation is provided in De et al. [2014] (see Assumption 3).

**Assumption 3.** *We assume that the learning algorithm can (i) query a sample oracle with input $(\mu, \sigma^2)$ and draw a sample from $\mathcal{Z}(\mu, \sigma^2)$, (ii) query a sample oracle with input $\boldsymbol{a} \in \varrho\text{-Cone}\mathcal{A}$ and draw a sample from $\mathcal{E}_{\boldsymbol{T}}(\boldsymbol{a})$ and (iii) query a $\beta$-approximate evaluation oracle $\text{EVAL}_D(\beta)$ for $D = \mathcal{E}_{\boldsymbol{T}}(\boldsymbol{a})$ with $\boldsymbol{a}$ as in (ii) with input $x \in \mathbb{Z}$ and obtain a value $p_x$ with $D(x)/(1 + \beta) \leq p_x \leq (1 + \beta)D(x)$ for some $\beta > 0$. Moreover, given learning accuracy $\epsilon \in (0, 1)$, we assume that $(1 + \beta)^2 \leq 1 + \epsilon/8$.*

We will replace Proposition 20 with the following statement. In a similar fashion, we can obtain the analogue of Proposition 21 (see Proposition 6 in De et al. [2014]).

**Proposition 25** (Lemma 22 in De et al. [2014]). *Assume that $X, H_1, H_2$ are distributions over $W \subseteq \mathbb{Z}$.[8] Let $\epsilon, \delta \in (0, 1)$. There exists an algorithm SELECTHYPOTHESIS$^X(H_1, H_2, \epsilon, \delta)$, which is given sample access to $X$ and (i) to independent samples from $H_i$ and (ii) to a $\beta$-approximate evaluation oracle $\text{EVAL}_{H_i}(\beta)$ for $i \in \{1, 2\}$, an accuracy parameter $\epsilon$ and a confidence parameter $\delta > 0$ and has the following behavior: It draws*

$$m = O(\log(1/\delta)/\epsilon^2)$$

*samples from each of $X, H_1$ and $H_2$, it performs $O(m)$ calls to the oracles $\text{EVAL}_{H_i}(\beta)$ for $i \in \{1, 2\}$, it performs $O(m)$ arithmetic operations and if some $H_i$ has $d_{TV}(X, H_i) \leq \epsilon$, then, with probability $1 - \delta$, it outputs an index $i^\star \in \{1, 2\}$ that satisfies $d_{TV}(X, H_{i^\star}) \leq 6\epsilon$.*

**Cover Construction.** Additionally, our algorithm has to construct the cover for $\mathcal{E}_{\boldsymbol{T}}(\mathcal{A})$-sums. For both the sparse and the dense case, given the parameters $\epsilon, \varrho, \theta, B$, we can consider the set $\mathcal{A}'$ of the parameters' space where $\mathcal{A}' = \varrho\text{-Cone}\mathcal{A} \cap \{\boldsymbol{a} : \|\boldsymbol{a}\| \leq r_{\text{crit}}\}$ for some sufficiently large $r_{\text{crit}} \leq (\varrho + \frac{1}{\theta}) \cdot \ln(1/\epsilon) + \frac{1}{2\theta} \cdot \ln(B) + O(\varrho + \frac{1}{\theta})$. Using Proposition 33, we can obtain a discretization for this set in time $T_c$. Let us define the total construction time of the cover of Theorem 3 as

$$T_c^{\text{total}} = T_c(\mathcal{A}, n, \epsilon/n'_{\text{crit}}, \boldsymbol{T}, \varrho, \theta, B) + T_c(\mathcal{A}, n, \epsilon/m, \boldsymbol{T}, \varrho, \theta, B). \tag{4}$$

The first term corresponds to the sparse cover construction ($n'_{\text{crit}} = \text{poly}(B, L, 1/\gamma)/\epsilon^2$) and the second for the dense one ($m \leq n' \cdot \sqrt{B}/\gamma$).

**Remark 2** (On the runtime). *Note that when $\mathcal{A}$ is not a very complicated set, then, for any $r > 0$, the set $\mathcal{A}_\varrho \cap \mathbb{B}_r[0]$ can be $O(\epsilon)$-covered in Euclidean distance in time polynomial to the size of the cover. Therefore, in such cases, $T_c$ can be omitted from the execution time of our learning algorithm, since it is dominated by the remaining terms.*

---

[7]In our setting, given an exponential family distribution $D = \mathcal{E}_{\boldsymbol{T}}(\boldsymbol{a})$, we should be able to compute (even approximately) the value $D(x)$ given a query $x \in \mathbb{Z}$. We will require access to approximate evaluation oracles; this is natural since we may use some approximation method to estimate the partition function $Z$ and then output the value $D(x) = \exp(-\boldsymbol{a} \cdot \boldsymbol{T}(x))/\widetilde{Z}$, where $\widetilde{Z}$ is the estimation of the partition function.

[8]De et al. [2014] provide this result in the context of distributions with finite support. However, the sample and time complexity bounds they derive do not depend on the size of the support. What is in fact dependent on the size of the support is the bit complexity of the algorithms (since a sample could require, in principle, an arbitrarily large representation). In our case, we do not account for bit complexity (in fact, we focus on sample complexity). One could either think that the bit complexity is a random variable, which will, in practice take only a small number of possible values, due to the concentration properties of the distributions we examine or impose a "hard bound" on the number of bits the algorithm reads for each sample.

### B.2.2 The Result and the Algorithm

The main learning result is stated in Theorem 26.

**Theorem 26** (Weakly Proper Learner for SIIERVs). *Let $k \in \mathbb{N}$ and consider the exponential family $\mathcal{E}_{\boldsymbol{T}}(\mathcal{A})$ with $\mathcal{A} \subseteq \mathbb{R}^k$. Let $n \in \mathbb{N}$, accuracy $\epsilon \in (0,1)$, confidence $\delta \in (0,1)$ and let $X$ be an unknown $\mathcal{E}_{\boldsymbol{T}}(\mathcal{A})$-sum over $\mathbb{Z}$ of order $n$. Assume that Assumption 2 holds with parameters $\varrho, B, \gamma, \Lambda, \theta$ and that Assumption 3 holds. There exists an algorithm $\mathrm{PROPERLEARNER}^X$ (see Figure 2) with the following properties: Given $n, \epsilon, \delta$, the algorithm uses*

$$m = O\left(\frac{1}{\epsilon^2}\log(1/\delta)\right) + k \cdot \widetilde{O}(1/\epsilon^2) \cdot \mathrm{poly}\left(B, L, \frac{1}{\gamma}\right) \cdot \log\left(\frac{\varrho\sqrt{\Lambda}}{\theta}\right)$$

*samples from $X$ and, in time*

$$\mathrm{poly}\left(m, \left(\frac{\varrho \cdot \sqrt{\Lambda}}{\theta}\right)^{k \cdot \mathrm{poly}\left(B, \frac{1}{\epsilon}, \frac{1}{\gamma}\right)}, \left(n^2 \cdot \mathrm{poly}\left(B, \frac{1}{\gamma}\right) \cdot O\left(\frac{\varrho \cdot \sqrt{\Lambda}}{\theta \cdot \epsilon}\right)\right)^k, T_c^{\mathrm{total}}\right),$$

*where $T_c^{\mathrm{total}}$ is given in (4), outputs a (succinct description of a) distribution $\widetilde{X}$ which satisfies $d_{TV}(X, \widetilde{X}) \leq \epsilon$, with probability at least $1 - \delta$ and, moreover, $\widetilde{X}$ is an $\mathcal{E}_{\boldsymbol{T}}(\mathcal{A}')$-sum of order at most $(\sqrt{B}/\gamma) \cdot n$ and $\mathcal{A}' = \varrho\text{-}\mathrm{Cone}\mathcal{A}$.*

The algorithm follows. As in the SIIURV case, there are two regimes resembling to the input sum $X$ having small (i.e., sparse) or large (i.e., dense) variance and a final hypothesis testing routine. We now shortly depict the two learning sub-routines, corresponding to the small and large variance cases. When $X$ is close to a sparse form, the learning algorithm runs a tournament between all possible distributions of the sparse case and chooses the hypothesis that won each pairwise competition, i.e., the tournament's winner . The dense proper case is more challenging: Our ultimate goal is to learn the dense form hypothesis that is close to $X$ with a sample complexity that does not depend on $n$. Crucially, we have to make use of the structure of the cover. In the dense regime, the input sum $X$ is close to a discretized Gaussian random variable and its parameters can be estimated using $O(1/\epsilon^2)$ samples. Having this approximation for $X$, we run the tournament hypothesis testing procedure between *the estimated Gaussian* and the distributions of the dense form. Hence, we draw no more samples from $X$, but instead we generate draws from the Gaussian. By a union bound on the two events (i.e., the Gaussian is close to $X$ and that the winner of the tournament is close to the Gaussian), we get a dense form that is close to $X$. In the following, we may omit the "weakly proper" phrasing and simply use the term "proper".

---

*Algorithm for SIIERVs*: $(\epsilon, \delta)$-Learning SIIERVS $X = \sum_{i \in [n]} X_i$.

1. Run $\mathrm{PROPERLEARNSPARSE}^X(n, \epsilon, \delta/3)$ of Claim 27 and get the distribution $H_S$.

2. Run $\mathrm{PROPERLEARNDENSE}^X(n, \epsilon, \delta/3)$ of Claim 28 and get the distribution $H_D$.

3. Return the distribution that is the output of $\mathrm{SELECTHYPOTHESIS}^X(H_S, H_D, \epsilon, \delta/3)$ of Proposition 25.

---

Figure 2: Proper Learning algorithm for SIIERVs.

After describing these fundamental procedures, we are now ready to provide a complete proof of our main learning result (see Theorem 26).

**The Proof of Theorem 26.** The analysis of Theorem 26 works as follows: Under Assumption 2, we analyze our proper learner. As we have already mentioned, the algorithm (Figure 2) runs as follows: First, it calls the $\mathrm{PROPERLEARNSPARSE}^X$ of Claim 27 with input $(n, \epsilon, \delta/3)$ and gets the distribution $H_S$. Then, it runs $\mathrm{PROPERLEARNDENSE}^X$ of Claim 28 with input $(n, \epsilon, \delta/3)$ and gets a distribution $H_D$, which lies inside the desired distribution class. In order to conclude the proper learning part, via Proposition 25, it runs the procedure $\mathrm{SELECTHYPOTHESIS}^X(H_S, H_D)$ with parameter $\epsilon, \delta/3$. In conclusion, with probability at least $1 - \delta$, the algorithm $\mathrm{PROPERLEARNER}^X$ will

satisfy the desiderata of Theorem 26. We divide the proof in a series of claims. Before presenting the proof, we remind the reader that the following hold for the $(B^{1/2}/\gamma, \varrho\text{-Cone})$-proper $\epsilon$-cover $\mathcal{D}_{\mathrm{E}} = \mathcal{D}_{\mathrm{E}}(\epsilon) = \mathcal{D}_{\mathrm{E}}^{(s)} \cup \mathcal{D}_{\mathrm{E}}^{(d)}$ with:

$$|\mathcal{D}_{\mathrm{E}}^{(s)}| + |\mathcal{D}_{\mathrm{E}}^{(d)}| = \left( \frac{\varrho \cdot \sqrt{\Lambda}}{\theta} \right)^{k \cdot \widetilde{O}\left( \frac{1}{\epsilon^2} \right) \cdot \mathrm{poly}\left( B, L, \frac{1}{\gamma} \right)} + \left( n^2 \cdot \mathrm{poly}\left( B, \frac{1}{\gamma} \right) \cdot O\left( \frac{\varrho \cdot \sqrt{\Lambda}}{\theta \cdot \epsilon} \right) \right)^k . \quad (5)$$

**Claim 27** (Proper Learning of Sparse Instances). *Assume that Assumption 2 and Assumption 3 hold. For any $n, \epsilon, \delta > 0$, there is an algorithm* PROPERLEARNSPARSE$^X(n, \epsilon, \delta)$ *that given*

$$m = O\left( \frac{1}{\epsilon^2} \log(1/\delta) \right) + O\left( k \cdot \widetilde{O}\left( \frac{1}{\epsilon^2} \right) \cdot \mathrm{poly}\left( B, L, \frac{1}{\gamma} \right) \cdot \log\left( \frac{\varrho \cdot \sqrt{\Lambda}}{\theta} \right) \right)$$

*samples from the target $\mathcal{E}_{\boldsymbol{T}}(\mathcal{A})$-sum $X$ over $\mathbb{Z}$ of order $n$, outputs a (succint description of a) hypothesis distribution $H_S$ with the following guarantee: If $X$ is $\epsilon$-close to some element of $\mathcal{D}_{\mathrm{E}}^{(s)}(\epsilon)$ (of Equation (5)), then it holds that $d_{TV}(H_S, X) \leq c_1 \cdot \epsilon$, for some universal constant $c_1 \geq 1$, with probability at least $1 - \delta$. Moreover, if $\mathcal{D}_{\mathrm{E}}^{(s)}(\epsilon)$ is the sparse subset of the weakly proper cover $\mathcal{D}_{\mathrm{E}}(\epsilon)$, then $H_S$ lies in $\mathcal{D}_{\mathrm{E}}^{(s)}(\epsilon)$ and the running time of the algorithm is*

$$\mathrm{poly}\left( m, 2^{k \cdot \mathrm{poly}(B, 1/\epsilon, 1/\gamma) \cdot \log(\varrho\sqrt{\Lambda}/\theta)}, T_{\mathrm{c}}^{\mathrm{sparse}} \right) .$$

*In particular,*

1. *$H_S$ will be an $\mathcal{E}_{\boldsymbol{T}}(\varrho\text{-Cone}\mathcal{A})$-sum of order $\frac{1}{\epsilon^2} \cdot \mathrm{poly}(B, L, 1/\gamma)$ and,*

2. *$T_{\mathrm{c}}^{\mathrm{sparse}} = T_{\mathrm{c}}(\mathcal{A}, n, \epsilon^3/\mathrm{poly}(B, L, 1/\gamma), \boldsymbol{T}, \varrho, \theta, B)$.*

*Proof.* Let $\epsilon \in (0, 1)$. Under Assumption 2, according to our structural result, there exists a sparse form cover $\mathcal{D}_{\mathrm{E}}^{(s)} = \mathcal{D}_{\mathrm{E}}^{(s)}(\epsilon)$ of radius $\epsilon$ whose size is equal to

$$M = \left| \mathcal{D}_{\mathrm{E}}^{(s)} \right| \leq \left( \frac{\varrho \cdot \sqrt{\Lambda}}{\theta} \right)^{k \cdot \widetilde{O}(1/\epsilon^2) \cdot \mathrm{poly}\left( B, L, \frac{1}{\gamma} \right)} ,$$

and each $\mathcal{E}_{\boldsymbol{T}}(\mathcal{A})$-SIIRV of order $n'_{\mathrm{crit}}$ (the sparse SIIERVs) can be $\epsilon$-approximated by some distribution $\mathcal{E}_{\boldsymbol{T}}(\varrho\text{-Cone}\mathcal{A})$-SIIRV of order $n'_{\mathrm{crit}}$. The algorithm has to construct the cover in time $T_{\mathrm{c}}^{\mathrm{sparse}}$ (with accuracy $\epsilon/n'_{\mathrm{crit}}$ since we are in the sparse regime; this is indicated by the proof of the sparse case). Let us assume that $X$ is $\epsilon$-close to a sparse form element in the cover $\mathcal{D}_{\mathrm{E}}(\epsilon)$. Using Assumption 3, we can apply the SELECTTOURNAMENT$^X$ algorithm of De et al. [2014] (see Proposition 6 which is a variant of Proposition 21) with input the distributions' collection $\mathcal{D}_{\mathrm{E}}^{(s)}$ with accuracy $\epsilon$ and confidence $\delta$. We observe that there exists a distribution from the collection that is $\epsilon$-close in total variation distance and, so, the requirements are satisfied. Moreover, we assume that we have sample oracle access and evaluation oracle access to any distribution in $\mathcal{D}_{\mathrm{E}}^{(s)}$. We can apply the variant of Proposition 21: The algorithm makes $O\left( \frac{1}{\epsilon^2} \left( \log(M) + \log(1/\delta) \right) \right)$ draws from $X$ and from each $Y$ that is in $\mathcal{D}_{\mathrm{E}}^{(s)}$, runs in time polynomial in the number of samples and in the size of the collection and, with probability at least $1 - \delta$, outputs an index $i^\star \in [M]$ so that the sum $Y^\star$ with the corresponding parameters satisfies $d_{TV}(X, Y^\star) \leq 6\epsilon$. We set $c_1 = 6$. Moreover, in order to obtain a detailed expression for the sample complexity, we have that

$$m = O\left( \frac{1}{\epsilon^2} \left( \log(M) + \log(1/\delta) \right) \right) .$$

The running time is $\mathrm{poly}\left( m, M, T_{\mathrm{c}}^{\mathrm{sparse}} \right)$. The result follows by replacing $M$. $\qquad\square$

As a next step, we analyze the learning phase concerning the dense instances:

**Claim 28** (Proper Learning of Dense Instances). *Under Assumption 2 and Assumption 3, for any $n, \epsilon, \delta > 0$, there is an algorithm* PROPERLEARNDENSE$^X(n, \epsilon, \delta)$ *that given*

$$m = O\left(\frac{1}{\epsilon^2}\log(1/\delta)\right)$$

*samples from the target $\mathcal{E}_{\boldsymbol{T}}(\mathcal{A})$-sum $X$ over $\mathbb{Z}$ of order $n$, outputs a (succint description of a) hypothesis distribution $H_D$ with the following guarantee: If $X$ is $\epsilon$-close to a dense form $\mathcal{E}_{\boldsymbol{T}}(\mathcal{A})$-sum in the $\mathcal{D}_{\mathrm{E}}(\epsilon)$ of Equation* (5) *, then it holds that $d_{TV}(H_D, X) \leq c_1 \cdot \epsilon$, for some absolute constant $c_1 \geq 1$, with probability at least $1 - \delta$. Moreover, if $\mathcal{D}_{\mathrm{E}}^{(d)}(\epsilon)$ is the dense subset in the proper cover $\mathcal{D}_{\mathrm{E}}(\epsilon)$, then $H_D$ lies in $\mathcal{D}_{\mathrm{E}}^{(d)}(\epsilon)$ and the running time of the algorithm is*

$$\mathrm{poly}\left(m, \left(n^2 \cdot \mathrm{poly}\left(B, \frac{1}{\gamma}\right) \cdot O\left(\frac{\varrho \cdot \sqrt{\Lambda}}{\theta \cdot \epsilon}\right)\right)^k, T_{\mathrm{c}}^{\mathrm{dense}}\right) .$$

*In particular,*

1. $H_D$ *will be an $\mathcal{E}_{\boldsymbol{T}}(\varrho\text{-}\mathrm{Cone}\mathcal{A})$-sum of order $(\sqrt{B}/\gamma) \cdot n$ and,*

2. $T_{\mathrm{c}}^{\mathrm{dense}} = T_{\mathrm{c}}(\mathcal{A}, n, \epsilon \cdot \gamma/(n \cdot \sqrt{B}), \boldsymbol{T}, \varrho, \theta, B)$.

*Proof.* Let us assume that $X$ is $\epsilon$-close to a dense form $\mathcal{E}_{\boldsymbol{T}}(\mathcal{A})$-sum in the $\mathcal{D}_{\mathrm{E}}(\epsilon)$. We apply Claim 24 to the target sum with some accuracy $\epsilon'$ and confidence $\delta/2$. Hence, with high probability, after drawing $O(\log(1/\delta)/\epsilon^2)$ samples from $X$, we get parameters $\widehat{\mu}$ and $\widehat{\sigma}^2$ so that the random variable $Z \sim \mathcal{Z}(\widehat{\mu}, \widehat{\sigma}^2)$ satisfies

$$d_{TV}(X, Z) \leq \epsilon .$$

The accuracy $\epsilon'$ is chosen so that the resulting total variation gap is $\epsilon$. Our assumption about $X$ implies that there exists a distribution $Y$ is a sum of i.i.d. random variables with common parameter vector $\boldsymbol{b}$ and distribution $\mathcal{E}_{\boldsymbol{T}}(\boldsymbol{b})$, then $X$ is close to $Y$. So, there exists a cover $\mathcal{D}_{\mathrm{E}}^{(d)} = \mathcal{D}_{\mathrm{E}}^{(d)}(\epsilon)$ of radius $\epsilon$ for dense instances of size

$$\left|\mathcal{D}_{\mathrm{E}}^{(d)}\right| = \left(n^2 \cdot \mathrm{poly}\left(B, \frac{1}{\gamma}\right) \cdot O\left(\frac{\varrho \cdot \sqrt{\Lambda}}{\theta \cdot \epsilon}\right)\right)^k .$$

The algorithm constructs the cover in time $T_{\mathrm{c}}^{\mathrm{dense}}$ (with accuracy $\epsilon/(n\sqrt{B}/\gamma)$ as indicated by the proof of the dense case). By the structure of the cover, there exists a distribution $Y$ so that

1. $Y$ lies in $\mathcal{D}_{\mathrm{E}}^{(d)}$,

2. and $d_{TV}(X, Y) \leq \epsilon$.

Hence, by the triangle inequality, we have that $d_{TV}(Y, Z) \leq 2\epsilon$. We then apply the algorithm SELECTTOURNAMENT$^X$ (we use the modification of Proposition 21 (see De et al. [2014] with hypothesis selection algorithm as in Proposition 25) with the following input:

1. Let the target $D$ be the distribution of $Z$ (i.e., the discretized Gaussian),

2. consider the collection of distributions corresponding to the set $\mathcal{D}_{\mathrm{E}}^{(d)}$,

3. and accuracy $2\epsilon$ and confidence $\delta/2$.

Note that there exists a distribution in the provided collection that is $2\epsilon$-close to the target distribution and, using Assumption 3, we have the required sample and evaluation oracle access. The SELECTTOURNAMENT$^X$ procedure makes $O\left(\frac{1}{\epsilon^2}\left(\log\left|\mathcal{D}_{\mathrm{E}}^{(d)}\right| + \log(1/\delta)\right)\right)$ draws from the target $Z$ and from each distribution in the collection. This is possible and it only costs *in runtime*. Recall that we have assumed samples access to a discretized Gaussian oracle and sample and evaluation oracle access to the elements of the cover. Moreover, it runs in time polynomial in the number of

samples and in the size of the collection (i.e., the runtime depends on $n$) and, with probability at least $1 - \delta/2$, outputs an index $i^\star \in \left[\left|\mathcal{D}_{\mathrm{E}}^{(d)}\right|\right]$ so that the sum $Y^\star$ with the corresponding parameters satisfies

$$d_{TV}(Z, Y^\star) \leq 12\epsilon.$$

Hence, it holds that $d_{TV}(X, Y^\star) \leq 13\epsilon$. Let $c_1 = 13$. Applying union bound, we have that, with probability at least $1 - \delta$, the algorithm PROPERLEARNDENSE$^X$ will use

$$m = O(\log(1/\delta)/\epsilon^2)$$

samples (from the target $X$) and, in time poly $\left(m, 1/\epsilon, \log(1/\delta), \left|\mathcal{D}_{\mathrm{E}}^{(d)}\right|\right)$, it will output a distribution $H_D$ so that

1. $H_D$ lies inside $\mathcal{D}_{\mathrm{E}}^{(d)}$,

2. and $d_{TV}(X, H_D) \leq 13\epsilon$.

The result follows. $\qquad\square$

By combining Claim 27 and Claim 28 with the guarantees of Proposition 25, Theorem 26 follows.

## C   The Proof of Theorem 2 (Structural Result for SIIURVs)

*Proof of Theorem 2.* Let us consider the SIIURV $X = \sum_{i \in [n']} X_i$ for some $n' \leq n$ where the distribution of each $X_i$ satisfies Assumption 1. There exists a critical threshold value $n'_{\mathrm{crit}}$, to be decided, that indicates whether $X$ belongs to the sparse or to the dense form. Let us first consider the case where $n' \geq n'_{\mathrm{crit}}$.

Dense Case. In this case, we will approximate $X$ with a suitable discretized Gaussian random variable. Let $\mu = \mathbf{E}[X] = \sum_{i \in [n']} \mu_i$, where $\mu_i = \mathbf{E}[X_i]$ and $\sigma^2 = \mathbf{Var}(X) = \sum_{i \in [n']} \sigma_i^2$, where $\sigma_i^2 = \mathbf{Var}(X_i)$ and consider some random variable $Z_X$ with $Z_X \sim \mathcal{Z}(\mu, \sigma^2)$. Moreover, we set $\beta = \sum_{i \in [n']} \beta_i$ where $\beta_i = \sum_{i \in [n']} \mathbf{E}\left[|X_i - \mathbf{E}[X_i]|^3\right]$ and consider $\delta \in [0, 1]$ to be such that

$$\delta = \max_{i \in [n']} d_{TV}(X - X_i, X - X_i + 1).$$

If we apply the Gaussian approximation lemma (see Lemma 19), we get that

$$d_{TV}(X, Z_X) \leq O(1/\sigma) + O(\delta) + O(\beta/\sigma^3) + O(\delta\beta/\sigma^2).$$

Our goal is to control the right-hand side of this inequality. In fact, it is reasonable to upper bound the ratio between the sum of third centered moments to the variance, to lower bound the variance of $X$ and to upper bound $\delta$. In what follows, we insist on these three desiderata.

**Claim 29** (Variance Lower Bound). *It holds that* $\mathbf{Var}(X) \geq n' \cdot \gamma/4$.

*Proof.* Let us focus on a particular $X_i$ in the sum that satisfies Assumption 1. Let $M$ be a mode of the unimodal random variable $X_i$. We have that $\mathbf{Var}(X_i) \geq \frac{1}{4} \sum_{x \in \mathbb{Z}} \mathbf{Pr}[x]|x - M|^2 \geq \frac{1}{4} \min_{x \neq M} |x - M|^2 \sum_{x \neq M} \mathbf{Pr}[x] = \Theta(\gamma)$, since we can sum over $x \neq M$ and this sum has mass at least $\gamma$, also $\min_{x \neq M} |x - M|^2 = 1$. Since the random variables $X_i$ are independent, the SIIRV $X$ has variance at least $\sigma^2 = \Omega(n' \cdot \gamma)$. $\qquad\square$

**Claim 30** (Third Centered Moment - Variance Ratio). *It holds that* $\beta/\sigma^2 = O\left(\frac{B}{\gamma}\right)$.

*Proof.* We have that

$$\frac{\beta}{\sigma^2} = \frac{\sum_{i \in [n']} \beta_i}{\sum_{i \in [n']} \sigma_i^2}$$

Note that each term in the above ratio is non-negative and so we can apply Lemma 13 in order to obtain

$$\frac{\beta}{\sigma^2} \le \max_{i \in [n']} \frac{\beta_i}{\sigma_i^2}$$

Using the proof of the previous claim, we have that $\sigma_i^2 = \Omega(\gamma)$. Moreover, the fourth centered moment is upper bounded by $B$ and so

$$\frac{\beta}{\sigma^2} = O\left(\frac{B}{\gamma}\right).$$

$\square$

**Claim 31** (TV Shift). *It holds that $\delta = O\left(\frac{1}{\sqrt{1+(n'-1)\cdot(1-\gamma)}}\right)$.*

*Proof.* For a single term $X_i$, it holds that

$$d_{TV}(X_i, X_i + 1) = \frac{1}{2} \cdot \sum_{x \in \mathbb{Z}} |\mathbf{Pr}[X_i = x] - \mathbf{Pr}[X_i = x - 1]|$$

Let $M$ be a mode of $X_i$. Since $X_i$ is unimodal, we get a telescopic sum and $d_{TV}(X_i, X_i + 1) = \mathbf{Pr}[X_i = M]$. Hence, we get that the TV shift is at most $1 - \gamma$. We now apply Lemma 18 and get

$$d_{TV}(X - X_i, X - X_i + 1) = d_{TV}\left(\sum_{j \ne i} X_j, 1 + \sum_{j \ne i} X_j\right) \le \frac{\sqrt{2/\pi}}{\sqrt{\frac{1}{4} + \sum_{j \ne i} d_{TV}(X_j, X_j + 1)}}.$$

This implies that

$$d_{TV}(X - X_i, X - X_i + 1) = O\left(\frac{1}{\sqrt{1 + (n' - 1) \cdot (1 - \gamma)}}\right).$$

Taking the supremum of $i \in [n']$, we get that

$$\delta = O\left(\frac{1}{\sqrt{1 + (n' - 1) \cdot (1 - \gamma)}}\right).$$

$\square$

**Claim 32.** *For $n' \ge \Omega\left(\frac{B^2}{\gamma^3 \epsilon^2}\right)$, we get that $d_{TV}(X, \mathcal{Z}_X) \le \epsilon$.*

*Proof.* We require that $\frac{1}{\sigma} \cdot \beta/\sigma^2 \le \epsilon$, which implies that $\frac{B}{\gamma \sqrt{n' \cdot \gamma}} \le \epsilon$ and so $n' = \Omega(B^2/(\gamma^3 \epsilon^2))$. Also, we require that $\delta B/\gamma \le \epsilon$, which implies that $\sqrt{1 + (n' - 1)(1 - \gamma)} \ge \frac{B}{\gamma \epsilon}$. This is satisfied by the above choice of $n'$. Hence, we can choose $n'_{\text{crit}} = \Omega\left(\frac{B^2}{\gamma^3 \epsilon^2}\right)$. $\square$

`Sparse Case.` Let us now focus on the case $n' \le \frac{B^2}{\gamma^3 \epsilon^2}$. For the term $X_i$ with mode $M$ ($M$ could be any mode of $X_i$, since $X_i$ might have many consequent modes and still be considered unimodal), we have that

$$\mathbf{E}\,|X_i - M|^4 = O(B),$$

since $|\mathbf{E}[X_i] - M|^2 \le 3\,\mathbf{Var}(X_i)$ whenever $X_i$ is unimodal (see Johnson and Rogers [1951]).

It holds that $|x - M|^4 \mathbf{Pr}[X_i = x] \le O(B)$ for any $x \in \mathbb{Z}$. Let us consider the points $x \in \mathbb{Z}$ so that $\mathbf{Pr}[X_i = x] \ge \frac{\epsilon}{|x - M|^{3.5}}$. It holds that

$$O(B) \ge |x - M|^4 \mathbf{Pr}[X_i = x] \ge \epsilon \cdot \sqrt{|x - M|}$$

and so these points lie in

$$|x - M| \le B^2/\epsilon^2$$

We have that

$$\sum_{x:|x-M|>B^2/\epsilon^2} \mathbf{Pr}[X_i = x] \leq \sum_{x:|x-M|>B^2/\epsilon^2} \frac{\epsilon}{|x-M|^{3.5}} = O(\epsilon),$$

since $\sum_{x:|x-M|>B^2/\epsilon^2} \frac{1}{|x-M|^{3.5}} \leq \sum_{i\geq 1} 1/i^2 = \pi^2/6$. This implies that there exists a distribution supported on the bounded interval $[M - B^2/\epsilon^2, M + B^2/\epsilon^2]$ which is $(1-\epsilon)$ close in total variation to $X_i$. In order to get the desired result, we have to make $\epsilon = \widetilde{\epsilon}/n'$ and so the SIIRV $X$ will be $\widetilde{\epsilon}$ close in statistical distance to a discrete random variable $Y$ whose support is included within an interval of size at most $(n')^3 \cdot B^2/\widetilde{\epsilon}^2 = \text{poly}(B/\gamma\widetilde{\epsilon})$ (due to convolution). Moreover, for each $X_i$, the mode takes some out of $L$ at most values (due to condition (2)) and therefore there are $L^{n'} = L^{\text{poly}(B,1/\gamma,1/\widetilde{\epsilon})}$ possible choices for the interval that contains the support of $Y$ (since fixing the modes fixes the intervals corresponding to each term $X_i$).

For every such interval $\mathcal{I}$, we know that it has size at most $s = \text{poly}(B, 1/\gamma, 1/\widetilde{\epsilon})$. Each point in the interval can be assigned by $Y$ a value within $[0,1]$. Therefore, if we quantize the possible values for each point in the interval $\mathcal{I}$ into $s/\widetilde{\epsilon}$ equidistant levels, then we get $s^{s/\widetilde{\epsilon}} = 2^{\text{poly}(B,1/\gamma,1/\widetilde{\epsilon})}$ possible distributions $Y'$, corresponding to $\mathcal{I}$. We know that for some $\mathcal{I}$ there exists some distribution $Y'$ that is $O(\widetilde{\epsilon})$ close to $Y$ (and hence to $X$) in total variation distance. The total number of possibilities is $L^{\text{poly}(B,1/\gamma,1/\widetilde{\epsilon})}$. $\qquad\square$

# D  The Proof of Theorem 3 (Structural Result for SIIERVs)

*Proof.* We consider some exponential family $\mathcal{E}_{\boldsymbol{T}}(\mathcal{A})$, the class $\mathcal{E}_{\boldsymbol{T}}(\mathcal{A})$-SIIRVs of order $n$ and any random variable $X$ with distribution within this class. That is

$$X = \sum_{i\in[n']} X_i, \text{ where } n' \leq n, X_i \sim \mathcal{E}_{\boldsymbol{T}}(\boldsymbol{a}_i), \boldsymbol{a}_i \in \mathcal{A} \text{ and } (X_i)_i \text{ independent.}$$

We will show that, under our assumptions (see Assumption 2) we can approximate the distribution of any such $X$ by some distribution lying within a small subset of $\mathcal{E}_{\boldsymbol{T}}(\mathcal{A}')$-SIIRVs of order $m$, where $m \leq \varrho \cdot n$ and $\mathcal{A}' = \mathtt{Op}\mathcal{A}$, for some $\varrho \geq 1$ and some extensive set operator $\mathtt{Op}$.

**Sparse Case.** The proof we will give has two main ingredients. The first one is Theorem 5, which states that, under our assumptions, we may sparsify the parameter space into a small set $\mathcal{B} \subseteq \varrho\text{-}\mathtt{Cone}\mathcal{A}$. The first ingredient directly implies that if $n'$ is small, then the distribution of $X$ must be close to the distribution of some random variable $Y = \sum_{i\in[n']} Y_i$, where $Y_i \sim \mathcal{E}_{\boldsymbol{T}}(\boldsymbol{b}_i)$, $\boldsymbol{b}_i \in \mathcal{B}$ and $(Y_i)_i$ are independent. In particular, considering some $n'_{\text{crit}} \in \mathbb{N}$ which will be specified later, whenever $n' \leq n'_{\text{crit}}$, Theorem 5, applied for $\epsilon \leftarrow \epsilon/n'_{\text{crit}}$ gives some set $\mathcal{B} \subseteq \varrho\text{-}\mathtt{Cone}\mathcal{A}$ with $|\mathcal{B}|^{1/k} \leq \widetilde{O}(\frac{n'_{\text{crit}} \cdot \varrho \cdot \sqrt{\Lambda}}{\epsilon\cdot\theta} \cdot \log(B))$ so that each $\mathcal{E}_{\boldsymbol{T}}(\mathcal{A})$-SIIRV of order $n'_{\text{crit}}$ can be $\epsilon$-approximated by some $\mathcal{E}_{\boldsymbol{T}}(\mathcal{B})$-SIIRV of order $n'_{\text{crit}}$. Observe that there are only

$$|\mathcal{B}|^{n'_{\text{crit}}} \leq \left( \widetilde{O}\left( \frac{n'_{\text{crit}} \cdot \varrho \cdot \sqrt{\Lambda}}{\epsilon \cdot \theta} \cdot \log(B) \right) \right)^{k \cdot n'_{\text{crit}}}$$

different $\mathcal{E}_{\boldsymbol{T}}(\mathcal{B})$-SIIRVs of order $n'_{\text{crit}}$. The value $n'_{\text{crit}}$ will be obtained by the upcoming dense case.

**Dense Case.** The second ingredient is Lemma 19, which indicates that when we have many terms in the sum, then the distribution of $X$ can be accurately represented by its mean and variance alone, since $X$ will be close to a Discretized Gaussian distribution. Therefore, when $n' \geq n'_{\text{crit}}$ (where $n'_{\text{crit}}$ sufficiently large for our purposes), we might try to find some SIIERV $Y$ that also has sufficiently many terms (and hence is accurately represented by its mean and variance alone) whose expectation and variance are close to the expectation and variance of $X$ respectively.

In particular, our proof consists of the following parts.

1. We prove that a random variable $Z_X$ that follows the discretized Gaussian distribution with mean $\mathbf{E}[X]$ and variance $\mathbf{Var}(X)$ is $O(\epsilon)$-close in total variation distance to $X$, given that $n'(\geq n'_{\text{crit}})$ is large enough.

2. We then find some random variables $Y'$ where $Y' = \sum_{i\in[m]} Y_i'$ and $Y_1', \ldots, Y_m'$ are i.i.d., each following the distribution $\mathcal{E}_T(b')$ for some $b' \in \mathcal{A}$ so that $\mathbf{E}[Y'] \approx \mathbf{E}[X]$ and $\mathbf{Var}(Y') \approx \mathbf{Var}(X)$.

3. Next, we show that $Y'$ is $O(\epsilon)$-close to a discretized Gaussian random variable $Z_{Y'}$ with mean $\mathbf{E}[Y']$ and variance $\mathbf{Var}(Y')$, when $n'$ is large enough.

4. Afterwards, we show that $Z_X$ and $Z_{Y'}$ are $O(\epsilon)$-close in total variation distance.

5. Finally, we use Theorem 5 for the distribution $\mathcal{E}_T(b')$ in order to discretize the parameter vector.

In fact, our goal here is to acquire a set of inequalities of the form $n' \geq n_i$, for $n_i > 0$ so that if $n'$ satisfies all of them, then each one of the steps presented above corresponds to some $O(\epsilon)$ deviation in total variation distance.

**Step 1: Gaussian Approximation.** Let $\mu = \mathbf{E}[X] = \sum_{i\in[n']} \mu_i$, where $\mu_i = \mathbf{E}[X_i]$ and $\sigma^2 = \mathbf{Var}(X) = \sum_{i\in[n']} \sigma_i^2$, where $\sigma_i^2 = \mathbf{Var}(X_i)$ and consider some random variable $Z_X$ with $Z_X \sim \mathcal{Z}(\mu, \sigma^2)$. Let $\beta = \sum_{i\in[n']} \beta_i$ where $\beta_i = \sum_{i\in[n']} \mathbf{E}[|X_i - \mathbf{E}[X_i]|^3]$ and let $\delta \in [0,1]$ such that

$$\delta = \max_{i\in[n']} d_{TV}(X - X_i, X - X_i + 1).$$

From Lemma 19, we get that

$$d_{TV}(X, Z_X) \leq O(1/\sigma) + O(\delta) + O(\beta/\sigma^3) + O(\delta \cdot \beta/\sigma^2).$$

First, we upper bound the quantity $\beta/\sigma^2$. Using Lemma 13, we have that

$$\frac{\beta}{\sigma^2} \leq \max_{i\in[n']} \frac{\beta_i}{\sigma_i^2} \leq B/\gamma,$$

due to assumptions (4) and (7). Next, we provide a lower bound for $\sigma^2$. In particular, we get that

$$\sigma^2 \geq n' \cdot \gamma,$$

due to assumption (7). We now demand that $n'$ is large enough so that

$$\left(1 + \frac{\beta}{\sigma^2}\right) \cdot \frac{1}{\sigma} \leq O(\epsilon),$$

thereby concluding to the following demand for the number of summands $n'$:

$$n' \geq n_1, \text{ where } n_1 = O\left(\frac{B^2}{\epsilon^2 \cdot \gamma^3}\right).$$

Finally, we calculate $\delta$ and provide another demand of the form $n' \geq n_i$ for some $n_i > 0$. For any $a \in \mathcal{A}$ and any $W \sim \mathcal{E}_T(a)$, we have that

$$2d_{TV}(W, W+1) = \frac{1}{Z_T(a)} \sum_{x\in\mathbb{Z}} |\exp(-a \cdot T(x)) - \exp(-a \cdot T(x+1))|.$$

By using the unimodatily assumption (2), we get that the summation in the right hand side of the above equation is telescopic on both sides around $M_a$ and therefore

$$d_{TV}(W, W+1) = \mathbf{Pr}[W = M_a].$$

We now bound $\mathbf{Pr}[W = M_a]$ for any $a \in \mathcal{A}$ by using Lemma 35 with $\eta = 1/2$, $s = 2$ and $\kappa > 0$ to be decided. Note that by picking smaller $\eta$ we can shrink the order of $\ell$ but cannot make it smaller than $O(B^{5/4})$. We get some $\ell \leq e^{2\kappa} \cdot O(B^{2.5})$ so that

$$\mathbf{E}[|W - M_a|^2] \leq \ell^2 \cdot \mathbf{Pr}[W \neq M_a] + e^{-\kappa} \cdot O(1).$$

We pick $\kappa = \ln(O(1/\gamma))$ to get

$$\mathbf{E}[|W - M_a|^2] \leq \ell^2 \cdot \mathbf{Pr}[W \neq M_a] + \gamma/2.$$

We also know that $\mathbf{E}[|W - M_{\boldsymbol{a}}|^2] \geq \mathbf{Var}_{\boldsymbol{a}}(W) \geq \gamma$. Hence, we have

$$1 - \mathbf{Pr}[W = M_{\boldsymbol{a}}] \geq \Omega(B^5/\gamma^5) \,.$$

Moreover, by using Lemma 18, we get that

$$d_{TV}(X, X - X_i + 1) \leq \frac{\sqrt{2/\pi}}{\sqrt{\frac{1}{4} + (n' - 1)\inf_{\boldsymbol{a} \in \mathcal{A}}(1 - d_{TV}(W_{\boldsymbol{a}}, 1 + W_{\boldsymbol{a}}))}} \,,$$

where $W_{\boldsymbol{a}} \sim \mathcal{E}_{\boldsymbol{T}}(\boldsymbol{a})$.

We conlcude to the following demand for the number of summands $n'$ (so that we have $(1 + \beta/\sigma^2) \cdot \delta \leq O(\epsilon)$):

$$n' \geq n_2 \,, \text{ where } n_2 = O\left(\frac{B^7}{\epsilon^2 \cdot \gamma^7}\right) \,.$$

**Step 2: Matching Variances and Expectations.** In this step, we will find some random variable $Y'$ that is the sum of a number of i.i.d. random variables within $\mathcal{E}_{\boldsymbol{T}}(\mathcal{A})$ such that the expectation (resp. variance) of $Y'$ is close to the expectation (resp. variance) of $X$. We will split cases according to the sign of the expectation $\mathbf{E}[X]$.

- If $\mathbf{E}[X] = 0$, then we have $\sum_{i \in [n']} \mathbf{E}[X_i] = 0$, which implies that either $\mathbf{E}[X_j] = 0$ for some $j \in [n']$ (in which case we may consider $\boldsymbol{b}' = \boldsymbol{a}_j$), or that $\mathbf{E}[X_i] \cdot \mathbf{E}[X_j] < 0$, for some $i, j \in [n']$, which, since $\mathbf{E}_{\boldsymbol{a}}[W]$ is a continuous function when $\boldsymbol{a} \in \mathcal{A}$ (see Lemma 36), gives by intermediate value theorem (and the fact that $\mathcal{A}$ is connected by assumption (6)) some $\boldsymbol{b}' \in \mathcal{A}$ with $\mathbf{E}_{\boldsymbol{b}'}[W] = 0$.

  We now pick

  $$m = \left\lceil \frac{\mathbf{Var}(X)}{\mathbf{Var}_{\boldsymbol{b}'}(W)} \right\rceil \,.$$

  We have that $\mathbf{Var}(Y') = m \cdot \mathbf{Var}_{\boldsymbol{b}'}(W) \in [\mathbf{Var}(X), \mathbf{Var}(X) + \mathbf{Var}_{\boldsymbol{b}'}(W)]$ and hence we get that $\mathbf{Var}(Y') \in [\mathbf{Var}(X), \mathbf{Var}(X) + \sqrt{B}]$, due to assumption (4) and the fact that $\boldsymbol{b}' \in \mathcal{A}$. Moreover, we have that $\mathbf{E}[X] = \mathbf{E}[Y'] = 0$.

- If $\mathbf{E}[X] > 0$, then we split $X = \sum_{i \in [n']} X_i$ into three summations $X = X^+ + X^- + X^0$, according to the sign of $\mathbf{E}[X_i]$ (for example $X^+ = \sum_{i \in I^+} X_i$, where $I^+$ is the set of $i \in [n']$ so that $\mathbf{E}[X_i] > 0$). We then have that $\mathbf{E}[X^+] > |\mathbf{E}[X^-]|$ (since $\mathbf{E}[X] > 0$) and

  $$\frac{\mathbf{Var}(X)}{\mathbf{E}[X]} = \frac{\mathbf{Var}(X^+) + \mathbf{Var}(X^-) + \mathbf{Var}(X^0)}{\mathbf{E}[X^+] - |\mathbf{E}[X^-]|} \geq \frac{\mathbf{Var}(X^+)}{\mathbf{E}[X^+]} \,.$$

  Moreover, we have that

  $$\frac{\mathbf{Var}(X^+)}{\mathbf{E}[X^+]} = \frac{\sum_{i \in I^+} \mathbf{Var}(X_i)}{\sum_{i \in I^+} \mathbf{E}[X_i]} \geq \min_{i \in [n']} \frac{\mathbf{Var}(X_i)}{\mathbf{E}[X_i]} \,,$$

  since $\mathbf{Var}(X_i), \mathbf{E}[X_i] > 0$, for any $i \in I^+$. Recall that the distribution of $X_i$ is $\mathcal{E}_{\boldsymbol{T}}(\boldsymbol{a}_i)$ for some $\boldsymbol{a}_i \in \mathcal{A}$.

  Suppose, first that there exists some $j \in [n']$ so that $\mathbf{E}[X_j] \leq 0$ (i.e., $I^0 \cup I^- \neq \emptyset$). Then, there exists some $\boldsymbol{a}_j \in \mathcal{A}$ such that $\mathbf{E}_{\boldsymbol{a}_j}[W] \leq 0$. Since $\mathcal{A}$ is connected, there exists some path connecting $\boldsymbol{a}_i$ and $\boldsymbol{a}_j$. Let $\boldsymbol{a}'$ be the first point in the path between $\boldsymbol{a}_i$ and $\boldsymbol{a}_j$ (beginning from $\boldsymbol{a}_i$) so that $\mathbf{E}_{\boldsymbol{a}'}[W] = 0$. We know that there exists such a point and that when $\boldsymbol{a}$ goes from $\boldsymbol{a}_i$ to $\boldsymbol{a}'$ through the path we described, $\mathbf{E}_{\boldsymbol{a}}[W]$ always remains positive (since it is a continuous function by Lemma 36). Moreover, as $\boldsymbol{a}$ approaches $\boldsymbol{a}'$, the expectation $\mathbf{E}_{\boldsymbol{a}}[W]$ becomes arbitrarily small, while the variance $\mathbf{Var}_{\boldsymbol{a}}(W)$ remains lower bounded by $\gamma$ (due to assumption (7)). Let $P$ denote the path from $\boldsymbol{a}_i$ to $\boldsymbol{a}'$, excluding $\boldsymbol{a}'$. Then the quantity $\mathbf{Var}_{\boldsymbol{a}}(W)/\mathbf{E}_{\boldsymbol{a}}[W]$ is a continuous function of $\boldsymbol{a}$ when $\boldsymbol{a} \in P$, due to Lemma 36 and the fact that $\mathbf{E}_{\boldsymbol{a}}[W] > 0$ for any $\boldsymbol{a} \in P$. Also, we have that as $\boldsymbol{a} \to \boldsymbol{a}'$

(through the path $P$), $\mathbf{Var}_a(W)/\mathbf{E}_a[W] \to \infty$ and therefore, due to the intermediate value theorem, there exists some $b' \in P \subseteq \mathcal{A}$ so that

$$\frac{\mathbf{Var}_{b'}(W)}{\mathbf{E}_{b'}[W]} = \frac{\mathbf{Var}(X)}{\mathbf{E}[X]} \in \left[\frac{\mathbf{Var}_{a_i}(W)}{\mathbf{E}_{a_i}[W]}, \infty\right) .$$

When $\mathbf{E}[X_i] > 0$ for any $i \in [n']$, we have that

$$\frac{\mathbf{Var}(X)}{\mathbf{E}[X]} = \frac{\mathbf{Var}(X^+)}{\mathbf{E}[X^+]} \in \left[\min_{i \in [n']} \frac{\mathbf{Var}(X_i)}{\mathbf{E}[X_i]}, \max_{i \in [n']} \frac{\mathbf{Var}(X_i)}{\mathbf{E}[X_i]}\right] ,$$

since all terms are positive. By a similar continuity argument we get once again that there exists some $b' \in \mathcal{A}$ so that

$$\frac{\mathbf{Var}_{b'}(W)}{\mathbf{E}_{b'}[W]} = \frac{\mathbf{Var}(X)}{\mathbf{E}[X]} .$$

We may pick

$$m = \left\lceil \frac{\mathbf{Var}(X)}{\mathbf{Var}_{b'}(W)} \right\rceil .$$

We have that $\mathbf{Var}(Y') = m \cdot \mathbf{Var}_{b'}(W) \in [\mathbf{Var}(X), \mathbf{Var}(X) + \mathbf{Var}_{b'}(W)]$ and hence we get that $\mathbf{Var}(Y') \in [\mathbf{Var}(X), \mathbf{Var}(X) + \sqrt{B}]$, due to assumption (4) and the fact that $b' \in \mathcal{A}$. Moreover, due to the selection of $b'$ we have that

$$m = \left\lceil \frac{\mathbf{E}(X)}{\mathbf{E}_{b'}[W]} \right\rceil .$$

Hence, we get the following bound for the expectation of $Y'$ with respect to the expectation of $X$

$$\mathbf{E}[Y'] = m \cdot \mathbf{E}_{b'}[W] \in \left[\mathbf{E}[X], \mathbf{E}[X] + \mathbf{E}_{b'}[W]\right]$$

- If $\mathbf{E}[X] < 0$, then we may use an analogous reasoning as for the case that $\mathbf{E}[X] > 0$ to prove the existence of some $b' \in \mathcal{A}$ so that $\mathbf{E}[Y'] \in [\mathbf{E}[X], \mathbf{E}[X] + \mathbf{E}_{b'}[W]]$ and also $\mathbf{Var}(Y') \in [\mathbf{Var}(X), \mathbf{Var}(X) + \sqrt{B}]$.

Therefore, in any case, we have proven that for some $b' \in \mathcal{A}$, the random variable $Y' = \sum_{i \in [m]} Y'_i$ where $Y'_i$ are i.i.d. random variables following the distribution $\mathcal{E}_T(b')$ has

$$\mathbf{E}[X] \le \mathbf{E}[Y'] \le \mathbf{E}[X] + \mathbf{E}_{b'}[W] \text{ and } \mathbf{Var}(X) \le \mathbf{Var}(Y') \le \mathbf{Var}(X) + \sqrt{B} .$$

One merit of the result presented above is that the difference between the variances (resp. expectations) of $X$ and $Y'$ does not depend on the number of terms $n'$ of $X$. This is crucial in order to be able to apply Lemma 17 to show that whenever $n'$ is large enough, $X$ is close to some $Y'$ as described above.

**Step 3: $Y'$ is similar to a Gaussian.** In this step, we use the same arguments as in **Step 1** to find a sufficient condition for $n'$ so that $Y'$ is $O(\epsilon)$-close in total variation distance to some $Z_{Y'} \sim \mathcal{Z}(\mathbf{E}[Y'], \mathbf{Var}(Y'))$. In particular, the quantities of interest are three. First, the ratio of the sum of the third centralized moments of $Y'_i$ to the variance of $Y'$, for which the upper bound we provided in **Step 1** continues to hold. Second, the lower bound for the variance of $Y'$, which is $m \cdot \gamma$. Third, the shift distance $\delta_{Y'}$ in which, $m$ will appear in the denominator in the position of $n'$.

We have that $m \ge \frac{\mathbf{Var}(X)}{\mathbf{Var}_{b'}(W)} \ge n' \cdot \gamma/\sqrt{B}$. Therefore, applying the similar demands for the shift distance as in **Step 1**, we get the following sufficient demand for $n'$

$$n' \ge n_3, \text{ where } n_3 = O\left(\frac{B^{7.5}}{\epsilon^2 \cdot \gamma^8}\right) .$$

**Step 4: The Gaussian approximations are close.** In this step, we make use of Lemma 17 in order to find sufficient conditions for $n'$ so that $X$ and $Y'$ are $O(\epsilon)$ close in total variation distance. We have that $|\mathbf{E}[X] - \mathbf{E}[Y']| \leq |\mathbf{E}_{\boldsymbol{b'}}[W]| \leq \mathbf{E}_{\boldsymbol{b'}}[|W - M_{\boldsymbol{b'}}|] + |M_{\boldsymbol{b'}}| \leq B^{1/4} + L$ (due to assumptions (4) and (3)) and $|\mathbf{Var}(X) - \mathbf{Var}(Y')| \leq \sqrt{B}$ and also $\mathbf{Var}(X) \leq \mathbf{Var}(Y')$. Hence, by Lemma 17, which bounds the total variation distance between two discretized Gaussians using the differences of their parameters, we get that it is sufficient that

$$\mathbf{Var}(X) \geq (B^{1/4} + L)^2/\epsilon^2 \text{ and } \mathbf{Var}(X) \geq \sqrt{B}/\epsilon \, .$$

We know that $\mathbf{Var}(X) \geq n' \cdot \gamma$ (by assumption (7)). We arrive to the following sufficient condition for $n'$.

$$n' \geq n_4, \text{ where } n_4 = O\left(\frac{L^2 + \sqrt{B}}{\epsilon^2 \cdot \gamma^2}\right) \, .$$

Gathering all of the conditions for $n'$, we get that $n'_{\text{crit}} = \frac{1}{\epsilon^2} \cdot \text{poly}(B, L, 1/\gamma)$.

**Step 5: Discretization.** Finally, we make use of Theorem 5 in order to discretize the space of possible parameter vectors. In particular, we find a sparse set (subset of $\mathcal{A}_\varrho$) that contains (for any input distribution of $X$) some $\boldsymbol{b}$ so that if $Y = \sum_{i \in [m]} Y_i$ with $(Y_i)_i$ i.i.d. with distribution $\mathcal{E}_{\boldsymbol{T}}(\boldsymbol{b})$, then $Y, Y'$ are $O(\epsilon/m)$ close in total variation distance. We apply Theorem 5 with error margin $\epsilon/m$, using the fact that $m \leq n' \cdot \sqrt{B}/\gamma$ to quantify our results.

We get that $X$ is $O(\epsilon)$ close to $Y$ in total variation distance. $\qquad\square$

# E  Bounding the Parameter Space (Theorem 6)

## E.1  The Proof of Theorem 6 (Bounding the Parameter Space)

We restate the theorem we are going to prove for readers' convenience.

**Theorem.** Under assumptions (1), (2), (3) and (4), there exists some value $\theta = \theta(\mathcal{A}, \boldsymbol{T}) > 0$ depending on the geometric properties of $\mathcal{A}$ and $\boldsymbol{T}$, such that for any $\epsilon \in (0, 1)$ and any $\boldsymbol{a} \in \mathcal{A}$, there exists some $\boldsymbol{b} \in \varrho\text{-}\mathsf{Cone}\mathcal{A}$ with $\|\boldsymbol{b}\| \leq (\varrho + \frac{1}{\theta}) \cdot \ln(1/\epsilon) + \frac{1}{2\theta} \cdot \ln(B) + O(\varrho + \frac{1}{\theta})$ such that

$$d_{TV}(\mathcal{E}_{\boldsymbol{T}}(\boldsymbol{a}), \mathcal{E}_{\boldsymbol{T}}(\boldsymbol{b})) \leq \epsilon \, .$$

In order to show this result, we make use of Lemma 7.

*Proof.* Let $\boldsymbol{a} \in \varrho\text{-}\mathsf{Cone}\mathcal{A}$ with $\|\boldsymbol{a}\|_2 \geq r_{\text{crit}}$ with $r_{\text{crit}} \geq \varrho$ to be decided. Our goal is to provide a parameter vector $\boldsymbol{b}$ so that $\|\boldsymbol{b}\|_2 = r_{\text{crit}}$ and $d_{TV}(\mathcal{E}_{\boldsymbol{T}}(\boldsymbol{a}), \mathcal{E}_{\boldsymbol{T}}(\boldsymbol{b})) = O(\epsilon)$.

Let $W \sim \mathcal{E}_{\boldsymbol{T}}(\boldsymbol{a})$ and $W' \sim \mathcal{E}_{\boldsymbol{T}}(\boldsymbol{b})$ (at first, $\boldsymbol{a}$ and $\boldsymbol{b}$ are unspecified). We have that

$$d_{TV}(W, W') = \frac{1}{2} \sum_{x \in \mathbb{Z}} \left| \frac{\exp(-\boldsymbol{a} \cdot \boldsymbol{T}(x))}{\sum_{y \in \mathbb{Z}} \exp(-\boldsymbol{a} \cdot \boldsymbol{T}(y))} - \frac{\exp(-\boldsymbol{b} \cdot \boldsymbol{T}(x))}{\sum_{y \in \mathbb{Z}} \exp(-\boldsymbol{b} \cdot \boldsymbol{T}(y))} \right| \, .$$

Consider some mode $M_{\boldsymbol{a}}$ of $\mathcal{E}_{\boldsymbol{T}}(\boldsymbol{a})$ and some mode $M_{\boldsymbol{b}}$ of $\mathcal{E}_{\boldsymbol{T}}(\boldsymbol{b})$. Note that $Z_{\boldsymbol{T}}(\boldsymbol{a}), Z_{\boldsymbol{T}}(\boldsymbol{b}) \geq 1$. Then, we have that

$$d_{TV}(W, W') = \frac{1}{2Z_{\boldsymbol{T}}(\boldsymbol{a})Z_{\boldsymbol{T}}(\boldsymbol{b})} \sum_{x \in \mathbb{Z}} \left| \sum_{y \in \mathbb{Z}} e^{-\boldsymbol{a} \cdot \boldsymbol{T}(x) - \boldsymbol{b} \cdot \boldsymbol{T}(y)} - e^{-\boldsymbol{b} \cdot \boldsymbol{T}(x) - \boldsymbol{a} \cdot \boldsymbol{T}(y)} \right| \, .$$

By moving the absolute value inside the sum over $y \in \mathbb{Z}$, the total variation distance is

$$d_{TV}(W, W') \leq \frac{1}{2Z_{\boldsymbol{T}}(\boldsymbol{a})Z_{\boldsymbol{T}}(\boldsymbol{b})} \sum_{(x,y) \in \mathbb{Z}^2} \left| e^{-\boldsymbol{a} \cdot \boldsymbol{T}(x) - \boldsymbol{b} \cdot \boldsymbol{T}(y)} - e^{-\boldsymbol{b} \cdot \boldsymbol{T}(x) - \boldsymbol{a} \cdot \boldsymbol{T}(y)} \right| \, .$$

We will apply Lemma 35 with $\eta = 1/2$, $s = 0$ and $k = 1$ and get $\ell \leq O(B^{1/2})$. This motivates us to partition $\mathbb{Z}$ into two sets

$$Z_1 = \{x \in \mathbb{Z} : |x - M_{\boldsymbol{a}}| > \ell\} \text{ and } Z_2 = \mathbb{Z} \setminus Z_1 \, .$$

Based on $Z_1, Z_2$, we can decompose $\mathbb{Z}^2$ into four sets: $N_1 = Z_1 \times Z_1, N_2 = Z_1 \times Z_2, N_3 = Z_2 \times Z_1, N_4 = Z_2 \times Z_2$.

Set $\Delta e_{a,b} := \frac{e^{-a \cdot T(x) - b \cdot T(y)} - e^{-b \cdot T(x) - a \cdot T(y)}}{Z_T(a) Z_T(b)}$ and $S := \sum_{(x,y) \in \mathbb{Z}^2} |\Delta e_{a,b}|$. We have that

$$S = S_1 + S_2 + S_3 + S_4,$$

where $S_i = \sum_{(x,y) \in N_i} |\Delta e_{a,b}|$ and observe that an upper bound on $S$ would control the total variation distance.

Let us choose $b$. In what follows, we consider $b$ to be the parameter vector given by Lemma 7, for $r = r_{\mathrm{crit}}$ and $a \in \varrho\text{-}\mathrm{Cone}\mathcal{A}$ with $\|a\| \geq r_{\mathrm{crit}}$. We also consider $M_a = M_b$. We next upper bound each term $S_i$ separately.

Term $S_1$: For the term $S_1$, we use the fact that if $Q_\ell = \mathbf{1}\{|W - M_a| \leq \ell\}$, we get

$$\Pr_a[|W - M_a| > \ell] = \mathbf{E}[|W - M_a|^0 \cdot (1 - Q_\ell)] \leq e^{-\|a\|/\varrho} \cdot O(1),$$

and similarly for $\Pr_b[|W - M_b| > \ell]$, since $a$ and $b$ belong to $\varrho\text{-}\mathrm{Cone}\mathcal{A}$ and $\ell$ is selected accordingly, as Lemma 35 suggests.

Moreover, we have that $|\Delta e_{a,b}| \leq \frac{e^{-a \cdot T(x) - b \cdot T(y)} + e^{-b \cdot T(x) - a \cdot T(y)}}{Z_T(a) Z_T(b)}$ and therefore

$$
\begin{aligned}
S_1 &\leq 2 \cdot \Pr_a[W \in Z_1] \cdot \Pr_b[W \in Z_1] \\
&= 2 \cdot \Pr_a[|W - M_a| > \ell] \cdot \Pr_b[|W - M_b| > \ell] \\
&\leq e^{-2r_{\mathrm{crit}}/\varrho} \cdot O(1).
\end{aligned}
$$

Terms $S_2, S_3$: For $S_2$ and $S_3$, we have for similar reasons that

$$
\begin{aligned}
S_2, S_3 &\leq \Pr_a[W \in Z_1] \cdot \Pr_b[W \in Z_2] + \Pr_b[W \in Z_1] \cdot \Pr_a[W \in Z_2] \\
&\leq \Pr_a[|W - M_a| > \ell] + \Pr_b[|W - M_b| > \ell] \\
&\leq e^{-r_{\mathrm{crit}}/\varrho} \cdot O(1).
\end{aligned}
$$

Term $S_4$: For the term $S_4$, we split $N_4$ to $N_4^{(1)}, N_4^{(2)}, N_4^{(3)}, N_4^{(4)}$ and form the four sums $S_4^{(1)}$, $S_4^{(2)}, S_4^{(3)}, S_4^{(4)}$ (which sum to $S_4$), similarly to how we split $\mathbb{Z}^2$ into $N_1, N_2, N_3, N_4$. In this case, we consider $Z_1' = \{x \in Z_2 : \Pr_a[W = x] \leq e^{-\theta r_{\mathrm{crit}}} \Pr_a[W = M]\}$ and $Z_2' = Z_2 \setminus Z_1'$.

We know that $|Z_2| \leq 2\ell$ and therefore $\Pr_a[W \in Z_1'] \leq 2\ell \cdot e^{-\theta r_{\mathrm{crit}}}$ and, due to the selection of $b$ (according to Lemma 7), we also have that $\Pr_b[W \in Z_1'] \leq 2\ell \cdot e^{-\theta r_{\mathrm{crit}}}$. Hence, with a similar reasoning as the one used for $S_1, S_2, S_3$ and since $\ell = O(B^{1/2})$ we have

$$S_4^{(1)} \leq e^{-2\theta r_{\mathrm{crit}}} \cdot O(B),$$
$$S_4^{(2)}, S_4^{(3)} \leq e^{-\theta r_{\mathrm{crit}}} \cdot O(B^{1/2}).$$

It remains to bound $S_4^{(4)}$. We have

$$
\begin{aligned}
S_4^{(4)} &= \sum_{(x,y) \in Z_2' \times Z_2'} |\Delta e_{a,b}| \\
&= \sum_{(x,y) \in N_4^{(4)}} \frac{\left| \frac{\Pr_a[W=x]}{\Pr_a[W=M_a]} \cdot \frac{\Pr_b[W=y]}{\Pr_b[W=M_b]} - \frac{\Pr_b[W=x]}{\Pr_b[W=M_b]} \cdot \frac{\Pr_a[W=y]}{\Pr_a[W=M_a]} \right|}{e^{a \cdot T(M_a)} \cdot Z_T(a) \cdot e^{b \cdot T(M_b)} \cdot Z_T(b)} = 0,
\end{aligned}
$$

due to the selection of $b$ according to Lemma 7. Therefore, in total, we pick

$$r_{\mathrm{crit}} = \varrho \cdot \ln(1/\epsilon) + \frac{1}{2\theta} \cdot \ln(B) + \frac{1}{\theta} \cdot \ln(1/\epsilon) + O(\varrho + 1/\theta),$$

and get that $d_{TV}(W, W') \leq \epsilon$. $\qquad\square$

## E.2 The Proof of Lemma 7 (Structural Distance & Bounding Norms)

In order to show Lemma 7, we will rely on the geometry induced by the exponential family distributions. Let us restate this result.

**Lemma** (Structural Distance & Bounding Norms). *Under assumptions* (1), (2) *and* (3), *there exists some constant $\theta > 0$ such that for any $r \geq \varrho$ and any $\boldsymbol{a} \in \mathcal{A}$ with $\|\boldsymbol{a}\| \geq r$, there exists some $\boldsymbol{b} \in \mathcal{A}_\varrho$ and $\|\boldsymbol{b}\| = r$ so that $d_{\mathrm{ST}}(\mathcal{E}_{\boldsymbol{T}}(\boldsymbol{a}), \mathcal{E}_{\boldsymbol{T}}(\boldsymbol{b})) \leq e^{-\theta \cdot r}$, i.e., for any $x \in \mathbb{Z}$, at least one of the following should hold:*

- *Either $x$ satisfies $\mathbf{Pr}_{\boldsymbol{a}}(x) \leq e^{-\theta \cdot r} \cdot \mathbf{Pr}_{\boldsymbol{a}}(M_{\boldsymbol{a}})$ and $\mathbf{Pr}_{\boldsymbol{b}}(x) \leq e^{-\theta \cdot r} \cdot \mathbf{Pr}_{\boldsymbol{b}}(M_{\boldsymbol{b}})$,*

- *or $x$ satisfies $\mathbf{Pr}_{\boldsymbol{a}}(x)/\mathbf{Pr}_{\boldsymbol{a}}(M_{\boldsymbol{a}}) = \mathbf{Pr}_{\boldsymbol{b}}(x)/\mathbf{Pr}_{\boldsymbol{b}}(M_{\boldsymbol{b}})$.*

*Proof of Lemma 7.* We decompose the proof into a number of steps.

**Alternative form of Lemma 7.** We can formulate a geometric framework through the observation that $\mathbf{Pr}_{\boldsymbol{a}}[W = x] \propto \exp(-\boldsymbol{a} \cdot \boldsymbol{T}(x))$ for any $\boldsymbol{a} \in \mathcal{A}_\varrho$. In particular, we have that

$$\Pr_{\boldsymbol{a}}[W = x] \geq \Pr_{\boldsymbol{a}}[W = y] \text{ is equivalent with } \boldsymbol{a} \cdot (\boldsymbol{T}(y) - \boldsymbol{T}(x)) \geq 0. \tag{6}$$

Using relation (6), we arrive to the following equivalent formulation for Lemma 7. In particular, the structural distance states that there exists some $\theta > 0$ such that for any $r \geq \varrho$ ($\varrho$ is defined in Assumption 2) and any $\boldsymbol{a} \in \mathcal{A}$ with $\|\boldsymbol{a}\| \geq r$ there exists some $\boldsymbol{b} \in \mathcal{A}_\varrho$ (recall that for $\varrho > 0$, $\mathcal{A}_\varrho = \varrho$-Cone $\mathcal{A}$, i.e., the superset of $\mathcal{A}$ that also contains every vector in the conical hull of $\mathcal{A}$ that has norm at least $\varrho$) with $\|\boldsymbol{b}\| = r$ such that $\mathbb{Z} = \mathcal{X}_1 \cup \mathcal{X}_2$ where $\mathcal{X}_1$ and $\mathcal{X}_2$ are defined as follows

1. $\mathcal{X}_1 \subseteq \mathbb{Z}$ so that for any $x \in \mathcal{X}_1$ we have
$$\boldsymbol{a} \cdot (\boldsymbol{T}(x) - \boldsymbol{T}(M_{\boldsymbol{a}})) \geq \theta r \text{ and } \boldsymbol{b} \cdot (\boldsymbol{T}(x) - \boldsymbol{T}(M_{\boldsymbol{b}})) \geq \theta r,$$

2. $\mathcal{X}_2 \subseteq \mathbb{Z}$ so that for any $x \in \mathcal{X}_2$ we have
$$\boldsymbol{a} \cdot (\boldsymbol{T}(x) - \boldsymbol{T}(M_{\boldsymbol{a}})) = \boldsymbol{b} \cdot (\boldsymbol{T}(x) - \boldsymbol{T}(M_{\boldsymbol{b}})),$$

where $M_{\boldsymbol{a}}$ (resp. $M_{\boldsymbol{b}}$) is any mode of $\mathcal{E}_{\boldsymbol{T}}(\boldsymbol{a})$ (resp. $\mathcal{E}_{\boldsymbol{T}}(\boldsymbol{b})$).

Our goal is to select the parameter $\theta > 0$ appropriately so that for any given $\boldsymbol{a} \in \mathcal{A}$, we can find $\boldsymbol{b} \in \mathcal{A}_\varrho$ with $\|\boldsymbol{b}\| = r$ such that any $x \in \mathbb{Z}$ either belongs in $\mathcal{X}_1$ or $\mathcal{X}_2$.

**Step 1.** First, note that any mode (global maximum point of the probability mass function) of the distribution $\mathcal{E}_{\boldsymbol{T}}(\boldsymbol{a})$ cannot be in $\mathcal{X}_1$ (since $\theta, r > 0$ and $\boldsymbol{a} \cdot \boldsymbol{T}(y) = \boldsymbol{a} \cdot \boldsymbol{T}(y')$ whenever $y, y'$ are modes). Therefore we get that $\mathcal{E}_{\boldsymbol{T}}(\boldsymbol{a})$ and $\mathcal{E}_{\boldsymbol{T}}(\boldsymbol{b})$ must have the same set of modes. We define the regions $\mathcal{R}_M$ of the parameter vectors that correspond to distributions with $M$ as a mode. In particular, such regions are defined by the property that for any $\boldsymbol{u} \in \mathcal{R}_M$ it holds that $\mathbf{Pr}_{\boldsymbol{u}}[W = M] \geq \mathbf{Pr}_{\boldsymbol{u}}[W = x]$, for any $x \in \mathbb{Z}$ (if $\mathcal{E}_{\boldsymbol{T}}(\boldsymbol{u})$ is well defined), or using relation (6), more generally as follows

$$\mathcal{R}_M = \{\boldsymbol{u} \in \mathbb{R}^k : \boldsymbol{u} \cdot (\boldsymbol{T}(x) - \boldsymbol{T}(M)) \geq 0, \text{ for any } x \in \mathbb{Z}\}. \tag{7}$$

Note that the sets $\mathcal{R}_M$ are convex cones that could be polyhedral cones in the case that a finite number of points $x \in \mathbb{Z}$ correspond to a set of restrictions that implies the remaining ones. We also define, for any $\mathcal{M} \subseteq \mathbb{Z}$, intersections of such sets as follows:

$$\mathcal{R}_{\mathcal{M}} = \bigcap_{M \in \mathcal{M}} \mathcal{R}_M. \tag{8}$$

For the demand that $\mathcal{M}_{\boldsymbol{a}} = \mathcal{M}_{\boldsymbol{b}}$ to be satisfied we must (at least) pick $\boldsymbol{b}$ so that

$$\boldsymbol{b} \in \mathcal{R}_{\mathcal{M}_{\boldsymbol{a}}}. \tag{9}$$

In order to develop some intuition about the regions of the form $\mathcal{R}_{\mathcal{M}}$, one might consider $\mathcal{M} = \{M, M'\} \subseteq \mathbb{Z}$. In this case

$$\begin{aligned} \mathcal{R}_{\mathcal{M}} &= \mathcal{R}_M \cap \mathcal{R}_{M'} \\ &= \{\boldsymbol{u} : \boldsymbol{u} \cdot (\boldsymbol{T}(x) - \boldsymbol{T}(M)) \geq 0 \text{ and } \boldsymbol{u} \cdot (\boldsymbol{T}(x) - \boldsymbol{T}(M')) \geq 0, \text{ for any } x \in \mathbb{Z}\} \\ &= \mathcal{R}_M \cap \{\boldsymbol{u} : \boldsymbol{u} \cdot (\boldsymbol{T}(M) - \boldsymbol{T}(M')) = 0\}. \end{aligned}$$

Therefore, if $\boldsymbol{T}(M) \neq \boldsymbol{T}(M')$, then the dimension of $\mathcal{R}_\mathcal{M}$ is at most $k-1$ (and this can be generalized for larger sets $\mathcal{M}$ by using the notion of affine independence). For any $M \in \mathbb{Z}$, the set $\mathcal{R}_M$ is a countable intersection of halfspaces of the form $\mathcal{H} = \{\boldsymbol{u} \in \mathbb{R}^k : \boldsymbol{u} \cdot \boldsymbol{h} \geq 0\}$. If $M \in \mathcal{M} \subseteq \mathbb{Z}$, then $\mathcal{R}_\mathcal{M}$ is a subset of the boundary of $\mathcal{R}_M$, since for any $M' \in \mathcal{M}$, the vector $\boldsymbol{T}(M') - \boldsymbol{T}(M)$ corresponds to some of the halfspaces that define $\mathcal{R}_M$.

**Step 2.** Our goal is to pick $\boldsymbol{b}$ so that any $x \in \mathbb{Z}$ lies in either $\mathcal{X}_1$ or $\mathcal{X}_2$. En route, we will use Assumption 2. In this step we will get rid of $x \in \mathbb{Z}$ for which we get for free that $x \in \mathcal{X}_1$, due to the fact that $\boldsymbol{a}, \boldsymbol{b} \in \mathcal{A}_\varrho$ anyway. We will use the following assumption.

Let $M_{\boldsymbol{a}}$ be any mode of $\mathcal{E}_{\boldsymbol{T}}(\boldsymbol{a})$. Then, we may define the sequence of vectors $(\boldsymbol{v}_x)_{x \in \mathbb{Z}}$ by $\boldsymbol{v}_x = \boldsymbol{T}(x) - \boldsymbol{T}(M_{\boldsymbol{a}})$ and reformulate $\mathcal{X}_1 = \{x \in \mathbb{Z} : \boldsymbol{a} \cdot \boldsymbol{v}_x \geq \theta r, \boldsymbol{b} \cdot \boldsymbol{v}_x \geq \theta r\}$ as well as $\mathcal{X}_2 = \{x \in \mathbb{Z} : \boldsymbol{a} \cdot \boldsymbol{v}_x = \boldsymbol{b} \cdot \boldsymbol{v}_x\}$. We are allowed to use $\boldsymbol{v}_x$ for both $\boldsymbol{a}$ and $\boldsymbol{b}$ in the definitions of $\mathcal{X}_1$ and $\mathcal{X}_2$, since, according to **Step 1**, vector $\boldsymbol{b}$ has to be selected within $\mathcal{R}_{\mathcal{M}_{\boldsymbol{a}}}$ anyway.

We will first classify (to $\mathcal{X}_1$) the points $x \in \mathbb{Z}$ for which the hyperplane defined by $\boldsymbol{v}_x$ does not correspond to any boundary of $\mathcal{R}_{\mathcal{M}_{\boldsymbol{a}}}$. That is to say, $\boldsymbol{v}_x \cdot \boldsymbol{u} > 0$ for any $\boldsymbol{u} \in \mathcal{R}_{\mathcal{M}_{\boldsymbol{a}}} \cap \mathcal{A}_\varrho$ with $\boldsymbol{u} \neq 0$. In particular, we define for any $\mathcal{M} \subseteq \mathcal{M}_\mathcal{A}$, the following set of points

$$\mathcal{Y}_\mathcal{M} = \{x \in \mathbb{Z} : \boldsymbol{u} \cdot (\boldsymbol{T}(x) - \boldsymbol{T}(M)) > 0, \text{ for any } \boldsymbol{u} \in \mathcal{R}_\mathcal{M} \cap \mathcal{A}_\varrho \text{ with } \boldsymbol{u} \neq 0 \text{ and } M \in \mathcal{M}\}.$$

Our goal here will be to show that there exists some constant $\theta_1 > 0$ such that for any $\boldsymbol{u} \in \mathcal{A}_\varrho$ and any $y \in \mathcal{Y}_{\mathcal{M}_{\boldsymbol{u}}}$ we have that $\boldsymbol{u} \cdot (\boldsymbol{T}(y) - \boldsymbol{T}(M_{\boldsymbol{u}})) \geq \theta_1 \|\boldsymbol{u}\|$.

To this end, observe, first, that due to assumption (3), the number of different possible $\mathcal{M} \subseteq \mathcal{M}_\mathcal{A}$ must be finite. Therefore, if we show that for every fixed $\mathcal{M} \subseteq \mathcal{M}_\mathcal{A}$ there exists some constant that satisfies the desired property for any $\boldsymbol{u} \in \mathcal{R}_\mathcal{M} \cap \mathcal{A}_\varrho$, then by taking the minimum over the selection of $\mathcal{M}$, we can find the target $\theta_1$ (swap of logical quantifiers).

For a fixed $\mathcal{M} \subseteq \mathcal{M}_\mathcal{A}$, we consider any vector $\boldsymbol{u} \in \mathcal{R}_\mathcal{M} \cap \mathcal{A}_\varrho$. Note that the only guarantee we have is that $\mathcal{M} \subseteq \mathcal{M}_{\boldsymbol{u}} \subseteq \mathcal{M}_\mathcal{A}$. Let $x_1, x_2 \in \mathcal{M}_{\boldsymbol{u}}$ be the smallest and largest elements of $\mathcal{M}_{\boldsymbol{u}}$, respectively (i.e., $x_1 \leq x \leq x_2$ for any $x \in \mathcal{M}_{\boldsymbol{u}}$). Note that $\mathcal{Y}_\mathcal{M} \cap \mathcal{M}_{\boldsymbol{u}} = \emptyset$ by construction since $\boldsymbol{u} \in \mathcal{R}_\mathcal{M} \cap \mathcal{A}_\varrho$, $\boldsymbol{u} \neq 0$ and therefore $x_1, x_2 \notin \mathcal{Y}_\mathcal{M}$. Consider $y_1, y_2 \in \mathcal{Y}_\mathcal{M}$ with $y_1 \leq x_1$ and largest possible and $y_2 \geq x_2$ and smallest possible. Then, due to unimodality (assumption (2)), we have that $\boldsymbol{u} \cdot \boldsymbol{T}(y) \geq \min\{\boldsymbol{u} \cdot \boldsymbol{T}(y_1), \boldsymbol{u} \cdot \boldsymbol{T}(y_2)\}$ for any $y \in \mathcal{Y}_\mathcal{M}$. Since $\mathcal{M}_{\boldsymbol{u}} \subseteq \mathcal{M}_\mathcal{A}$, the possible values for $(x_1, x_2)$ are finite and therefore the possible values for $(y_1, y_2)$ are also finite (since given $\mathcal{M}$, there is a 1-1 correspondence between $(x_1, x_2)$ and $(y_1, y_2)$). We may, therefore split $\mathcal{R}_\mathcal{M} \cap \mathcal{A}_\varrho$ into a finite number of equivalence classes with respect to the minimum and maximum point $(x_1, x_2)$ of the set of modes corresponding to the vector $\boldsymbol{u}$. It is sufficient to find for any equivalence class a (possibly different) constant that satisfies the desired property for any $\boldsymbol{u}$ in the class. Then we could minimize over the equivalence classes to find $\theta_1 > 0$ as desired.

Consider now the equivalence class $\mathfrak{C}$ corresponding to some fixed pair $(x_1, x_2)$ (which gives $(y_1, y_2)$). Then, for any $\boldsymbol{u} \in \mathfrak{C}$, we have $\boldsymbol{u} \cdot \boldsymbol{T}(y) \geq \min\{\boldsymbol{u} \cdot \boldsymbol{T}(y_1), \boldsymbol{u} \cdot \boldsymbol{T}(y_2)\}$ or equivalently, that $\boldsymbol{u}_\varrho \cdot (\boldsymbol{T}(y) - \boldsymbol{T}(M_{\boldsymbol{u}})) \geq \min\{\boldsymbol{u}_\varrho \cdot (\boldsymbol{T}(y_1) - \boldsymbol{T}(M_{\boldsymbol{u}})), \boldsymbol{u}_\varrho \cdot (\boldsymbol{T}(y_2) - \boldsymbol{T}(M_{\boldsymbol{u}}))\}$ for any $y \in \mathcal{Y}_\mathcal{M}$, where $\boldsymbol{u}_\varrho = \varrho \cdot \boldsymbol{u}/\|\boldsymbol{u}\|$. Also, $\mathfrak{C} \subseteq \mathcal{R}_\mathcal{M} \cap \mathcal{A}_\varrho$ and $\mathcal{R}_\mathcal{M} \cap \mathtt{Cone}\mathcal{A}$ is a cone and therefore $\mathcal{R}_\mathcal{M} \cap \mathcal{A}_\varrho$ contains all vectors $\boldsymbol{u}_\varrho$ where $\boldsymbol{u} \in \mathfrak{C}$. Moreover, $\mathcal{R}' := \mathcal{R}_\mathcal{M} \cap \mathcal{A}_\varrho \cap \{\boldsymbol{u}' : \|\boldsymbol{u}'\| = \varrho\}$ is closed (since $\mathcal{A}$ is closed by assumption (1)). Therefore, for any $\boldsymbol{u} \in \mathfrak{C}$ and any $y \in \mathcal{Y}_\mathcal{M}$:

$$\boldsymbol{u}_\varrho \cdot (\boldsymbol{T}(y) - \boldsymbol{T}(M_{\boldsymbol{u}})) \geq \min\left\{\inf_{\boldsymbol{u}' \in \mathcal{R}'} \boldsymbol{u}' \cdot (\boldsymbol{T}(y_1) - \boldsymbol{T}(M_{\boldsymbol{u}})), \inf_{\boldsymbol{u}' \in \mathcal{R}'} \boldsymbol{u}' \cdot (\boldsymbol{T}(y_2) - \boldsymbol{T}(M_{\boldsymbol{u}}))\right\}.$$

We know that $\boldsymbol{u}' \cdot (\boldsymbol{T}(y_1) - \boldsymbol{T}(M_{\boldsymbol{u}})), \boldsymbol{u}' \cdot (\boldsymbol{T}(y_2) - \boldsymbol{T}(M_{\boldsymbol{u}})) > 0$ for any $\boldsymbol{u}' \in \mathcal{R}_\mathcal{M} \cap \mathcal{A}_\varrho$, since $y_1, y_2 \in \mathcal{Y}_\mathcal{M}$. Since, additionally $\mathcal{R}'$ is closed, the infima in the above inequality are attained for some vectors $\boldsymbol{u}'_1, \boldsymbol{u}'_2 \in \mathcal{R}'$ and correspond to positive values $\theta'_{11}, \theta'_{12} > 0$.

We have proven that there exists some $\theta_1 > 0$ so that for any $\boldsymbol{u} \in \mathcal{A}_\varrho$ and any $y \in \mathcal{Y}_{\mathcal{M}_{\boldsymbol{u}}}$ we have that $\boldsymbol{u} \cdot (\boldsymbol{T}(y) - \boldsymbol{T}(M_{\boldsymbol{u}})) \geq \theta_1 \|\boldsymbol{u}\|$. As a consequence, returning to our vectors $\boldsymbol{a}$ (given vector) and the desired $\boldsymbol{b}$, since $\boldsymbol{a}, \boldsymbol{b} \in \mathcal{A}_\varrho$, we have that if we pick $\theta \leq \theta_1$, then $\mathcal{Y}_{\mathcal{M}_{\boldsymbol{a}}} \subseteq \mathcal{X}_1$.

**Step 3.** It remains to account for the points $x \in \mathbb{Z} \setminus \mathcal{Y}_{\mathcal{M}_{\boldsymbol{a}}}$ (i.e., find conditions for the selection of $\boldsymbol{b}$ so that any such $x$ is classified either in $\mathcal{X}_1$ or $\mathcal{X}_2$). The first crucial observation is that the set $\mathbb{Z} \setminus \mathcal{Y}_{\mathcal{M}_{\boldsymbol{a}}}$ must be finite. In particular, $\mathbb{Z} \setminus \mathcal{Y}_{\mathcal{M}_{\boldsymbol{a}}}$ consists of points $x$ such that the boundary of the halfspace defined by $\boldsymbol{v}_x$ intersects the set $\mathcal{R}_{\mathcal{M}_{\boldsymbol{a}}} \cap \mathcal{A}_\varrho$, due to the definition of $\mathcal{Y}_{\mathcal{M}_{\boldsymbol{a}}}$. Consider some

vector $\boldsymbol{u} \in \mathcal{R}_{\mathcal{M}_a} \cap \mathcal{A}_\varrho$ with $\boldsymbol{u} \cdot \boldsymbol{v}_x = 0$. The vector $\boldsymbol{u}$ corresponds to some distribution in $\mathcal{E}_{\boldsymbol{T}}(\mathcal{A}_\varrho)$ and $x$ is a mode of $\boldsymbol{u}$ since $\boldsymbol{u} \cdot (\boldsymbol{T}(x) - \boldsymbol{T}(M_a)) = 0$ and $\boldsymbol{u} \in \mathcal{R}_{\mathcal{M}_a}$. Hence $\mathbb{Z} \setminus \mathcal{Y}_{\mathcal{M}_a} \subseteq \mathcal{M}_{\mathcal{A}_\varrho}$. Due to assumption (3), $|\mathcal{M}_{\mathcal{A}_\varrho}|$ is finite and so does $|\mathbb{Z} \setminus \mathcal{Y}_{\mathcal{M}_a}|$.

The next observation we will use is that $\mathcal{R}_{\mathcal{M}_a} \cap \mathtt{Cone}\mathcal{A}$ is a polyhedral cone, due to assumption (1), and for any $x \in \mathbb{Z} \setminus \mathcal{Y}_{\mathcal{M}_a}$, $\boldsymbol{v}_x \cdot \boldsymbol{u} \geq 0$ for any $\boldsymbol{u} \in \mathcal{R}_{\mathcal{M}_a} \cap \mathcal{A}_\varrho$, while $\mathbb{Z} \setminus \mathcal{Y}_{\mathcal{M}_a}$ is finite. In particular, we consider the matrix $H \in \mathbb{R}^{k \times t}$, where $t \in \mathbb{N}$ and $H$ contains as columns all the vectors of the form $\boldsymbol{v}_x$ for $x \in \mathbb{Z} \setminus \mathcal{Y}_{\mathcal{M}_a}$. However, $H$ could have some additional columns so that $\mathcal{R}_{\mathcal{M}_a} \cap \mathcal{A}_\varrho = \{\boldsymbol{u} \in \mathbb{R}^k : H^T \boldsymbol{u} \geq 0\}$. We may apply Theorem 8 accordingly to get a bound for $\theta$ and a way to pick $\boldsymbol{b}$ that imply the desired result (with appropriate rescaling). Note that the bound we get for $\theta$ can be considered independent from $\boldsymbol{a}$, since there is only a finite number of possible selections of $\mathcal{M}_a$ and we may minimize over them to get a global bound. $\qquad \square$

We now briefly discuss the intuition behind Theorem 8. In particular, since for any $x \in \mathcal{X}_1$ we must have that $\boldsymbol{a} \cdot \boldsymbol{v}_x$ is large enough, there should be some threshold for $\boldsymbol{a} \cdot \boldsymbol{v}_x$ below which we know that $x$ has to be classified in $\mathcal{X}_2$. One idea would be to decide the set in which $x$ should be classified by considering $x \in \mathcal{X}_2$ exactly when $\boldsymbol{a} \cdot \boldsymbol{v}_x$ is below some threshold (of the form $\theta \cdot r$). Then, if we want to classify $x$ to $\mathcal{X}_1$, we could pick any $\boldsymbol{b}$ so that $\|\boldsymbol{b}\| = r$ and $\cos(\boldsymbol{b}, \boldsymbol{v}_x)$ is large enough. To this end, we may only perturb the direction of $\boldsymbol{b}$, since its norm is restricted a priori. Consequently, (recall that $\boldsymbol{b} \cdot \boldsymbol{v}_x = \|\boldsymbol{b}\| \cdot \cos(\boldsymbol{b}, \boldsymbol{v}_x) \cdot \|\boldsymbol{v}_x\|$) $\boldsymbol{b} \cdot \boldsymbol{v}_x$ is also at least equal to the threshold. If $x$ should be classified to $\mathcal{X}_2$, then we should pick $\boldsymbol{b} = \boldsymbol{a} + \boldsymbol{u}$, where $\boldsymbol{u} \cdot \boldsymbol{v}_x = 0$. The main complication here is that we are not interested in classifying only a single point $x$, but, rather, any point $x \in \mathbb{Z} \setminus \mathcal{Y}_{\mathcal{M}_a}$. For different points in $\mathbb{Z} \setminus \mathcal{Y}_{\mathcal{M}_a}$ we would then have different restrictions for $\boldsymbol{b}$, which could be mutually exclusive. Theorem 8 states that, due to the structure of polyhedral cones, there is a way to satisfy all such restrictions simultaneously.

### E.3 The Proof of Theorem 8 (Geometric Result for Polyhedral Cones)

Let us restate the theorem for reader's convinience.

**Theorem.** *Consider any polyhedral cone $\mathcal{C} \subseteq \mathbb{R}^k$, $k \in \mathbb{N}$, where $\mathcal{C} = \{\boldsymbol{u} : H^T \boldsymbol{u} \geq 0\}$ for some matrix $H \in \mathbb{R}^{k \times t}$, $t \in \mathbb{N}$ is a description of $\mathcal{C}$ as an intersection of halfspaces. Then there exists some $\theta > 0$ such that for any $\boldsymbol{u} \in \mathcal{C}$ with $\|\boldsymbol{u}\| \geq 1$, there exists $\boldsymbol{u}' \in \mathcal{C}$ with $\|\boldsymbol{u}'\| = 1$ so that for any column $\boldsymbol{h}$ of $H$ at least one of the following is true:*

1. *Either $\boldsymbol{h} \cdot \boldsymbol{u} \geq \theta$ and $\boldsymbol{h} \cdot \boldsymbol{u}' \geq \theta$,*

2. *or $\boldsymbol{h} \cdot \boldsymbol{u} = \boldsymbol{h} \cdot \boldsymbol{u}'$.*

*Proof.* Due to Minkowski-Weyl theorem (Proposition 11), there exists some $s \in \mathbb{Z}$ and some matrix $Z \in \mathbb{R}^{k \times s}$ so that $\mathcal{C} = \{\boldsymbol{u} \in \mathbb{R}^k : \boldsymbol{u} = Z\boldsymbol{x}, \boldsymbol{x} \geq 0\}$. Let $(\boldsymbol{h}_i)_{i \in [t]}$ be the columns of $H$ and $(\boldsymbol{z}_i)_{i \in [s]}$ the columns of $Z$. Suppose without loss of generality that $\|\boldsymbol{z}_j\| = 1$ for any $j \in [s]$.

**Step 1.** In this step, we will find some bound for $\theta$ which implies the existence of a vector $\boldsymbol{w}$ which has some useful properties for the next step. We will use $\boldsymbol{w}$ as a point of reference for moving $\boldsymbol{u}$ to get $\boldsymbol{u}'$. We consider the quantity

$$\theta_1 := \min_{i,j}\{\boldsymbol{h}_i \cdot \boldsymbol{z}_j \mid i \in [t], j \in [s] : \boldsymbol{h}_i \cdot \boldsymbol{z}_j > 0\}.$$

Note that since $s, t < \infty$, the quantity $\theta_1$ (when it is well defined) is positive (and only depends on the columns of $H$ and the geometry of $\mathcal{C}$).

In case $\boldsymbol{h}_i \cdot \boldsymbol{z}_j = 0$ for any $i, j$, we have that for any $\boldsymbol{u} \in \mathcal{C}$, $\boldsymbol{h}_i \cdot \boldsymbol{u} = 0$ for any $i \in [t]$ (since $\mathcal{C}$ is generated by the columns of $Z$) and therefore we may consider $\boldsymbol{u}' = \boldsymbol{u}/\|\boldsymbol{u}\|$, which satisfies all the required properties.

Next, we observe that there must exist some vector $\boldsymbol{x} \in \mathbb{R}^s$ with $\boldsymbol{x} \geq \boldsymbol{1}$ (component-wise) and $Z\boldsymbol{x} = \sum_{j \in [s]} x_j \boldsymbol{z}_j \neq 0$. Otherwise, $Z$ would have zero rank (its null space would be of full dimension) and $\mathcal{C} = \{\boldsymbol{0}\}$ (our statement would then be trivially satisfied). Hence, we have $Z\boldsymbol{x} \neq 0$. Consider the value $N = 2 \cdot \max_{\mathcal{J} \subseteq [s]} \left\|\sum_{j \in \mathcal{J}} x_j \boldsymbol{z}_j\right\| (> 0)$ and the vector $\boldsymbol{w}$ with

$$\boldsymbol{w} = Z\boldsymbol{x}/N.$$

We have that for any $i \in [t]$ and $j \in [s]$ with $i, j$ such that $\boldsymbol{h}_i \cdot \boldsymbol{z}_j > 0$ it holds

$$\frac{x_j}{N} \cdot \boldsymbol{h}_i \cdot \boldsymbol{z}_j \geq \frac{x_j}{N} \cdot \theta_1 \,,$$

and therefore, for any $i \in [t]$ such that there exists $j \in [s]$ with $\boldsymbol{h}_i \cdot \boldsymbol{z}_j > 0$, $\boldsymbol{h}_i \cdot \boldsymbol{w} \geq \theta_1/N$ (since $x_j \geq 1$). Consider the following quantity

$$\theta_2 := \theta_1/N \,.$$

Note that again, $\theta_2 > 0$ and only depends on the geometry of $\mathcal{C}$. We demand that $\theta \leq \theta_2$. Then, $\boldsymbol{w} \in \mathcal{C}$, $\|\boldsymbol{w}\| \leq 1/2$ (and the norm bound also holds for any part of $\boldsymbol{w}$ of the form $\sum_{j \in \mathcal{J}} w_j \boldsymbol{z}_j$, for $\mathcal{J} \subseteq [s]$) and $\boldsymbol{h}_i \cdot \boldsymbol{w} \geq \theta$ for any interesting $\boldsymbol{h}_i$ (i.e., for any $\boldsymbol{h}_i$ that is not orthogonal to every point in $\mathcal{C}$).

**Step 2.** Consider, now, any $\boldsymbol{u} \in \mathcal{C}$. We will find $\boldsymbol{u}' \in \mathcal{C}$ with the desired properties. We have that

$$\boldsymbol{u} = \sum_{j \in [s]} u_j \boldsymbol{z}_j \,, \text{ where } u_j \geq 0 \,.$$

Consider the set $\mathcal{I} \subseteq [t]$ as follows

$$\mathcal{I} \subseteq [t] : \boldsymbol{h}_i \cdot \boldsymbol{u} < \theta \text{ for } i \in \mathcal{I} \, \& \\ \boldsymbol{h}_i \cdot \boldsymbol{u} \geq \theta \text{ for } i \notin \mathcal{I} \,.$$

The set $\mathcal{I}$ includes the columns on $H$ that correspond to halfspaces with boundaries to which $\boldsymbol{u}$ is close. Moreover, we define for any $\mathcal{I} \subseteq [t]$ the set $\mathcal{J}_\mathcal{I}$ as follows

$$\mathcal{J}_\mathcal{I} = \{ j \in [s] : \boldsymbol{h}_i \cdot \boldsymbol{z}_j = 0 \text{ for all } i \in \mathcal{I} \} \,.$$

The set $\mathcal{J}_\mathcal{I}$ corresponds to the generating vectors that lie within the nullspace of the set of columns of $\mathcal{H}$ corresponding to $\mathcal{I}$. The vectors corresponding to $\mathcal{J}_\mathcal{I}$ control the part of $\boldsymbol{u}$ that is parallel to the boundaries to which $\boldsymbol{u}$ is close.

We use the notation $\boldsymbol{u}_\mathcal{I}$ (resp. $\boldsymbol{w}_\mathcal{I}$) to refer to the vector $\boldsymbol{u}_\mathcal{I} = \sum_{j \in \mathcal{J}_\mathcal{I}} u_j \boldsymbol{z}_j$, i.e., the part of $\boldsymbol{u}$ that corresponds to nearby boundaries. We let

$$\boldsymbol{u}' = \boldsymbol{u} - c(\boldsymbol{u}_\mathcal{I} - \boldsymbol{w}_\mathcal{I}) \,,$$

for some $c \in [0, 1]$ to be disclosed. We show that $\boldsymbol{u}'$ (for appropriate $c$ and possibly some additional bounds on $\theta$) has all the desired properties.

1. Consider any $i \in \mathcal{I}$. We have that $\boldsymbol{h}_i \cdot \boldsymbol{u}_\mathcal{I} = 0 = \boldsymbol{h}_i \cdot \boldsymbol{w}_\mathcal{I}$, due to the definition of $\mathcal{J}_\mathcal{I}$. Therefore $\boldsymbol{h}_i \cdot \boldsymbol{u} = \boldsymbol{h}_i \cdot \boldsymbol{u}'$.

2. For $i \in [t] \setminus \mathcal{I}$, we have that: $\boldsymbol{h}_i \cdot \boldsymbol{u}' = \boldsymbol{h}_i \cdot \boldsymbol{u} - c \boldsymbol{h}_i \cdot \boldsymbol{u}_\mathcal{I} + c \boldsymbol{h}_i \cdot \boldsymbol{w}_\mathcal{I}$, where if $\boldsymbol{h}_i \cdot \boldsymbol{z}_j = 0$ for every $j \in \mathcal{J}_\mathcal{I}$, we have $\boldsymbol{h}_i \cdot \boldsymbol{u}' = \boldsymbol{h}_i \cdot \boldsymbol{u} \geq \theta$, since $i \notin \mathcal{I}$. Otherwise, $\boldsymbol{h}_i \cdot \boldsymbol{w}_\mathcal{I} \geq \theta$ (due to **Step 1**) and we get

$$\boldsymbol{h}_i \cdot \boldsymbol{u}' \geq (1 - c) \boldsymbol{h}_i \cdot \boldsymbol{u} + c \boldsymbol{h}_i \cdot \boldsymbol{w}_\mathcal{I} \geq (1 - c)\theta + c\theta = \theta \,.$$

3. We have that $\boldsymbol{u}' = \sum_{j \in [s]} u_j' \boldsymbol{z}_j$, where $u_j' \geq 0$ since $c \leq 1$. Therefore $\boldsymbol{u}' \in \mathcal{C}$. It remains to show that for some selection of $c \in [0, 1]$, we achieve $\|\boldsymbol{u}'\| = 1$. Consider, temporarily the case $c = 1$. If we show that for this value of $c$, $\boldsymbol{u}'$ is within the unitary ball, then, since $\boldsymbol{u}$ is outside (and corresponds to $c = 0$), there must exist some $c \in [0, 1]$ (i.e., a point on the line segment connecting $\boldsymbol{u}$ and $\boldsymbol{u} - \boldsymbol{u}_\mathcal{I} + \boldsymbol{w}_\mathcal{I}$) for which $\|\boldsymbol{u}'\| = 1$.

   For $c = 1$ we have that

   $$\|\boldsymbol{u}'\| = \|\boldsymbol{u} - \boldsymbol{u}_\mathcal{I} + \boldsymbol{w}_\mathcal{I}\| \leq \|\boldsymbol{u} - \boldsymbol{u}_\mathcal{I}\| + \|\boldsymbol{w}_\mathcal{I}\| \,.$$

   Observe, now that, $\|\boldsymbol{w}_\mathcal{I}\| \leq 1/2$ (due to the definition of $N$ in **Step 1**) and that as $\theta$ decreases, $\|\boldsymbol{u} - \boldsymbol{u}_\mathcal{I}\|$ tends to become zero for fixed $\mathcal{I}$, independently from the selection of $\boldsymbol{u}$ given that $\boldsymbol{u}$ corresponds to $\mathcal{I}$ (we may minimize over finitely many possible $\mathcal{I}$). Therefore, for small enough $\theta > 0$, we have that $\|\boldsymbol{u}'\| \leq 1$ (for $c = 1$, which implies that $\|\boldsymbol{u}'\| = 1$ for some appropriate selection of $c \in [0, 1)$).

More specifically, we have that

$$\|\boldsymbol{u} - \boldsymbol{u}_{\mathcal{I}}\|^2 = \sum_{j,j' \notin \mathcal{J}_{\mathcal{I}}} u_j u_{j'} (\boldsymbol{z}_j \cdot \boldsymbol{z}_{j'}) \leq \left( \sum_{j \notin \mathcal{J}_{\mathcal{I}}} u_j \right)^2 .$$

For each $j \notin \mathcal{J}_{\mathcal{I}}$, there must exist some $i_j \in \mathcal{I}$ so that $\boldsymbol{h}_{i_j} \cdot \boldsymbol{z}_j > 0$ (otherwise, $j \in \mathcal{J}_{\mathcal{I}}$). Recall that due to the definition of $\theta_1$, we have $\boldsymbol{h}_{i_j} \cdot \boldsymbol{z}_j \geq \theta_1$. Moreover, we have

$$\boldsymbol{h}_{i_j} \cdot \boldsymbol{u} < \theta ,$$

since $i_j \in \mathcal{I}$. Hence, since $\boldsymbol{u} = \sum_{j' \in [s]} u_{j'} \boldsymbol{z}_{j'}$ and $\boldsymbol{h}_{i_j} \cdot \boldsymbol{z}_{j'} \geq 0$ by the fact that $\boldsymbol{z}_{j'} \in \mathcal{C}$, we get that

$$u_j (\boldsymbol{h}_{i_j} \cdot \boldsymbol{z}_j) < \theta , \text{ or } u_j < \theta/\theta_1 , \text{ for any } j \notin \mathcal{J}_{\mathcal{I}} .$$

Therefore $(\sum_{j \notin \mathcal{J}_{\mathcal{I}}} u_j) \leq \theta \cdot s / \theta_1$. We demand that

$$\theta \leq \frac{\theta_1}{2 \cdot s} \text{ (and } \theta \leq \theta_2 \text{ from \textbf{Step 1}) },$$

which concludes our proof. $\qquad\square$

# F   The Proof of Theorem 5 (Sparsifying the Parameter Space)

We restate the result for convenience.

**Theorem.** *Under assumptions* (1), (2), (3), (4) *and* (5), *there exists some value* $\theta = \theta(\mathcal{A}, \boldsymbol{T}) > 0$ *depending on the geometric properties of $\mathcal{A}$ and $\boldsymbol{T}$, such that for any $\epsilon \in (0, 1)$, there exists a set* $\mathcal{B} \subseteq \varrho\text{-Cone}\mathcal{A}$ *with* $|\mathcal{B}| \leq \left( \widetilde{O}\left( \frac{\sqrt{\Lambda} \cdot \varrho}{\epsilon} + \frac{\sqrt{\Lambda}}{\epsilon \cdot \theta} \right) + O\left( \frac{\sqrt{\Lambda}}{\epsilon \cdot \theta} \cdot \log(B) \right) \right)^k$ *such that, for any $\boldsymbol{a} \in \mathcal{A}$, it holds that*

$$d_{TV}(\mathcal{E}_{\boldsymbol{T}}(\boldsymbol{a}), \mathcal{E}_{\boldsymbol{T}}(\boldsymbol{b})) \leq \epsilon , \text{ for some } \boldsymbol{b} \in \mathcal{B} .$$

*Proof.* According to Theorem 6, if we consider $\mathcal{A}' = \varrho\text{-Cone}\mathcal{A} \cap \{\boldsymbol{a} : \|\boldsymbol{a}\| \leq r_{\text{crit}}\}$ for some sufficiently large $r_{\text{crit}} \leq (\varrho + \frac{1}{\theta}) \cdot \ln(1/\epsilon) + \frac{1}{2\theta} \cdot \ln(B) + O(\varrho + \frac{1}{\theta})$, then the exponential family $\mathcal{E}_{\boldsymbol{T}}(\mathcal{A}')$ $\epsilon$-covers the family $\mathcal{E}_{\boldsymbol{T}}(\mathcal{A})$. However, $\mathcal{E}_{\boldsymbol{T}}(\mathcal{A}')$ might contain infinitely many elements.

In order to sparsify $\mathcal{E}_{\boldsymbol{T}}(\mathcal{A}')$, we make use of Lemma 14 (applied to $\text{Conv}\mathcal{A}'$), combined with assumption (5). In particular, we get that for any $\boldsymbol{a}, \boldsymbol{b} \in \mathcal{A}' (\subseteq \text{Conv}\mathcal{A}')$ it holds

$$d_{TV}(\mathcal{E}_{\boldsymbol{T}}(\boldsymbol{a}), \mathcal{E}_{\boldsymbol{T}}(\boldsymbol{b})) \leq \|\boldsymbol{a} - \boldsymbol{b}\| \cdot \sqrt{\Lambda/2} ,$$

by making use of Pinsker's inequality. Therefore, the problem of sparsely covering $\mathcal{E}_{\boldsymbol{T}}(\mathcal{A}')$ in total variation distance is reduced to sparsely covering $\mathcal{A}'$ in Euclidean distance.

The cover in Euclidean distance is given by Proposition 33 and we get that $\mathcal{E}_{\boldsymbol{T}}(\mathcal{A}')$ is $\epsilon$-covered by $\mathcal{E}_{\boldsymbol{T}}(\mathcal{B})$ for some $\mathcal{B} \subseteq \mathcal{A}'$ where $|\mathcal{B}| \leq (1 + r_{\text{crit}} \cdot \sqrt{2\Lambda}/\epsilon)^k$. $\qquad\square$

**Proposition 33.** *For any $\epsilon > 0$, any $k \in \mathbb{N}$, any $r > 0$ and any subset $\mathcal{B}$ of $\mathbb{R}^k$ with $\sup_{\boldsymbol{b} \in \mathcal{B}} \|\boldsymbol{b}\| \leq r$, there exists an $\epsilon$-cover of $\mathcal{B}$ with respect to the Euclidean distance with size at most $(1 + 2r/\epsilon)^k$.*

*Proof.* We use a simple greedy algorithm: We create the cover incrementally by adding in each step an arbitrary point $\boldsymbol{b}$ of the remaining set (initially, the remaining set is $\mathcal{B}$) and remove from the remaining set the ball $\mathbb{B}_\epsilon[\boldsymbol{b}]$.

Let $(\boldsymbol{b}_i)_{i \in [N]}$ be the points of the cover. Note that it might be possible that $N = \infty$. However, as we will show, this is not the case.

First, note that $\|\boldsymbol{b}_i - \boldsymbol{b}_j\| > \epsilon$, whenever $i \neq j$, since (assuming wlog $j > i$) $\boldsymbol{b}_j \notin \mathbb{B}_\epsilon[\boldsymbol{b}_i]$. Therefore, $\mathbb{B}_{\epsilon/2}[\boldsymbol{b}_i] \cap \mathbb{B}_{\epsilon/2}[\boldsymbol{b}_j] = \emptyset$ whenever $i \neq j$. Note that $N$ must be finite.

Let $\text{Vol}(\cdot)$ denote the volume measure that inputs a set and outputs its volume. Since $\text{Vol}(\cdot)$ is a measure and $(\mathbb{B}_{\epsilon/2}[\boldsymbol{b}_i])_i$ are disjoint, we have

$$\text{Vol}\left( \bigcup_{i \in [N]} \mathbb{B}_{\epsilon/2}[\boldsymbol{b}_i] \right) = \sum_{i \in [N]} \text{Vol}\left( \mathbb{B}_{\epsilon/2}[\boldsymbol{b}_i] \right) = N \cdot \left( \frac{\epsilon}{2} \right)^k \cdot \text{Vol}\left( \mathbb{B}_1[\boldsymbol{0}_k] \right) .$$

Also, $\cup_{i \in [N]} \mathbb{B}_{\epsilon/2}[\boldsymbol{b}_i]$ has to be a subset of $\mathbb{B}_{r+\epsilon/2}[\boldsymbol{0}_k]$, since $\mathcal{B}$ is a subset of $\mathbb{B}_r[\boldsymbol{0}_k]$. Therefore

$$\text{Vol}\left( \bigcup_{i \in [N]} \mathbb{B}_{\epsilon/2}[\boldsymbol{b}_i] \right) \le \left( r + \frac{\epsilon}{2} \right)^k \cdot \text{Vol}(\mathbb{B}_1[\boldsymbol{0}_k]) \,.$$

We get that $N \le (1 + 2r/\epsilon)^k$. $\qquad\square$

# G    Technical Lemmata for the Proof of Theorem 3

This lemma shows that for any $\boldsymbol{a}$ in the $\varrho$-Cone$\mathcal{A}$, the partition function is bounded under the unimodality and the bounded central fourth moment conditions.

**Lemma 34** (Bounded Partition Function). *Consider parameter space $\mathcal{A}$ and sufficient statistics vector $\boldsymbol{T}$. Under assumptions* (2) *and* (4)*, we have that*

$$Z_{\boldsymbol{T}}(\boldsymbol{a}) := \sum_{x \in \mathbb{Z}} \exp(-\boldsymbol{a} \cdot \boldsymbol{T}(x)) \le \exp(-\boldsymbol{a} \cdot \boldsymbol{T}(M_{\boldsymbol{a}})) \cdot O(B^{1/4}) \,,$$

*for any $\boldsymbol{a} \in \varrho$-Cone$\mathcal{A}$.*

*Proof.* From assumption (4) and the fact that $(\mathbf{E}[|W - \mathbf{E}[W]|^2])^2 \le \mathbf{E}[|W - \mathbf{E}[W]|^4]$, we get the following inequality

$$\mathbf{Var}_{\boldsymbol{a}}(W) \le O(\sqrt{B}) \,.$$

We also know that $\mathbf{E}_{\boldsymbol{a}}[|W - M_{\boldsymbol{a}}|^2] \le 4\,\mathbf{Var}_{\boldsymbol{a}}(W)$, due to unimodality of the random variable $W$ (which implies that $|\,\mathbf{E}_{\boldsymbol{a}}[W] - M_{\boldsymbol{a}}| \le \sqrt{3\,\mathbf{Var}_{\boldsymbol{a}}(W)}$ as shown by Johnson and Rogers [1951]).

Therefore, $\mathbf{E}[|W - M_{\boldsymbol{a}}|^2] \le O(\sqrt{B})$. Consider the random variable $U := |W - M_{\boldsymbol{a}}|$. We have that $\mathbf{Var}_{\boldsymbol{a}}(U) \ge 0$ and hence $\mathbf{E}_{\boldsymbol{a}}[U^2] \ge (\mathbf{E}_{\boldsymbol{a}}[U])^2$. Therefore

$$\mathbf{E}_{\boldsymbol{a}}[|W - M_{\boldsymbol{a}}|] \le O(B^{1/4}) \,.$$

We have

$$\begin{aligned}
\mathbf{E}_{\boldsymbol{a}}[|W - M_{\boldsymbol{a}}|] &= \frac{\sum_{x \in \mathbb{Z}} |x - M_{\boldsymbol{a}}| \cdot \exp(-\boldsymbol{a} \cdot \boldsymbol{T}(x))}{\sum_{x \in \mathbb{Z}} \exp(-\boldsymbol{a} \cdot \boldsymbol{T}(x))} \\
&= \frac{\sum_{x=0}^{\infty} x(e^{-\boldsymbol{a} \cdot \boldsymbol{T}(x+M_{\boldsymbol{a}})} + e^{-\boldsymbol{a} \cdot \boldsymbol{T}(M_{\boldsymbol{a}}-x)})}{\sum_{x \in \mathbb{Z}} e^{-\boldsymbol{a} \cdot \boldsymbol{T}(x)}} \\
&= \frac{\sum_{y=1}^{\infty} \sum_{x=y}^{\infty} (e^{-\boldsymbol{a} \cdot \boldsymbol{T}(x+M_{\boldsymbol{a}})} + e^{-\boldsymbol{a} \cdot \boldsymbol{T}(M_{\boldsymbol{a}}-x)})}{e^{-\boldsymbol{a} \cdot \boldsymbol{T}(M_{\boldsymbol{a}})} + \sum_{x=1}^{\infty} (e^{-\boldsymbol{a} \cdot \boldsymbol{T}(x+M_{\boldsymbol{a}})} + e^{-\boldsymbol{a} \cdot \boldsymbol{T}(M_{\boldsymbol{a}}-x)})} \\
&= \sum_{y=1}^{\infty} \left( 1 - \frac{e^{-\boldsymbol{a} \cdot \boldsymbol{T}(M_{\boldsymbol{a}})} + \sum_{x=1}^{y-1} (e^{-\boldsymbol{a} \cdot \boldsymbol{T}(x+M_{\boldsymbol{a}})} + e^{-\boldsymbol{a} \cdot \boldsymbol{T}(M_{\boldsymbol{a}}-x)})}{e^{-\boldsymbol{a} \cdot \boldsymbol{T}(M_{\boldsymbol{a}})} + \sum_{x=1}^{\infty} (e^{-\boldsymbol{a} \cdot \boldsymbol{T}(x+M_{\boldsymbol{a}})} + e^{-\boldsymbol{a} \cdot \boldsymbol{T}(M_{\boldsymbol{a}}-x)})} \right) \\
&= \sum_{y=1}^{\infty} \left( 1 - \frac{1 + \sum_{x=1}^{y-1} (e^{-\boldsymbol{a} \cdot (\boldsymbol{T}(x+M_{\boldsymbol{a}})-\boldsymbol{T}(M_{\boldsymbol{a}}))} + e^{-\boldsymbol{a} \cdot (\boldsymbol{T}(M_{\boldsymbol{a}}-x)-\boldsymbol{T}(M_{\boldsymbol{a}}))})}{e^{\boldsymbol{a} \cdot \boldsymbol{T}(M_{\boldsymbol{a}})} \cdot Z_{\boldsymbol{T}}(\boldsymbol{a})} \right) \\
&\ge \sum_{y=1}^{t} \left( 1 - \frac{1 + \sum_{x=1}^{y-1} (e^{-\boldsymbol{a} \cdot (\boldsymbol{T}(x+M_{\boldsymbol{a}})-\boldsymbol{T}(M_{\boldsymbol{a}}))} + e^{-\boldsymbol{a} \cdot (\boldsymbol{T}(M_{\boldsymbol{a}}-x)-\boldsymbol{T}(M_{\boldsymbol{a}}))})}{e^{\boldsymbol{a} \cdot \boldsymbol{T}(M_{\boldsymbol{a}})} \cdot Z_{\boldsymbol{T}}(\boldsymbol{a})} \right) \\
&\ge \sum_{y=1}^{t} \left( 1 - \frac{1 + 2(y-1)}{e^{\boldsymbol{a} \cdot \boldsymbol{T}(M_{\boldsymbol{a}})} \cdot Z_{\boldsymbol{T}}(\boldsymbol{a})} \right) = t - \frac{t^2}{e^{\boldsymbol{a} \cdot \boldsymbol{T}(M_{\boldsymbol{a}})} \cdot Z_{\boldsymbol{T}}(\boldsymbol{a})} \,,
\end{aligned}$$

where $t \in \mathbb{N}$ is arbitrary and the last inequality follows from the fact that $M_{\boldsymbol{a}}$ is a mode (which implies that $\boldsymbol{a} \cdot (\boldsymbol{T}(x) - \boldsymbol{T}(M_{\boldsymbol{a}})) \ge 0$ for any $x \in \mathbb{Z}$).

We can pick $t = \frac{1}{2} \cdot \exp(\boldsymbol{a} \cdot \boldsymbol{T}(M_{\boldsymbol{a}})) \cdot Z_{\boldsymbol{T}}(\boldsymbol{a}) \pm O(1)$ in order to get the following bound

$$\exp(\boldsymbol{a} \cdot \boldsymbol{T}(M_{\boldsymbol{a}})) \cdot Z_{\boldsymbol{T}}(\boldsymbol{a}) \le 4\,\mathbf{E}_{\boldsymbol{a}}[|W - M_{\boldsymbol{a}}|] \pm O(1) \,,$$

which concludes the proof since, as we have shown, $\mathbf{E}_{\boldsymbol{a}}[|W - M_{\boldsymbol{a}}|] \le O(B^{1/4})$. $\qquad\square$

The next key lemma shows that, under unimodality and bounded fourth central moment, the mass of points that are sufficiently far from the modes of the distribution decays exponentially. Moreover, the centered moments of order at most 2 can be roughly controlled by points that lie only in a bounded interval around the mode.

**Lemma 35.** *Under assumptions (2) and (4), for any $\kappa > 0$, any $\eta > 0$ and any $s \in \{0, 1, 2\}$ there exists some $\ell = e^{\kappa/(3-\eta-s)} \cdot O(B^{\frac{5}{4\cdot(3-\eta-s)}})$ such that for any $\boldsymbol{a} \in \varrho\text{-Cone}\mathcal{A}$ and any mode $M_{\boldsymbol{a}}$ of the corresponding distribution we have*

1. $\mathbf{Pr}_{\boldsymbol{a}}[W = x] \leq \frac{e^{-\kappa\cdot\max\left\{1, \frac{\|\boldsymbol{a}\|}{\varrho}\right\}}}{|x - M_{\boldsymbol{a}}|^{1+\eta+s}} \cdot \mathbf{Pr}_{\boldsymbol{a}}[W = M_{\boldsymbol{a}}]$, *for any $x \in \mathbb{Z}$ with $|x - M_{\boldsymbol{a}}| \geq \ell$.*

2. *If $Q_\ell = \mathbf{1}\{|W - M_{\boldsymbol{a}}| \leq \ell\}$ then*

$$\mathbf{E}_{\boldsymbol{a}}[|W - M_{\boldsymbol{a}}|^s] \leq \mathbf{E}_{\boldsymbol{a}}[|W - M_{\boldsymbol{a}}|^s \cdot Q_\ell] + e^{-\kappa\cdot\max\left\{1, \frac{\|\boldsymbol{a}\|}{\varrho}\right\}} \cdot O(1/\eta)\,.$$

*In particular, for $s = 0$, we use the convention $\mathbf{E}[W^0] = \mathbf{Pr}[W \neq 0]$, for any random variable $W$.*

*Proof.* From assumption (4), we get the following inequality for any $\boldsymbol{a} \in \varrho\text{-Cone}\mathcal{A}$

$$\mathbf{Var}_{\boldsymbol{a}}(W) \leq O(\sqrt{B})\,.$$

We also know that $\mathbf{E}_{\boldsymbol{a}}[|W - M_{\boldsymbol{a}}|^2] \leq 4\,\mathbf{Var}_{\boldsymbol{a}}(W)$, due to unimodality of the random variable $W$ (which implies that $|\mathbf{E}_{\boldsymbol{a}}[W] - M_{\boldsymbol{a}}| \leq \sqrt{3\,\mathbf{Var}_{\boldsymbol{a}}(W)}$ as shown by Johnson and Rogers [1951]). Therefore $\mathbf{E}_{\boldsymbol{a}}[|W - M_{\boldsymbol{a}}|^2] \leq O(\sqrt{B})$ and similarly $\mathbf{E}_{\boldsymbol{a}}[|W - M_{\boldsymbol{a}}|^4] \leq O(B)$.

**Proof of Part (1).** For any parameter vector $\boldsymbol{a} \in \varrho\text{-Cone}\mathcal{A}$ (and some fixed corresponding mode $M_{\boldsymbol{a}}$), we have that $\mathbf{E}_{\boldsymbol{a}}[|W - M_{\boldsymbol{a}}|^4] = O(B)$ (since $\mathbf{E}_{\boldsymbol{a}}[|W - \mathbf{E}_{\boldsymbol{a}}[W]|^4] = O(B)$ and $\mathbf{Var}_{\boldsymbol{a}}(W) = O(\sqrt{B})$).

Let $\boldsymbol{b} \in \varrho\text{-Cone}\mathcal{A}$. Suppose that for some $x \in \mathbb{Z}$ with $x \neq M_{\boldsymbol{b}}$, we have that

$$\boldsymbol{b} \cdot (\boldsymbol{T}(x) - \boldsymbol{T}(M_{\boldsymbol{b}})) < (1 + \eta + s) \cdot \ln(|x - M_{\boldsymbol{b}}|) + \kappa\,.$$

Then, we have that

$$\mathbf{E}_{\boldsymbol{b}}[|W - M_{\boldsymbol{b}}|^4] \cdot e^{\boldsymbol{b}\cdot\boldsymbol{T}(M_{\boldsymbol{b}})} \cdot Z_{\boldsymbol{T}}(\boldsymbol{b}) > |x - M_{\boldsymbol{b}}|^4 \cdot e^{-(1+\eta+s)\ln(|x-M_{\boldsymbol{b}}|)-\kappa} = e^{-\kappa} \cdot |x - M_{\boldsymbol{b}}|^{3-\eta-s}\,.$$

Since, additionally, $\mathbf{E}_{\boldsymbol{b}}[|W - M_{\boldsymbol{b}}|^4] = O(B)$, and using Lemma 34, it must be that $|x - M_{\boldsymbol{b}}| < \ell$, for some $\ell$ with $\ell \leq e^{\frac{\kappa}{3-\eta-s}} \cdot O\left(B^{\frac{5}{4\cdot(3-\eta-s)}}\right)$.

For $x \in \mathbb{Z}$ with $|x - M_{\boldsymbol{b}}| \geq \ell$ we therefore get that

$$\boldsymbol{b} \cdot \boldsymbol{T}(x) \geq \ \boldsymbol{b} \cdot \boldsymbol{T}(M_{\boldsymbol{b}}) + (1 + \eta + s)\ln(|x - M_{\boldsymbol{b}}|) + \kappa\,. \tag{10}$$

Consider now any $\boldsymbol{a} \in \varrho\text{-Cone}\mathcal{A}$ with $\|\boldsymbol{a}\| \geq \varrho$. Then, there exists some $\boldsymbol{b} \in \varrho\text{-Cone}\mathcal{A}$ with $\|\boldsymbol{b}\| = \varrho$ so that $\boldsymbol{b} = \varrho\boldsymbol{a}/\|\boldsymbol{a}\|$. We multiply both sides of Equation (10) with $\|\boldsymbol{a}\|/\varrho \geq 1$ and use the fact that rescaling the parameter vector does not change the set of modes to get

$$\boldsymbol{a} \cdot \boldsymbol{T}(x) \geq \boldsymbol{a} \cdot \boldsymbol{T}(M_{\boldsymbol{a}}) + (1 + \eta + s)\ln(|x - M_{\boldsymbol{b}}|) + \kappa\|\boldsymbol{a}\|/\varrho, \quad \text{or}$$

$$\mathbf{Pr}_{\boldsymbol{a}}[W = x] \leq \exp(-\kappa\|\boldsymbol{a}\|/\varrho) \cdot \frac{1}{|x - M_{\boldsymbol{b}}|^{(1+\eta+s)}} \cdot \mathbf{Pr}_{\boldsymbol{a}}[W = M_{\boldsymbol{a}}]\,,$$

for any $x \in \mathbb{Z}$ with $|x - M_{\boldsymbol{a}}| \geq \ell$. Note that in the above we did not multiply $(1 + \eta + s)\ln(|x - M_{\boldsymbol{b}}|)$ with $\|\boldsymbol{a}\|/\varrho \geq 1$ since it only helps the inequality. Since $M_{\boldsymbol{a}} = M_{\boldsymbol{b}}$ due to the choice of $\boldsymbol{b}$, we accomplish our goal for the case $\|\boldsymbol{a}\| \geq \varrho$.

On the other side, if $\|\boldsymbol{a}\| < \varrho$, then $\kappa \geq \kappa \cdot \|\boldsymbol{a}\|/\varrho$ and we may pick $\boldsymbol{b} = \boldsymbol{a}$, concluding the proof of the first part of the Lemma.

**Proof of Part (2).** Let us set $s \in \{0, 1, 2\}$. We have that

$$\mathbf{E}_{\boldsymbol{a}}[|W - M_{\boldsymbol{a}}|^s] = \frac{\sum_{x \in \mathbb{Z}} |x - M_{\boldsymbol{a}}|^s \cdot \exp(-\boldsymbol{a} \cdot (\boldsymbol{T}(x) - \boldsymbol{T}(M_{\boldsymbol{a}})))}{\exp(\boldsymbol{a} \cdot \boldsymbol{T}(M_{\boldsymbol{a}})) \cdot Z_{\boldsymbol{T}}(\boldsymbol{a})}$$

$$= \mathbf{E}_{\boldsymbol{a}}[|W - M_{\boldsymbol{a}}|^s \cdot Q_\ell] + \sum_{x:|x-M_{\boldsymbol{a}}|>\ell} \frac{|x - M_{\boldsymbol{a}}|^s \cdot \exp(-\boldsymbol{a} \cdot (\boldsymbol{T}(x) - \boldsymbol{T}(M_{\boldsymbol{a}})))}{\exp(\boldsymbol{a} \cdot \boldsymbol{T}(M_{\boldsymbol{a}})) \cdot Z_{\boldsymbol{T}}(\boldsymbol{a})}$$

$$\leq \mathbf{E}_{\boldsymbol{a}}[|W - M_{\boldsymbol{a}}|^s \cdot Q_\ell] + \sum_{x:|x-M_{\boldsymbol{a}}|>\ell} |x - M_{\boldsymbol{a}}|^s \cdot \exp(-\boldsymbol{a} \cdot (\boldsymbol{T}(x) - \boldsymbol{T}(M_{\boldsymbol{a}})))$$

$$\leq \mathbf{E}_{\boldsymbol{a}}[|W - M_{\boldsymbol{a}}|^2 \cdot Q_\ell] + e^{-\kappa \cdot \max\{\|\boldsymbol{a}\|/\varrho, 1\}} \cdot 2\zeta(1 + \eta)$$

$$\leq \mathbf{E}_{\boldsymbol{a}}[|W - M_{\boldsymbol{a}}|^2 \cdot Q_\ell] + e^{-\kappa \cdot \max\{\|\boldsymbol{a}\|/\varrho, 1\}} \cdot O(1/\eta),$$

where the first inequality follows from the fact that $\exp(\boldsymbol{a} \cdot \boldsymbol{T}(M_{\boldsymbol{a}})) \cdot Z_{\boldsymbol{T}}(\boldsymbol{a}) = \sum_{x \in \mathbb{Z}} \exp(-\boldsymbol{a} \cdot (\boldsymbol{T}(x) - \boldsymbol{T}(M_{\boldsymbol{a}}))) \geq \exp(-\boldsymbol{a} \cdot (\boldsymbol{T}(M_{\boldsymbol{a}}) - \boldsymbol{T}(M_{\boldsymbol{a}}))) = 1$. The second inequality follows by applying **Part (1)** and noting that $\sum_{x:|x-M_{\boldsymbol{a}}|>\ell} \frac{1}{|x-M_{\boldsymbol{a}}|^{1+\eta}} \leq 2\zeta(1 + \eta)$, where $\zeta(\cdot)$ denotes the Riemann zeta function and $\zeta(1 + \eta) = \Theta(1/\eta)$ as $\eta \to 0$. $\qquad\square$

We next show that the expectation and the variance for parameters inside $\varrho$-Cone$\mathcal{A}$ are continuous functions with respect to the parameter vectors.

**Lemma 36.** *Under assumptions* (2) *and* (4)*, the expectation* $\mathbf{E}_{\boldsymbol{a}}[W]$ *and variance* $\mathbf{Var}_{\boldsymbol{a}}(W)$ *are continuous functions of* $\boldsymbol{a}$ *on* $\varrho$-Cone$\mathcal{A}$.

*Proof.* We will prove that sums of the form $\sum_{x \in \mathbb{Z}} \exp(-\boldsymbol{a} \cdot \boldsymbol{T}(x))$ and $\sum_{x \in \mathbb{Z}} x^s \cdot \exp(-\boldsymbol{a} \cdot \boldsymbol{T})$, where $s = 1, 2$, are continuous functions of $\boldsymbol{a}$ on $\mathcal{A}$. Then, $\mathbf{E}_{\boldsymbol{a}}[W]$ and $\mathbf{Var}_{\boldsymbol{a}}(W)$ have to be continuous.

We proceed with the proof for $S := \sum_{x \in \mathbb{Z}} x^2 \cdot \exp(-\boldsymbol{a} \cdot \boldsymbol{T}(x))$, since the other cases can be proven similarly.

Fix some $\boldsymbol{a} \in \mathcal{A}$, some $\epsilon > 0$ and consider any $\delta\boldsymbol{a} \in \mathbb{R}^k$ so that $\boldsymbol{a}' := \boldsymbol{a} + \delta\boldsymbol{a} \in \mathcal{A}$ and $\|\delta\boldsymbol{a}\| \leq \delta$, where $\delta > 0$ to be decided (possibly dependent on $\epsilon$ and $\boldsymbol{a}$). We may apply Lemma 35 to $\mathcal{E}_{\boldsymbol{T}}(\mathcal{A})$, with $s = 2$, $\eta = 1/2$ and $\kappa \in (0, \infty)$ to be decided. Therefore, we get some $\ell = \ell(\kappa)$ such that for any $\boldsymbol{b} \in \mathcal{A}'$ and any $x \in \mathbb{Z}$ with $|x - M_{\boldsymbol{b}}| > \ell$ we have

$$\exp(-\boldsymbol{b} \cdot \boldsymbol{T}(x)) \leq e^{-\kappa \|\boldsymbol{b}\|/\varrho} \cdot \frac{1}{|x - M_{\boldsymbol{b}}|^{3.5}} \cdot \exp(-\boldsymbol{b} \cdot \boldsymbol{T}(M_{\boldsymbol{b}})).$$

Hence we have that

$$\sum_{x:|x-M_{\boldsymbol{b}}|>\ell} x^2 \cdot \exp(-\boldsymbol{b} \cdot \boldsymbol{T}(x)) \leq e^{-\kappa \|\boldsymbol{b}\|/\varrho} \cdot \frac{x^2}{|x - M_{\boldsymbol{b}}|^{3.5}} \cdot \exp(-\boldsymbol{b} \cdot \boldsymbol{T}(M_{\boldsymbol{b}}))$$

$$= e^{-\kappa \|\boldsymbol{b}\|/\varrho} \cdot \exp(-\boldsymbol{b} \cdot \boldsymbol{T}(M_{\boldsymbol{b}})) \cdot \sum_{y:|y|>\ell} \frac{|y + M_{\boldsymbol{b}}|^2}{|y^4|}.$$

We have that $\|\boldsymbol{a}\|, \|\boldsymbol{a}'\| > 0$, since otherwise $\mathcal{E}_{\boldsymbol{T}}(\mathcal{A})$ would not be well defined. Let $f(\boldsymbol{b}) = \exp(-\boldsymbol{b} \cdot \boldsymbol{T}(M_{\boldsymbol{b}})) \cdot \sum_{y:|y|>\ell} \frac{|y+M_{\boldsymbol{b}}|^2}{|y^4|}$. Therefore, we can pick $\kappa$ as a function of the quantities $1/\epsilon$, $\max\{\|\boldsymbol{a}\|^{-1}, \|\boldsymbol{a}'\|^{-1}\}$, $\varrho$ and $\max\{f(\boldsymbol{a}), f(\boldsymbol{a}')\}$, which are all finite, for any $\boldsymbol{a}$ and $\boldsymbol{a}'$ as hypothesised, so that $\sum_{x:|x-M_{\boldsymbol{b}}|>\ell} x^2 \cdot \exp(-\boldsymbol{b} \cdot \boldsymbol{T}(x)) \leq \epsilon/4$. Hence, if we consider $N = \{x \in \mathbb{Z} : |x - M_{\boldsymbol{a}}| \leq \ell$ or $|x - M_{\boldsymbol{a}'}| \leq \ell\}$, we have that

$$\sum_{x \in \mathbb{Z}} x^2 \cdot \exp(-\boldsymbol{a} \cdot \boldsymbol{T}(x)) = \sum_{x \in N} x^2 \cdot \exp(-\boldsymbol{a} \cdot \boldsymbol{T}(x)) \pm \epsilon/4,$$

and similarly for $\sum_{x \in \mathbb{Z}} x^2 \cdot \exp(-\boldsymbol{a}' \cdot \boldsymbol{T}(x))$. Therefore

$$\left| \sum_{x \in \mathbb{Z}} x^2 \cdot e^{-\boldsymbol{a} \cdot \boldsymbol{T}(x)} - \sum_{x \in \mathbb{Z}} x^2 \cdot e^{-\boldsymbol{a}' \cdot \boldsymbol{T}(x)} \right| \leq \sum_{x \in N} x^2 \cdot e^{-\boldsymbol{a} \cdot \boldsymbol{T}(x)} \cdot \left| 1 - e^{-\delta\boldsymbol{a} \cdot \boldsymbol{T}(x)} \right| + \epsilon/2.$$

We have that $N$ is finite and we can pick $\delta$ so that since $\|\delta\boldsymbol{a}\| \leq \delta$, the distance is at most $\epsilon$ (by upper bounding the sum of the right hand side with $|N|$ times the maximum term). $\qquad\square$

# H  Applications and Examples

## H.1  Examples of Distributions that we capture

Our assumptions for proper learning and covering of SIIERVs (see Assumption 2), capture a wide variety of families of discrete distributions, including discretized versions of many fundamental distributions, like Gaussian, Laplacian, etc. Although we focus on the case where the family $\mathcal{E}_{\boldsymbol{T}}(\mathcal{A})$ includes distributions supported on $\mathbb{Z}$, our results (and our assumptions) naturally extend to the cases where the support is some subset of $\mathbb{Z}$, like $\mathbb{N}_0$. In some cases, for example for distributions with finite support, our assumptions can be relaxed. In the following table, we represent examples of distributions with infinite support that our results capture.

Table 1: A collection of pairs $(\boldsymbol{T}, \mathcal{A})$ on which our results on learning and covering apply.

| Sufficient Statistic $\boldsymbol{T}$ | Support | Extended Parameter Space $\mathcal{A}_\varrho$ | Distribution |
|---|---|---|---|
| $T(x) = \ln(x)$ | $x \in \mathbb{N}$ | $[5 + \eta, \infty), \eta > 0$ | Zeta |
| $T(x) = x$ | $x \in \mathbb{N}_0$ | $[\eta, \infty), \eta > 0$ | Geometric |
| $T(x) = |x|$ | $x \in \mathbb{Z}$ | $[\eta, \infty), \eta > 0$ | Discrete Laplacian |
| $\boldsymbol{T}(x) = (x, x^2)$ | $x \in \mathbb{Z}$ | $\{\boldsymbol{a} : a_2 \geq |a_1|/L\} \setminus \mathbb{B}_\eta(\boldsymbol{0}), L > 0$ | Discrete Gaussian |
| $\boldsymbol{T}(x) = (|x|, x, x^2)$ | $x \in \mathbb{Z}$ | $\{\boldsymbol{a} : a_3 \geq |a_2|/L, a_1 \geq 0\} \setminus \mathbb{B}_\eta(\boldsymbol{0})$ | Gaussian-Laplacian Interpolation |

## H.2  Parametric Application: Proper Covers for PNBDs

In this section, we provide a parametric application that is captured by our techniques. We study the class of Poisson Negative Binomial random variables, i.e., sums of independent but not necessarily identically distributed Geometric random variables. We provide the following structural result.

**Theorem 37** (Proper Cover of Poisson Negative Binomials)**.** *Let $p_{\text{low}} \in (0, 1)$. For any $\epsilon > 0$, the family of Poisson Negative Binomial distributions (i.e., sums of Geometric random variables with success probability at least $p_{\text{low}}$) of order $n$ admits an $\epsilon$-proper cover of size $O(n^2/\text{poly}(p_{\text{low}})) + 2^{\text{poly}(1/\epsilon, 1/p_{\text{low}})}$. Moreover, for any PNBD $X$, there exists $Y$ so that $d_{TV}(X, Y) \leq \epsilon$ and (i) either $Y$ is a PNBD of order $O(\text{poly}(1/\epsilon, 1/p_{\text{low}}))$ among $2^{\text{poly}(1/\epsilon, 1/p_{\text{low}})}$ candidates (sparse form) or (ii) $Y$ is a Negative Binomial random variable of order $O(n) \cdot \text{poly}(1/p_{\text{low}})$ (dense form).*

The essentially important part of the proof is that we do not need to assume a variance lower bound (as we did in assumption (7)), since this is assured using the so-called Massage step of Daskalakis and Papadimitriou [2015]. The main tool of this trick is the Poisson approximation technique. Hence, in the proof of the above theorem, we solely focus on this massage procedure and we omit the details on how to handle the sparse and the dense case since they follow by adapting the techniques of our main results.

## H.3  The proof of Theorem 37

*Proof.* Let $\epsilon > 0$. Consider $X = \sum_{i \in [n]} X_i$, where $X_1, ..., X_n$ are independent and for and $i \in [n]$, $X_i \sim \text{Geo}(p_i)$ with $p_i \in [p_{\text{low}}, 1]$. Our proof involves three main parts. First, we perform a massage step to discard the terms with low variance from the sum. Then, we split two cases according to the number of terms that have survived. If the number of surviving terms is smaller than some (appropriately selected) $n'_{\text{crit}}$, then it is sufficient to approximate each term with accuracy $O(\epsilon/n'_{\text{crit}})$. If the number of surviving terms is higher than $n'_{\text{crit}}$, then we prove that $X$ is close to some discretized Gaussian and from that we find a Negative Binomial random variable that matches the first two moments of the sum and so is close to the Gaussian. The proximity follows by the triangle inequality of the TV distance. For the following, consider $\kappa > 1$ where $1/\kappa = O(\epsilon)$.

**Massage Step.** Consider the index set $I = \{i \in [n] : p_i > 1 - 1/\kappa\}$. For any $i \notin I$, we let $X_i' \sim \text{Geo}(p_i)$ and, using Lemma 15, we get that

$$d_{TV}\left(\sum_{i\in[n]} X_i, \sum_{i\in[n]} X_i'\right) \leq d_{TV}\left(\sum_{i\in I} X_i, \sum_{i\in I} X_i'\right).$$

For any $i \in I$, we either set $X_i' \sim \text{Geo}(p_i')$ with $p_i' = 1 - 1/\kappa$. or we set $X_i' = 0$ almost surely. Since $X_1, \ldots, X_n$ are independent geometric random variables, we can apply the following technical lemma:

**Lemma 38** (Corollary 2.5 of Barbour [1987])**.** *Consider $n$ independent random variables $X_1, ..., X_n$ that are geometrically distributed with success probabilities $p_1, ..., p_n$ respectively. Let $\lambda = \sum_{i\in[n]} \frac{1-p_i}{p_i}$. Then, it holds that*

$$d_{TV}\left(\sum_{i\in[n]} X_i, \text{Poi}(\lambda)\right) \leq \lambda^{-1}(1 - e^{-\lambda}) \cdot \sum_{i\in[n]} \left(\frac{1-p_i}{p_i}\right)^2.$$

Note that $\lambda^{-1}(1 - e^{-\lambda}) \leq \min\{1, \lambda^{-1}\}$. We make use of the above Poisson approximation lemma on the set of indices $I$ and get that the random variable $\sum_{i\in I} X_i$ can be approximated by a Poisson random variable with distribution $\text{Poi}\left(\sum_{i\in I} \mathbf{E}[X_i]\right)$. Specifically, we get that

$$d_{TV}\left(\sum_{i\in I} X_i, \text{Poi}\left(\sum_{i\in I} \mathbf{E}[X_i]\right)\right) \leq \frac{\sum_{i\in I} \mathbf{E}[X_i]^2}{\sum_{i\in I} \mathbf{E}[X_i]} \leq \max_{i\in I} \mathbf{E}[X_i],$$

where we applied Lemma 13 to the sequences of non-negative real numbers $(\mathbf{E}[X_i]^2)_{i\in I}$ and $(\mathbf{E}[X_i])_{i\in I}$. Hence, we have that

$$d_{TV}\left(\sum_{i\in I} X_i, \text{Poi}\left(\sum_{i\in I} \mathbf{E}[X_i]\right)\right) \leq \max_{i\in I}\left\{\frac{1-p_i}{p_i}\right\} = \frac{1}{\kappa - 1}. \tag{11}$$

We get the same upper bound for the total variation distance between $X_I' := \sum_{i\in I} X_i'$ and $\text{Poi}(\mathbf{E}[X_I'])$, similarly. We continue with the following claim.

**Claim 39** (Correct Rounding)**.** *We can partition the set $I \subseteq [n]$ into two sets $I_\star, I_0$ and set $X_i' \sim \text{Geo}(p_i')$ with $p_i' = 1 - 1/\kappa$, for any $i \in I_\star$ and $X_i' = 0$ almost surely for any $i \in I_0$ so that*

$$\left|\sum_{i\in I} \mathbf{E}[X_i] - \sum_{i\in I} \mathbf{E}[X_i']\right| \leq \frac{1}{\kappa - 1}.$$

*Proof.* If $i \in I_\star$, we have that $\mathbf{E}[X_i'] \leq 1/(\kappa - 1)$, whereas $\mathbf{E}[X_i'] = 0$ if $i \in I_0$. In the extreme case where $I_\star = I$, note that the expectation of the Geometric is non-increasing and so we have that $\mathbf{E}[X_i'] \geq \mathbf{E}[X_i]$ for any $i \in I$ and, so, we have that

$$\sum_{i\in I} \mathop{\mathbf{E}}_{X_i'\sim\text{Geo}(p_i')}[X_i'] \geq \sum_{i\in I} \mathbf{E}[X_i].$$

Hence, we can pick $I_\star$ to be any minimal subset of $I$ so that

$$\sum_{i\in I_\star} \mathop{\mathbf{E}}_{X_i'\sim\text{Geo}(p_i')}[X_i'] \geq \sum_{i\in I} \mathop{\mathbf{E}}_{X_i\sim\text{Geo}(p_i)}[X_i].$$

This choice of $I_\star$ yields

$$\left|\sum_{i\in I_\star} \mathbf{E}[X_i'] - \sum_{i\in I} \mathbf{E}[X_i]\right| \leq 1/(\kappa - 1),$$

and this provides Claim 39. □

By Lemma 15, we conclude that

$$d_{TV}\left(\sum_{i \in I} X_i, \sum_{i \in I} X_i'\right) \leq \frac{3}{\kappa - 1},$$

using Poisson approximation for $(X_i)_{i \in I}$ and $(X_i')_{i \in I}$ and combining the upper bound for the total variation distance of two Poisson distributions (see Lemma 16) with Claim 39.

Without loss of generality, we consider $n' = |I_\star|$, rearrange the terms and discard the trivial ones so that $X' = \sum_{i \in [n']} X_i'$, with $d_{TV}(X, X') \leq 3/(\kappa - 1)$ and $X_i' \sim \mathrm{Geo}(p_i')$, with $p_i' \in [p_{\mathrm{low}}, 1 - 1/\kappa]$.

The next steps are similar to the general case. Using Gaussian approximation, we compute $n'_{\mathrm{crit}}$. In particular, we can get that $n'_{\mathrm{crit}} = \mathrm{poly}(\kappa/p_{\mathrm{low}}) = \mathrm{poly}(1/(\epsilon \cdot p_{\mathrm{low}}))$. If $n' \leq n'_{\mathrm{crit}}$, the PNBD is close to a sparse form that is a sum of Geometric random variables consisting of at most $n'_{\mathrm{crit}}$ terms. In this case, it is sufficient to approximate each term $X_i$ separately using a random variable $Y_i \sim \mathrm{Geo}(q_i)$. Due to sub-additivity of the statistical distance, it suffices to control the TV distance between $X_i$ and $Y_i$ by $\epsilon/n'_{\mathrm{crit}}$. Then, it will hold that $d_{TV}(\sum_{i \in [n']} X_i, \sum_{i \in [n']} Y_i) \leq \epsilon$ for $n' \leq n'_{\mathrm{crit}}$. We have to discretize the interval $[p_{\mathrm{low}}, 1 - 1/\kappa]$ with appropriate accuracy in order to get the result. The discretization depends on the TV distance between two Geometric random variables that can be easily computed.

Otherwise, we first approximate it using a discretized Gaussian random variable and then match the expectation and the variance in order to find a Negative Binomial that is close to the input PNBD. This gives the bounds presented in the statement but we omit the details. □

## H.4 Verification of Assumptions

Although Assumption 2 might not be efficiently verifiiable for every selection of the sufficient statistics, assuming a simple given description of the sufficient statistics vector, analytic methods can potentially reduce the assumptions to restrictions on the space of parameters. In particular, we have the following.

- For conditions 2 and 3 (unimodality and localization of modes) we have already identified an algebraic condition in terms of an appropriate set of linear inequalities (see Appendix E.2).

- Condition 4 (bounded central moments) is linked to a lower bound on the minimum norm of the parameter space. For instance, if the sufficient statistics is a scalar logarithmic function (corresponding to Zeta distribution), the fourth moment is bounded when the parameter takes values bounded away above 5, according to the convergence of the zeta function (see Appendix H.1).

- For condition 5 (spectral bound on the covariance matrix), it is sufficient to show an upper bound on the expected value of the squared norm of the sufficient statistics vector, i.e., $\mathbf{E}[\|\boldsymbol{T}(W)\|^2]$. Such an upper bound may correspond to the exclusion of some parameter values when different coordinates of the sufficient statistics have different behavior in the limit $x \to \infty$. For example, if one coordinate is polynomial, while another one is logarithmic, we would like to ensure that when the parameter corresponding to the polynomial statistic is zero, the other parameter will be bounded away above a value that depends on the degree of the polynomial statistic, i.e., if $\boldsymbol{T}(x) = (x^r, \log x)$, then $(0, a_2) \in \mathcal{A}$ implies $a_2 \geq f(r)$, where $f$ is some appropriate (increasing) function.

- Finally, for condition 7 (variance lower bound), consider the simple example of the Geometric distributions, where the sufficient statistics is a scalar linear function over $\mathbb{N}$ (i.e., $\boldsymbol{T}(x) = x$). Then, the variance lower bound is equivalent to an upper bound on the parameter space (i.e., $a \in \mathcal{A}$ implies $a \leq a_{\max}$). We note, however, that the variance lower bound does not always imply that the parameter space is bounded. In particular, when a distribution has two or more subsequent modes, then, as the norm of the parameter increases to the limit, the variance remains bounded away above zero. Therefore, the variance lower bound may correspond to a different upper bound on the norm for each direction of the parameter space (since the parameter vector's direction defines the set of modes; see Appendix E.2).