# OpenReview forum: "Learning and Covering Sums of Independent Random Variables with Unbounded Support"
_NeurIPS.cc/2022/Conference — NeurIPS 2022 Accept_

### Official Review · Reviewer_HJNQ · 2022-07-11

**Rating:** 7
**Confidence:** 4
**Soundness:** 3 good
**Presentation:** 3 good
**Contribution:** 4 excellent

**Summary:**

This paper considers a specific problem in structured distribution learning, namely learning random variables that are expressed as sums of n integer-valued independent random variables (SIIRVs).  Previous work considered the setting of bounded support and bounded maximal support element. Interestingly, it was known that for unbounded support, the sample complexity includes dependence on n. Also, for bounded support of size 4, sample complexity includes dependence of maximal support element. This paper focuses on the infinite support case and finds assumptions on distributions such that the sample complexity is independent of n and the maximal support element. For the exponential family, under certain further assumptions, it shows that proper learning is possible.

In general, the learning algorithm is a hypothesis selection between two sets of candidate distributions, one that’s close when n is large (“dense form”) and the other that’s close when n is small (“sparse form”). The former is a discretized Gaussian with identical first and second moments and the latter is a cover of the family of distribution that each of the n random variables are drawn from. For proper learning, both the sparse and dense covers are mandated to also be in the specified exponential-SIIRVs. For the dense case in particular, the algorithm runs a tournament procedure with an estimated discretized Gaussian and a separate dense cover. Bulk of the later part of the paper focuses on showing that even when the parameter space of the exponential family is unbounded, sparse covers can be found.

**Questions:**

My central question is in the section on strength and weaknesses.

**Limitations:**

The authors have addressed limitations to my satisfaction.

**Strengths And Weaknesses:**

Learning SIIRVs has received attention and found applications in Game Theory, Stochastic optimization etc. This paper takes a step towards understanding the problem in the infinite support case which is practically important. In my view, the paper’s key technical contribution is the structural result: i.e., to identify assumptions under which SIIRVs can be covered ‘properly’. As I explain below, because of the fact that this is a strength, it is also a weakness. Having said that, the assumptions certainly cover a wide-range of practical distributions. Other technical contributions involve identifying covers even when the parameter space is unbounded which was not obvious at first glance.

The paper is well presented and clear. My only minor complaint about the presentation is not motivating the proper learning requirement well in the introduction (Section 1). Some of the motivation comes in the start of Section 3 which would have been good to know in the introduction!

While the authors somewhat justify some assumptions pointing to previous work and their necessity in their proofs, it would strengthen the paper if distributions were identified where all but one assumption fails to hold and the resultant sample complexity is shown to have some dependence on n. In other words, I wasn’t fully convinced that the assumptions that the paper proposes are minimal. Since the assumptions are in my view, the strength of the paper, some more explanation geared to showing that these are minimal would greatly strengthen the paper.

---

> ### Author Response · Authors · 2022-08-01
> **Response to Reviewer HJNQ**
>
> We thank the reviewer for their constructive feedback and suggestions. As recommended, we will further motivate the proper learning requirement in the introduction.
>
> The reviewer is right in that a more detailed discussion on the minimality of assumptions would be quite useful and we plan to include such a discussion in the final version of the paper. For Assumption 1, we have the following cases.
> + Waiving condition 1, there is a known lower bound on the sample complexity that involves some dependence on the number of terms (Observation 1.3 from [DDDST'13]). The terms considered in the lower bound all have zero as a mode (essentially satisfying condition 2 in the case of multimodal distributions) and all of their moments are upper bounded by a sequence of values (stronger than condition 3). They are, however, not unimodal and they assign almost all of their mass to zero (condition 1 does not hold).
> + Waiving condition 2 enables one to form a family which does not have a sparse cover, even when the sums only have a single term. In particular, if the modes are not localized, then we can consider a sequence of arbitrarily shifted Bernoulli distributions with parameter $1-\gamma$, each of which has a distance at least equal to $1-2\gamma$ from any other distribution of the sequence. Moreover, the aforementioned sequence does not violate conditions 1 or 3.
> + Waiving condition 3 would enable to include every geometric random variable in a valid (single-term) distribution family (no lower bound on the success probability). In this case, there is some constant $\epsilon>0$, such that we may consider an infinite sequence of geometric distributions with diminishing success probabilities ($p_n= 2^{-n}$) such that every pair of distributions in the sequence have statistical distance at least $\epsilon$. The degree of the moment we assume to be bounded is $4$, which is useful in the dense case, to establish the rate of convergence of the sum to a discretized Gaussian distribution; importantly, the degree is constant.
>
> Assumption 2 essentially introduces two new conditions; the spectral bound of the covariance of the sufficient statistics (condition 5) and the variance lower bound (condition 7). The spectral bound ensures some kind of smoothness, or, in other words, that (geometric) closeness of the parameter vectors implies (statistical) closeness of the corresponding distributions, according to a linear rate (see Lemma 14 combined with Proposition 12). This assumption is also standard in the literature of properly learning exponential families (see [SSW'21], [DKSS'21]). The variance lower bound is a relaxed version of the requirement in Assumption 1 -- Condition 1 that the mass assigned on the mode is bounded away below $1$ (the relaxation is possible due to the structure of exponential families). In the specific case of Poisson Binomial distributions (see [DP'13]) as well as the Poisson Negative Binomial distributions (see Appendix H.2), the variance lower bound can be waived completely, due to some particularly subtle results from Poisson approximation theory. It is not clear, however, whether Assumption 7 can be waived completely in the general case of SIIERVs.
>
> &nbsp;
>
> *References:*
>
> [DDDST'13] Daskalakis, C., Diakonikolas, I., ODonnell, R., Servedio, R. A., & Tan, L. Y. (2013, October). Learning sums of independent integer random variables. In 2013 IEEE 54th Annual Symposium on Foundations of Computer Science (pp. 217-226). IEEE.
>
> [SSW'21] Shah, A., Shah, D., & Wornell, G. (2021). A Computationally Efficient Method for Learning Exponential Family Distributions. Advances in Neural Information Processing Systems, 34, 15841-15854.
>
>
> [DKSS'21] Diakonikolas, I., Kane, D.M., Stewart, A. &amp; Sun, Y.. (2021). Outlier-Robust Learning of Ising Models Under Dobrushin’s Condition. Proceedings of Thirty Fourth Conference on Learning Theory, in Proceedings of Machine Learning Research 134:1645-1682 Available from https://proceedings.mlr.press/v134/diakonikolas21e.html.
>
> [DP'13] Daskalakis, Constantinos and Christos H. Papadimitriou. Sparse covers for sums of indicators. Probability Theory and Related Fields 162 (2013): 679-705.

---

> > ### Comment · Reviewer_HJNQ · 2022-08-09
> > **Response to authors**
> >
> > Thanks for the very detailed response. I am fairly convinced about the minimality of the assumptions and have increased my score to reflect that.

---

### Official Review · Reviewer_9A41 · 2022-07-11

**Rating:** 8
**Confidence:** 2
**Soundness:** 4 excellent
**Presentation:** 4 excellent
**Contribution:** 4 excellent

**Summary:**

This paper considers distributions of sums of independent, integer-valued random variables. The authors study both covering and learning the distributions arising from these sums. In contrast to prior work, the authors address the case where the random variables have infinite support. The results apply to sums of random variables sampled, where each random variable is sampled from an exponential family. The random variables appearing in the sum do not necessarily have the same distribution; rather, their parameters are assumed to come from a common parameter space. Under some assumptions on the parameter space, the authors give a covering guarantee. The covering result guarantees the existence of a set of distributions that cover the sum distribution, such that each covering distribution is drawn from an exponential family whose parameters appear in a slightly enlarged version of the original parameter space. The accompanying learning algorithm samples a random variable which is close in total variational distance to the summation distribution, and also corresponds to an enlargement of the exponential family parameter space.

**Questions:**

•	Do your results give any implications for concentration of sums of independent random variables?

•	It would have been nice to see example corollaries for certain types of random variables in the main text.

•	The authors did a great job of motivating and justifying their assumptions. Are the assumptions easy to verify? It would be nice to give sample calculations, or examples of sufficient conditions.

•	Towards the question of proper learning, the authors show that SIIRVs can be nearly properly learned, by a limited expansion of the parameter space. It would be an interesting future direction to explore nearly proper learning in other contexts; perhaps comment on this.




**Limitations:**

Yes

**Strengths And Weaknesses:**

•	This paper gives significant improvement over prior work, and is applicable to many common random variables.

•	The writing an presentation are excellent. The problem is well-motivated, with a thorough description of prior work. The techniques are well-motivated and explained at a high level.

•	Neat techniques: structural results for exponential families, cones, and distribution distance (“structural distance”). Interestingly, the results show that even when the parameter space of an exponential family is unbounded, there is a bounded subspace that approximately generates any distribution in the family. The result is proven by relating probabilistic properties of exponential families to geometric properties of cones.

•	pg. 9: compilcation → complication

---

> ### Author Response · Authors · 2022-08-01
> **Response to Reviewer 9A41**
>
> We wish to thank the anonymous reviewer for their constructive comments and for appreciating our results.
>
> *Concentration:* As the reviewer points out, our results are closely related to concentration bounds for sums of independent random variables, but our results are stronger; we show that the sums we consider are, in the dense case, statistically close to some (discretized) Gaussian distribution (and then use a continuity argument to form proper covers), which implies that they are accordingly concentrated. In the sparse case, the sums are not necessarily concentrated. We also provide concentration bounds for a single random variable within an exponential family (Lemma 9).
>
> *Example - Corollaries:* Motivated by the reviewer's suggestion, we will include some specific corollaries in the main text, which might be of independent interest. For instance, sums of geometric random variables are either statistically close to a sparse sum or close to a negative binomial random variable (see Appendix H.2).
>
> *Verification of Assumptions:* As per the reviewer's proposal, we will also include a set of examples of the verification of the assumptions. Although the assumptions cannot be verified efficiently for every selection of the sufficient statistics, assuming a simple given description of the sufficient statistics vector, analytic methods can potentially reduce the assumptions to restrictions on the space of parameters. For Assumption 2 we have the following.
> + For conditions 2 and 3 (unimodality and localization of modes) we have already identified an algebraic condition in terms of an appropriate set of linear inequalities (see Appendix E.2).
> + Condition 4 (bounded central moments) may imply a lower bound on the minimum norm of the parameter space. For instance, if the sufficient statistics is a scalar logarithmic function (corresponding to Zeta distribution), the fourth moment is bounded when the parameter takes values bounded away above $5$, according to the convergence of the zeta function (see Appendix H.1).
> + For condition 5 (spectral bound on the covariance matrix), it is sufficient to show an upper bound on the expected value of the squared norm of the sufficient statistics vector, i.e., $\mathbb{E}[\|{\vec{T}}(W)\|^2]$. Such an upper bound may correspond to the exclusion of some parameter values when different coordinates of the sufficient statistics have different behavior in the limit $x\to\infty$. For example, if one coordinate is polynomial, while another one is logarithmic, we would like to ensure that when the parameter corresponding to the polynomial statistic is zero, the other parameter will be bounded away above a value that depends on the degree of the polynomial statistic, i.e., if $\vec{T}(x) = (x^r,\log x)$ then $(0,a_2)\in\mathcal{A}$ implies $a_2\ge f(r)$, where $f$ is some appropriate (increasing) function.
> + Finally, for condition 7 (variance lower bound), consider the simple example of the Geometric distributions, where the sufficient statistics is a scalar linear function over $\mathbb{N}$ (i.e., $\vec{T}(x)=x$). Then, the variance lower bound is equivalent to an upper bound on the parameter space (i.e., $a\in\mathcal{A}$ implies $a\le a_{\max}$). We note, however, that the variance lower bound does not always imply that the parameter space is bounded. In particular, when a distribution has two or more subsequent modes, then, as the norm of the parameter increases to the limit, the variance remains bounded away above zero. Therefore, the variance lower bound may correspond to a different upper bound on the norm for each direction of the parameter space (since the parameter vector's direction defines the set of modes; see Appendix E.2).
>
> *Almost Proper Learning:* This is a very interesting question. It is well known that various hardness results (either statistical or computational) arise in various learning contexts (PAC learning, distribution learning) when proper learning is a requirement. These hardness results usually can be bypassed when this properness condition is waived and one simply has to design an improper learner/estimator. Depending on the context, it may be the case that some notion of almost-proper learning could achieve the best of both worlds: sufficient structure for the output learning model and avoidance of hardness results emerging due to properness.

---

> > ### Comment · Reviewer_9A41 · 2022-08-08
> > **Reply**
> >
> > Thank you very much for your detailed response! I also appreciated the discussion of the minimality of the assumptions in your other response to the other reviewer.

---

### Official Review · Reviewer_vvMx · 2022-07-12

**Rating:** 6
**Confidence:** 2
**Soundness:** 3 good
**Presentation:** 3 good
**Contribution:** 2 fair

**Summary:**

This paper studies the problem of learning a random variable X which can be written as the sum of n independent integer random variables X_1, ..., X_n (X is called a SIIRV). They mainly consider the question of learning X up to low total variation distance if the X_i's are "sufficiently nice" distributions. This paper's main contribution is to show that such learning can occur if the X_i's all satisfy some set of assumptions. The number of samples of sample variables X that are needed is quite low, having no dependence on the support size of the distribution (which could be unbounded), and almost no dependence on n, the number of independent variables that are summed to get X. Their techniques are based on covering arguments that demonstrate that there is a small but dense set of SIIRVs in the space of possible random variables X.

**Questions:**

My main question refers to the above, where I ask whether there are some important families of distributions that your methods can learn properly but previous methods were unable to learn. Alternatively, I would be interested in more understanding of how reasonable the assumptions in Assumption 1 and Assumption 2 are in real distributions.

**Limitations:**

Yes.

**Strengths And Weaknesses:**

1) This paper appears to be technically impressive, though it was very hard for me to judge as I am not an expert in this area.
2) The authors also mention several applications of this learning problem - while I do not know about the importance of this problem, the stated applications seem reasonable.
3) One concern I have with this problem is that there are a very large number of assumptions that need to be made about the exponential family. While many distributions satisfy these assumptions (Poisson, Bernoulli, Gaussian, etc.), there are probably much easier ways to learn sums of these random variables. (For instance, learning sums of Poissons is easy as it is just another Poisson.) So, it is unclear whether there are important families of distributions that this paper tackles but previous techniques were unable to tackle.
4) Another concern I have is that this algorithm seems highly impractical. While the sample complexity is reasonable, the runtime is exponential and therefore probably wouldn't be as valuable.

---

> ### Author Response · Authors · 2022-08-01
> **Response to Reviewer vvMx**
>
> We wish to thank the anonymous reviewer for their feedback and for appreciating our technical contributions.
>
> *Clarification:* We would first like to clarify that the sample complexity bounds we provide have no dependence on the number of terms (rather than "almost no dependence on $n$", as stated in the reviewer's comment).
>
> *Assumptions:* Our assumptions are indeed specialized, but not necessarily strict; we believe their accurate statement to be a strength rather than a weakness, since the problem is significantly general, thereby restrictions are unavoidable. Our Assumption 2 does not exclude any reasonable exponential family and our methods capture (among others) Geometric, Bernoulli, Poisson, Zeta, Gamma, (discrete) Gaussian, Laplacian distributions and interpolations thereof (see Appendix H.1). For instance, our results apply to sums with both Gaussian and Laplacian terms (semi-parametric setting). In particular, the naturally occurring Zeta distributions (Zipf's law) are not log-concave and no non-trivial learning results are known on sums thereof, even without requiring proper learning. Moreover, Assumption 1 practically includes any (non-parametric) family of unimodal (even heavy tailed) distributions with localized modes. For reference, we mention that even for the simple case of sums of independent Bernoulli random variables, there is a series of recent works that target the corresponding covering and learning problems (see [DP'13], [DDS'12] and [DKS'16]). We capture significantly more general classes of distributions, including most practical exponential families, as well as sums with terms in qualitatively different distribution classes, by exploiting the expressivity of the exponential family paradigm.
>
> *Runtime:* The runtime of the proposed learning algorithm is indeed exponential in some (but not all) of the relevant parameters. However, even in the special case of Poisson Binomial distributions (PBDs), the covering method is provably not sufficient for the design of fully polynomial learning algorithms (see [DKS'16] and [DKS'16b]). In fact, to the best of our knowledge, there are no known fully polynomial algorithms for properly learning an unknown PBD. On the contrary, there are Fourier-based algorithms that (improperly) learn PBDs efficiently (see [DDKT'16]), but it is unclear whether such methods could lead to a proper learning algorithm. We focus on the covering method for two main reasons. First, the covering method provides insights on the structure of the examined random variables (Theorem 3). Many of the results that motivated the now widely studied problem of learning SIIRVs (e.g., in Game Theory and in Stochastic Combinatorial Optimization) use the delicate structure of such sums. Our results identify such structure in a quite broader class of random variables and, importantly, demonstrate that the size of the support is not an utter impediment in acquiring such delicate results. Second, the covering method provides strong upper bounds for the sample complexity of the learning problem (Theorem 4). Our results provide sample complexity bounds for general distribution classes that are comparable to the best known bounds for the primitive class of PBDs, even though there have been results that hinted to the impossibility of acquiring such bounds.
>
> &nbsp;
>
> *References:*
>
> [DP'13] Daskalakis, Constantinos and Christos H. Papadimitriou. “Sparse covers for sums of indicators.” Probability Theory and Related Fields 162 (2013): 679-705.
>
> [DDS'12] Constantinos Daskalakis, Ilias Diakonikolas, and Rocco A. Servedio. 2012. Learning poisson binomial distributions. In Proceedings of the forty-fourth annual ACM symposium on Theory of computing (STOC '12). Association for Computing Machinery, New York, NY, USA, 709–728. https://doi.org/10.1145/2213977.2214042
>
> [DKS'16] Diakonikolas, Ilias, Daniel M. Kane and Alistair Stewart. “Properly Learning Poisson Binomial Distributions in Almost Polynomial Time.” COLT (2016).
>
> [DKS'16b] Ilias Diakonikolas, Daniel M. Kane, Alistair Stewart. Optimal Learning via the Fourier Transform for Sums of Independent Integer Random Variables. In Vitaly Feldman, Alexander Rakhlin, Ohad Shamir, editors, Proceedings of the 29th Conference on Learning Theory, COLT 2016, New York, USA, June 23-26, 2016. Volume 49 of JMLR Workshop and Conference Proceedings, pages 831-849, JMLR.org, 2016.
>
> [DDKT'16] Constantinos Daskalakis, Anindya De, Gautam Kamath, and Christos Tzamos. 2016. A size-free CLT for poisson multinomials and its applications. In Proceedings of the forty-eighth annual ACM symposium on Theory of Computing (STOC '16). Association for Computing Machinery, New York, NY, USA, 1074–1086. https://doi.org/10.1145/2897518.2897519

---

> > ### Comment · Reviewer_vvMx · 2022-08-08
> > **Response to Rebuttal**
> >
> > Thank you for describing many families of distributions that fit with your assumptions, and for describing why your assumptions for the distributions and exponential runtimes in certain parameters are perhaps necessary. I believe this would still be nicer if one could have fewer assumptions and better guarantees on the runtime, but I appreciate that obtaining such a result is perhaps very difficult or impossible. Because of this I will also increase my score slightly from a 5 to a 6.

---

### Meta-Review · Area_Chair_xeyd · 2022-08-24

**Recommendation:** Accept
**Confidence:** Certain

**Metareview:**

The authors address the fundamental problem of learning the sum of n independent random variables (not necessarily identically distributed). They concentrate on the setting where the variables have infinite support. In general, it's known that the sample complexity for this problem is not bounded, but they get around this barrier by making parametric assumptions on the distributions of the variables.

The reviewers appreciated the strength of the results, and the authors engaged with them to dispel remaining questions. It is not exactly clear where an application of the results here would make a concrete impact; the authors hint in the rebuttal that understanding the "delicate structure" of sums of independent random variables is important in game theory and stochastic combinatorial optimization, but this is not made explicit. In any case, the theoretical contributions are solid enough to merit acceptance.

**Award:**

No

---

### Decision · Program_Chairs · 2022-09-14

Accept